# Status of the Tibetan Plateau observatory (Tibet-Obs) and a 10-year (2009-2019) surface soil moisture dataset

Pei Zhang[1,2], Donghai Zheng[2], Rogier van der Velde[1], Jun Wen[3], Yijian Zeng[1], Xin Wang[4], Zuoliang Wang[4], Jiali Chen[2,5], Zhongbo Su[1]

[1]Faculty of Geo-Information Science and Earth Observation (ITC), University of Twente, Enschede, 7514AE, the Netherlands
[2]National Tibetan Plateau Data Center, Key Laboratory of Tibetan Environmental Changes and Land Surface Processes, Institute of Tibetan Plateau Research, Chinese Academy of Sciences, Beijing, 100101, China
[3]College of Atmospheric Sciences, Chengdu University of Information Technology, Chengdu, 610225, China
[4]Northwest Institute of Eco-Environment and Resources, Chinese Academy of Sciences, Lanzhou, 730000, China
[5]College of Earth and Environmental Sciences, Lanzhou University, Lanzhou, 730000, China

*Correspondence to*: Donghai Zheng (zhengd@itpcas.ac.cn), Z. (Bob) Su (z.su@utwente.nl)

**Abstract.** The Tibetan Plateau observatory of plateau scale soil moisture and soil temperature (Tibet-Obs) was established ten years ago, which has been widely used to calibrate/validate satellite- and model-based soil moisture (SM) products for their applications to the Tibetan Plateau (TP). This paper reports on the status of the Tibet-Obs and presents a 10-year (2009-2019) surface SM dataset produced based on *in situ* measurements taken at a depth of 5 cm collected from the Tibet-Obs that consists of three regional-scale SM monitoring networks, i.e. the Maqu, Naqu, and Ngari (including Ali and Shiquanhe) networks. This surface SM dataset includes the original 15-min *in situ* measurements collected by multiple SM monitoring sites of the three networks, and the spatially upscaled SM records produced for the Maqu and Shiquanhe networks. Comparisons between four spatial upscaling methods, i.e. arithmetic averaging, Voronoi diagram, time stability, and apparent thermal inertia, show that the arithmetic average of the monitoring sites with long-term (i.e. ≥ six years) continuous measurements are found to be most suitable to produce the upscaled SM records. Trend analysis of the 10-year upscaled SM records indicates that the Shiquanhe network in the western part of the TP is getting wet while there is no significant trend found for the Maqu network in the east. To further demonstrate the uniqueness of the upscaled SM records in validating existing SM products for long term period (~10 years), the reliability of three reanalysis datasets are evaluated for the Maqu and Shiquanhe networks. It is found that current model-based SM products still show deficiencies in representing the measured SM dynamics in the Tibetan grassland (i.e. Maqu) and desert ecosystems (i.e. Shiquanhe). The dataset would also be valuable for calibrating/validating long-term satellite-based SM products, evaluation of SM upscaling methods, development of data fusion methods, and quantifying the coupling of SM and precipitation at 10-year scale. The dataset is available in the 4TU.ResearchData repository at https://doi.org/10.4121/12763700.v7 (Zhang et al., 2020).

## 1 Introduction

The Tibetan Plateau observatory (Tibet-Obs) of plateau scale soil moisture and soil temperature (SMST) was setup in 2006 and became fully operational in 2010 with as objective of the calibration/validation of satellite- and model-based soil moisture (SM) products at regional scale (Su et al., 2011). The Tibet-Obs mainly consists of three regional-scale SMST monitoring networks, i.e. Maqu, Naqu, and Ngari, which cover different climate and land surface conditions across the Tibetan Plateau (TP) and each includes multiple *in situ* SMST monitoring sites. The SM data collected from the Tibet-Obs have been widely used in past decade to calibrate/validate satellite- and model-based SM products (e.g. Su et al., 2013; Zheng et al., 2015a; Colliander et al., 2017), and to evaluate and develop SM upscaling methods (e.g. Qin et al., 2013; 2015), to assess algorithms for the retrieval of SM for microwave remote sensing observations (e.g. van der Velde et al., 2014a; 2014b; Zheng et al., 2018a; 2018b; 2019) and fusion methods to merge *in situ* SM and satellite- or model-based products (e.g. Yang et al., 2020; Zeng et al., 2016).

Key information and outcomes of the main scientific applications using the Tibet-Obs SM data are summarized in Table 1. As shown in Table 1, the state-of-the-art satellite- and model-based products are useful but still show various types of deficiencies specific to the hydro-meteorological conditions on the TP, and further evaluation and improvement of these products remain imperative. In general, previous studies mainly focused on the evaluation of SM products using the Tibet-Obs data for short term period (i.e. less than five years), while up to now the Tibet-Obs has collected *in situ* measurements for more than 10 years. Development of a close to 10-year Tibet-Obs *in situ* SM dataset would further enhance the calibration/validation of long-term satellite- and model-based products, and is valuable for better understanding the hydro-meteorological response to climate change. However, SM is highly variable in both space and time, and data gaps in the availability of measurements taken from individual monitoring sites hinder scientific studies covering longer time periods, e.g. more than five years. Therefore, it is still challenging to obtain accurate long-term regional-scale SM due to the sparse nature of monitoring networks and highly variable soil conditions.

Spatial upscaling is usually necessary to obtain the regional-scale SM of an *in situ* network from multiple monitoring sites to match the footprint of satellite- or grid cell of model-based products. A frequently used approach for upscaling point-scale SM measurements to a spatial domain is the arithmetic average, mostly because of its simplicity (Su et al. 2011; 2013). Many other studies also adopted weighted averaging methods, whereby the weights are assigned to account for spatial heterogeneity in the area covered by *in situ* monitoring sites within the network. For instance, Colliander et al. (2017) employed Voronoi diagrams to determine the weights of individual monitoring sites within core regional-scale networks used for the worldwide validation of the Soil Moisture Active/Passive (SMAP) SM products. Dente et al. (2012a) established weights based on the topography and soil texture for the sites of the Tibet-Obs' Maqu network. Qin et al. (2013, 2015) derived the weights by minimizing a cost function between *in situ* SM of individual monitoring sites and a representative SM of the network that is estimated using the apparent-thermal-inertia-based (ATI) method (Gao et al., 2017). Alternative methods, such as time stability and ridge regression, have

been adopted in other investigations (i.e. Zhao et al., 2013, Kang et al., 2017). While a large number of studies have assessed the performance of different upscaling methods in other areas such as the Tonzi Ranch network in California and the Heihe watershed (Moghaddam et al., 2014, Wang et al., 2014), only a few investigations have been done for the TP (Gao et al., 2017, Qin et al., 2015). Since the number of monitoring sites changes over time due to damage of SM sensors in the Tibet-Obs, it is essential to evaluate and select an appropriate upscaling method for a limited number of monitoring sites (i.e. ≤ four sites).

This paper reports on the status of the Tibet-Obs and presents a long-term *in situ* SM and spatially upscaled SM dataset for the period between 2009 and 2019. The 10-year SM dataset of Tibet-Obs includes the original 15-min *in situ* measurements taken at a depth of 5 cm collected from the three regional-scale networks (i.e. Maqu, Naqu, and Ngari as shown in Fig. 1), and the continuous regional-scale SM produced using an appropriately selected spatial upscaling method. To achieve this, four methods are studied namely the arithmetic average (AA), Voronoi diagram (VD), time stability (TS), and apparent thermal inertia (ATI) methods. The seasonal dynamic and trend of the regional-scale SM time series are analysed and the 10-year SM dataset is used to validate three model-based SM products, e.g. ERA5-land (Muñoz-Sabater et al., 2018), MERRA2 (Modern-Era Retrospective Analysis for Research and Applications, version 2) (GMAO, 2015), and GLDAS Noah (Global Land Data Assimilation System with Noah Land Surface Model) (Rodell et al., 2004).

This paper is organized as follows. Section 2 describes the status of the Tibet-Obs and the *in situ* SM measurements, as well as the precipitation data and the three model-based SM products. Section 3 introduces the four SM spatial upscaling methods, Mann Kendall trend test and Sen's slope estimate, and performance metrics. Section 4 presents the inter-comparison of the four SM spatial upscaling methods, the production and analysis of regional-scale SM dataset for a 10-year period, and its application to validate the three model-based SM products. Section 5 provides the discussion and suggestions for maintaining Tibet-Obs. Section 6 documents the information on data availability and the conclusions are drawn in Section 7.

**2 Data**

**2.1 Status of the Tibet-Obs**

The Tibet-Obs consists of the Maqu, Naqu, and Ngari (including Shiquanhe and Ali) regional-scale SMST monitoring networks (Fig. 1) that cover the cold humid climate, cold semiarid climate, and cold arid climate, respectively. Each network includes a number of monitoring sites that measure the SMST at different soil depths. Brief descriptions of each network and corresponding surface SM measurements taken at a depth of 5 cm are given in following subsections. The readers are referred to the existing literature (Su et al., 2011; Dente et al. 2012a; Zhao et al., 2018) for additional information on the networks.

### 2.1.1 Maqu network

The Maqu network is located in the north-eastern edge of the TP (33°30'-34°15'N, 101°38'-102°45'E) at the first major bend of the Yellow River. The landscape is dominated by the short grass at elevations varying from 3400 to 3800 m. The climate type is characterized as cold-humid with cold dry winters and rainy summers. The mean annual air temperature is about 1.2 ℃, with -10 ℃ for the coldest month (January) and 11.7 ℃ for the warmest month (July) (Zheng et al., 2015a). The annual precipitation is about 600 mm that falls mainly in the warm season (May-October).

The Maqu network covers an area of approximately 40 km by 80 km and consists originally of 20 SMST monitoring sites installed in 2008 (Dente et al. 2012a). During the period between 2014 and 2016, eight new sites were installed due to the damage of several old monitoring sites by local people or animals. The basic information of each monitoring site is summarized in Table A1 (Su et al., 2011), and the typical characteristics of topography and land cover within the network are shown in Fig. 2 as well.

The Decagon 5TM ECH$_2$O probes are used to measure the SMST at nominal depths of 5, 10, 20, 40, and 80 cm (Fig. 3). The 5TM probe is a capacitance sensor measuring the dielectric permittivity of soil, and the Topp equation (Topp et al., 1980) is used to convert the dielectric permittivity to the volumetric SM. The accuracy of the 5TM volumetric SM was improved via a soil-specific calibration performed under laboratory conditions for each soil type found in the Maqu area (Dente et al. 2012a), leading to a decrease in the root mean square error (RMSE) from 0.06 to 0.02 m$^3$ m$^{-3}$ (Dente et al. 2012a). Table 2 provides the specific periods of data missing during each year and the total data lengths of surface SM for each monitoring site. Among these sites, the CST05, NST01, and NST03 have collected more than nine years of SM measurements, while the data records for the NST21, NST22, and NST31 are less than one year. In May 2019, there were still 12 sites that provided SM data.

### 2.1.2 Ngari network

The Ngari network is located in the western part of the TP at the headwater of the Indus River. It consists of two SMST networks established around the cities of Ali and Shiquanhe, respectively. The landscape is dominated by a desert ecosystem at elevations varying from 4200 to 4700 m. The climate is characterized as cold-arid with a mean annual air temperature of 7.0 ℃. The annual precipitation is less than 100 mm that falls mainly in the monsoon season (July-August) (van der Velde et al., 2014b).

The Shiquanhe network consisted originally of 16 SMST monitoring sites installed in 2010 (Su et al. 2011), and five new sites were installed in 2016. The basic information of each monitoring site is summarized in Table A3 (Su et al., 2011), and the typical characteristics of topography and land cover within the network are also shown in Fig. 4. The Decagon 5TM ECH$_2$O probes were installed at depths of 5, 10, 20, 40, and 60/80 cm to measure the SMST (Fig. 3). Table 3 provides the specific periods of data missing during each year and the total data lengths of surface SM for each site. Among these sites, the SQ02, SQ03, SQ06, and SQ14 have collected more than eight years of SM measurements, while the data records for the SQ13, SQ15, and SQ18 are less than two years. In August 2019, there were still 12 sites that provided SM data. The Ali

network comprises of four SM monitoring sites (Table A3), which will not be used for further analysis in

this study due to limited number of monitoring sites and the availability of data records (Table 3).

**2.1.3 Naqu network**

The Naqu network is located in the Naqu river basin with an average elevation of 4500 m. The climate is

characterized as cold semi-arid with cold dry winters and rainy summers. Over three-quarters of the total

annual precipitation sum (400 mm) falls between June and August (Su et al., 2011). The landscape is

dominated by short grass.

The network consists originally of five SMST monitoring sites installed in 2006 (Su et al. 2011), and six new

sites were installed between 2010 and 2016. The basic information of each monitoring site is summarized in

Table A5, and the typical topography and land cover within the network are shown in Fig. 5 as well. The

Decagon 5TM ECH$_2$O probes were installed at depths of 5/2.5, 10/7.5, 15, 30, and 60 cm to measure the

SMST, and an on-site soil-specific calibration is reported in van der Velde (2010) and yielded a RMSE of

0.029 m$^3$ m$^{-3}$. Table 4 provides the specific periods of data missing during each year and the total data lengths

of surface SM for each site. Among these sites, only two sites (Naqu and MS sites in Table A5) have collected

SM measurements for more than six years, while the data records for the others are less than four years.

Similar to the Ali network, the Naqu network will also not be used for the further analysis in this study due

to limited number of monitoring sites and the availability of data records.

**2.2 Precipitation data**

Precipitation data is available from the dataset of daily climate data from Chinese surface meteorological

stations. This dataset is maintained by the China Meteorological Administration (CMA) and based on the

measurements from 756 basic and reference surface meteorological observation and automatic weather

stations (AWS) in China from 1951 to present. The online dataset mainly includes seven meteorological

variables such as air pressure, air temperature, relative humidity, wind speed, evaporation, sunshine duration,

and precipitation. The precipitation data from two weather stations (see Fig. 1), i.e. Maqu (34°00'N,

102°05'E) and Shiquanhe (32°30'N, 80°05'E) are used in this study. The available daily precipitation is the

cumulative value for the period between 20h of previous day and 20h of current day at Beijing time, which

is available from https://data.cma.cn/data/detail/dataCode/SURF_CLI_CHN_MUL_DAY.html (last access

11 March 2021). The daily precipitation is summed up for each month to obtain the monthly cumulative

value in this study, which can be found at https://doi.org/10.4121/12763700.v7 (last access 16 April 2021).

The monthly precipitation data for the period between 2009 and 2019 is mainly used in this study for the

trend analysis (see Section 4.2).

### 2.3 Model-based soil moisture products

### 2.3.1 ERA5-land soil moisture product

ERA5-land is a reanalysis dataset produced by running land component of the ECMWF (European Centre for Medium-Range Weather Forecasts) ERA5 climate model (Albergel et al., 2018). ERA5-land provides SM data currently available from 1981 to present for every hour with a spatial resolution of 0.1°, and the data is available from https://cds.climate.copernicus.eu/cdsapp#!/dataset/reanalysis-era5-land?tab (last access 11 March 2021). More information about the ERA5-land product readers are referred to Muñoz-Sabater et al., (2018). The data of volumetric total soil water content for the top soil layer (0-7 cm) is used in this study.

### 2.3.2 MERRA2 soil moisture product

MERRA2 is an atmospheric reanalysis dataset produced by NASA using the Goddard Earth Observing System Model version 5 (GEOS-5) and atmospheric data assimilation system (ADAS), version 5.12.4. MERRA2 provides SM data currently available from 1980 to present at hourly time interval and spatial resolution of 0.5° (latitude) by 0.625° (longitude). The data is available from https://disc.gsfc.nasa.gov/datasets/M2T1NXLND_5.12.4/summary (last access 11 March 2021). For more information about the MERRA2 product readers are referred to GMAO (2015). The liquid volumetric soil water content of the surface layer (0-5 cm) is used in this study.

### 2.3.3 GLDAS Noah soil moisture product

GLDAS-2.1 Noah is a combination of model-based and satellite observed meteorological data, such as Global Precipitation Climatology Project (GPCP) version 1.3, forced onto the Noah Model 3.6 in Land Information System (LIS) version 7 to simulate water and energy exchanges between land and atmosphere. GLDAS-2.1 Noah provides SM data currently available from 2000 to present at a 3-hourly time interval with a spatial resolution of 0.25°. The data is available from https://disc.gsfc.nasa.gov/datasets/GLDAS _NOAH025_3H_2.1/summary (last access 11 March 2021). More details on the GLDAS Noah product can be found in Rodell et al. (2004). The liquid soil water content of the top soil layer (0-10 cm) is used in this study.

### 3 Methods

### 3.1 Spatial upscaling of soil moisture measurements

The principle of spatial upscaling a set of point measurements to an area is based on assigning weights to individual sites, often using additional information, in such way that the selected collection is representative for the selected domain. The method can in its simplest form be represented by a linear equation mathematically as follows:

$$\overline{\theta}_t^{ups} = \theta_t^{obs}\boldsymbol{\beta} \tag{1a}$$

$\boldsymbol{\theta}_t^{obs} = [\boldsymbol{\theta}_{t,1}^{obs}, \boldsymbol{\theta}_{t,2}^{obs}, \dots, \boldsymbol{\theta}_{t,N}^{obs}]^T$                                                              (1b)

where $\overline{\boldsymbol{\theta}}_t^{ups}$ [m³ m⁻³] represents the upscaled SM, $\boldsymbol{\theta}_t^{obs}$ [m³ m⁻³] represents the vector of SM measurements, $N$

represents the total number of SM monitoring sites, $t$ represents the time (e.g. the $t^{th}$ day), and $\beta$ [-] represents

the vector with weights.

In this study, only the surface SM measurements taken from the Maqu and Shiquanhe networks are upscaled

to obtain the regional-scale SM for 10-year (2009-2019) periods due to the availability of much longer

records in comparison to the Naqu and Ali networks (see Section 2.1). Four upscaling methods are

investigated and inter-compared with each other to find the most suitable method for the application to the

Tibet-Obs. Brief descriptions of the selected upscaling methods are given in Appendix B. The arithmetic

average (hereafter "AA") assigns an equal weight coefficient to each SM monitoring site (see Appendix B.1),

and the Voronoi diagram (hereafter "VD") determines the weight based on the geographic distribution of all

the SM monitoring sites (see Appendix B.2). The time stability method (hereafter "TS") regards the most

stable site as representative site for the network (see Appendix B.3), and the apparent thermal inertia (ATI)

method is based on the close relationship between apparent thermal inertia ($\tau$) and SM (see Appendix B.4).

**3.2 Trend analysis**

The Mann-Kendall test and Sen's slope estimate (Gilbert, 1987; Mann, 1945; Smith et al., 2012) are adopted

to analyze the trend of the 10-year time series for the upscaled SM, model-based SM products (i.e. ERA5-

land, GLDAS Noah, and MERRA2), and precipitation. Specifically, the trend analysis is based on the

monthly data, and all the missing data is regarded as an equal value smaller than other valid data. The test

consists of calculating the seasonal statistics S and its variance VAR(S) separately for each month during the

10-year period, and the seasonal statistics are then summed to obtain the Z metric.

For month $i$ (e.g. January), the statistics $S_i$ can be computed as:

$S_i = \sum_{k=1}^{9} \sum_{l=k+1}^{10} sgn(X_{i,l} - X_{i,k})$                                                          (2a)

$sgn(X_{i,l} - X_{i,k}) = \begin{cases} 1 & X_{i,l} > X_{i,k} \\ 0 & X_{i,l} = X_{i,k} \\ -1 & X_{i,l} < X_{i,k} \end{cases}$

where $k$ and $l$ represent the different year and $l > k$, $X_{i,l}$ and $X_{i,k}$ represent the monthly value of the variable

for the month $i$ of the year $k$ and $l$, respectively.

The $VAR(S_i)$ is computed as:

$VAR(S_i) = \frac{1}{18} [N_i(N_i - 1)(2N_i + 5) - \sum_{p=1}^{g_i} t_{i,p}(t_{i,p} - 1)(2t_{i,p} + 5)]$                      (2b)

where $N_i$ is the length of the record for the month $i$ (e.g. the 10 year data record in this study with $N_i$=10),

$g_i$ is the number of equal-value data in month $i$, $t_{i,p}$ is the number of equal-value data in the $p^{th}$ group for

235     month $i$.

After obtaining the $S_i$ and $VAR(S_i)$, the statistic $S'$ and $VAR(S')$ for the selected season (e.g. warm season

is from May up to October and cold season is from November to April) can be summed as:

$S' = \sum_{i=1}^{M} S_i$                                                                 (2c)

$VAR(S') = \sum_{i=1}^{M} VAR(S_i)$                                                                   (2d)

where M represents the number of months in the selected season, e.g. M is 12 for the full year, and M is 6

for the warm and cold seasons.

Subsequently, the Z metric can be computed as:

$Z = \begin{cases} \dfrac{S'-1}{\sqrt{Var(S')}} & if\ S' > 0 \\ 0 & if\ S' = 0 \\ \dfrac{S'+1}{\sqrt{Var(S')}} & if\ S' < 0 \end{cases}$                                 (2e)

If the statistics $Z$ is positive (negative) and its absolute value is greater than $Z_{1-\alpha/2}$ (here $\alpha = 0.05$, $Z_{1-\alpha/2} =$

1.96), the trend of the time series is regarded as upward (downward) at the significance level of $\alpha$. Otherwise,

we accept the hypothesis that no significant trend is found.

If the trend shows upward or downward, we will further estimate the slope (change per unit time) with Sen's

method (Sen, 1968). The slopes of each month can be calculated as:

$Q_i = \dfrac{X_{i,l} - X_{i,k}}{l-k}$                                                                   (2f)

Then rank all the individual slopes ($Q_i$) for all months and find the median, which is considered as the

seasonal Kendall slope estimate.

**3.3 Comparison metrics**

The metrics used to evaluate the accuracy of the upscaled SM are the bias [$m^3\ m^{-3}$], RMSE [$m^3\ m^{-3}$], and

unbiased RMSE (ubRMSE [$m^3\ m^{-3}$]), which can be formulated as:

Bias $= \dfrac{\sum_{t=1}^{M}(\theta_t^{tru} - \overline{\theta}_t^{ups})}{M}$                                                    (3a)

RMSE $= \sqrt{\dfrac{\sum_{t=1}^{M}(\theta_t^{tru} - \overline{\theta}_t^{ups})^2}{M}}$                                              (3b)

ubRMSE $= \sqrt{RMSE^2 - BIAS^2}$                                                   (3c)

where $\theta_t^{tru}$ represents the SM that is considered as the ground truth, and $\overline{\theta}_t^{ups}$ represents the upscaled SM.

The closer the metric is to zero, the more accurate the estimation is.

The metric used to assess the correlation between two time series is the Nash-Sutcliffe efficiency coefficient

(NSE [-]), expressed by:

NSE $= 1 - \dfrac{\sum_{t=1}^{n}(\theta_t^{tru} - \overline{\theta}_t^{ups})^2}{\sum_{t=1}^{n}(\theta_t^{tru} - \overline{\theta_t^{tru}})^2}$                                        (4)

The value of the NSE ranges from $-\infty$ to 1, and the closer the metric is to 1, the better the match of the

estimated SM with the reference ($\theta_t^{tru}$).

The metrics used to define the most representative SM time series (i.e. the best upsclaed SM) is the

comprehensive evaluation criterion (*CEC* [-]) obtained by combining the mean relative difference (*MRD* [-

]) and standard deviation of the relative difference ($\sigma(RD)$ [-]) (Jacobs et al., 2004). Detailed description of

above mentioned three metrics are given in Appendix B.3. It should be noted that the $\theta_{t,i}^{obs}$ and $\overline{\theta}_t^{obs}$ in Eqs.

(B4) and (B5) represent the upscaled SM using four different methods and their average when using the *CEC*

to determine the best upscaled SM. The most representative time series is identified by the lowest *CEC* value.

### 3.4 Preprocessing of model-based soil moisture products

The performance of the ERA5-land, MERRA2, and GLDAS Noah SM products are assessed using the

upscaled SM data of the Maqu and Shiquanhe networks for a 10-year period. The corresponding regional-

scale SM for each product has been obtained by averaging the data from all the grid cells falling in the

respective network areas. The numbers of grid cells covering the Maqu and Shiquanhe networks are 77 and

20 for the ERA5-land product, 12 and 4 for the GLDAS Noah product, and only one for the MERRA2

product. For the ERA5-land and MERRA2 products the data available at hourly and 3-hourly time steps are

averaged to daily value and the units of GLDAS Noah SM is converted from kg m$^{-2}$ to m$^3$ m$^{-3}$. Further it

should be noted that the uppermost soil layer of the ERA5-land (0-7 cm), MERRA2 (0-5 cm), and GLDAS

Noah (0–10 cm) SM products are assumed to match the *in situ* observations at depth of 5 cm considering the

4 cm influence zone found under laboratory conditions for the 5TM sensor by Benninga et al. (2018).

## 4 Results

### 4.1 Inter-comparison of soil moisture upscaling methods

In this section, four upscaling methods (see Section 3.1) are inter-compared first with the input of the

maximum number of available SM monitoring sites for a single year in the Maqu and Shiquanhe networks

to find the most suitable upscaled SM that can best represent the areal conditions (i.e. ground truth, $SM_{truth}$).

Later on, the performance of the four upscaling methods is further investigated with the input of reducing

number of SM monitoring sites to find the most suitable method for producing long-term (~10 year) upscaled

SM for the Maqu and Shiquanhe networks.

Fig. 6 shows the time series of daily average SM for the Maqu and Shiquanhe networks produced by the four

upscaling methods based on the maximum number of available SM monitoring sites (hereafter "$SM_{AA-max}$",

"$SM_{VD-max}$", "$SM_{TS-max}$", and "$SM_{ATI-max}$"). Two different periods are selected for the two networks due to the

fact that the number of available monitoring sites reaches the maximum in different periods for the two

networks, e.g. 17 sites for Maqu between November 2009 and October 2010 and 12 sites for Shiquanhe

between August 2018 and July 2019, respectively (see Tables A2 and A4 in the Appendix A). For the Maqu

network, the $SM_{AA-max}$, $SM_{VD-max}$, and $SM_{TS-max}$ are comparable to each other, while the $SM_{ATI-max}$ deviates

substantially during the winter (between December and February) and summer periods (between June and

August). On the other hand, the $SM_{ATI-max}$ for the Shiquanhe network is comparable to $SM_{AA-max}$ and $SM_{VD-max}$,

while $SM_{TS-max}$'s behavior is clearly different from the others. It seems that the ATI method performs

better in the Shiquanhe network due to the existence of a stronger relationship between $\tau$ and $\theta$ in the desert

ecosystem.

Table B1 lists the values of *MRD* (see Eq. (B4) in Appendix B), $\sigma(RD)$ (Eq. (B3)), and *CEC* (Eq. (B6))

calculated for the upscaled SM produced by the four upscaling methods. The *CEC* is used here to determine

the most suitable upscaled SM that can best represent the areal conditions for the two networks. It can be

found that the $SM_{AA\text{-}max}$ yields consistently the lowest *CEC* values for both networks, indicating that the

$SM_{AA\text{-}max}$ can be used to represent actual areal conditions, which will thus be regarded as the ground truth for

following analysis (i.e. $SM_{truth}$). The arithmetic average of the dense *in situ* measurements was also used as

the ground truth in other studies (Qin et al., 2013; Su et al., 2013) and found to yield reliable results by van

der Velde et al. (2021).

As shown in Tables A2 and A4 (see Appendix A), the number of available SM monitoring sites decreased as

time progressed. There are only three (i.e. CST05, NST01, and NST03) and four (i.e. SQ02, SQ03, SQ06,

and SQ14) monitoring sites that provided more than nine years of *in situ* SM measurement data for the Maqu

and Shiquanhe networks, respectively (see Tables 2 and 3). This indicates that the minimum number of

available monitoring sites can be used to produce the long-term (~10 year) consistent upscaled SM are three

and four for the Maqu and Shiquanhe networks, respectively. Fig. 7 shows the daily average SM time series

produced by the four upscaling methods based on the minimum available monitoring sites (hereafter "AA-

318     min", "TS-min", "VD-min", and "ATI-min"). The $SM_{truth}$ obtained by the AA-max is also shown for

comparison purposes. For the Maqu network, the upscaled SM produced by the AA-min, VD-min, and TS-

320     min generally capture well the $SM_{truth}$ variations, while the upscaled SM of the ATI-min shows dramatic

deviations. Similarly, the upscaled SM produced by the AA-min and VD-min are consistent with the $SM_{truth}$

for the Shiquanhe network with slight overestimations, while significant deviations are noted for the upscaled

SM of the TS-min and ATI-min. Table B2 lists the error statistics (e.g. Bias, RMSE, ubRMSE, and NSE)

computed between the upscaled SM produced by these four upscaling methods with the input of the minimum

available sites and the $SM_{truth}$. The upscaled SM produced by the AA-min shows better performance for both

networks as indicated by the lower RMSE and higher NSE values in comparison to the other three upscaling

methods.

Apart from the maximum and minimum number of available SM monitoring sites mentioned above, there

are about 14, 10, 8, and 6 available monitoring sites during different time spans for the Maqu network, and

for the Shiquanhe network are about 11, 10, 6, and 5 available monitoring sites (see Tables A2 and A4 in the

Appendix A). Fig. B2 shows the radar diagram of error statistics (i.e. RMSE and NSE) computed between

the $SM_{truth}$ and the upscaled SM produced by the four upscaling methods for different numbers of available

monitoring sites. For the Maqu network, the performances of the AA and VD methods are better than the TS

and ATI methods as indicated by smaller RMSEs and higher NSEs for all the estimations. A similar

conclusion can be drawn for the Shiquanhe network, while the performance of the ATI method is largely

improved when the number of available monitoring sites is not less than 10. It is interesting to note that the

upscaled SM produced by the AA-min is comparable to those obtained with more sites (e.g. 10 sites) as

indicated by comparable RMSE and NSE values for both networks. It indicates that the AA-min is suitable

to produce long-term (~10 years) upscaled SM for both networks, which yield RMSEs of 0.022 and 0.011

$m^3$ $m^{-3}$ for the Maqu and Shiquanhe networks in comparison to the $SM_{truth}$ produced by the AA-max based

on the maximum available monitoring sites.

**4.2 Long-term analysis of upscaled soil moisture measurements**

In this section, the AA-min is first adopted to produce the consecutive upscaled SM time series (hereafter

"$SM_{AA-min}$") for approximately an 10-year period for the Maqu and Shiquanhe networks, respectively. In

addition, the other time series of upscaled SM are produced by the AA method with input of all available SM

monitoring sites regardless of the continuity (hereafter "$SM_{AA-valid}$"), which is widely used to validate the

various SM products (Dente et al. 2012a; Chen et al. 2013; Zheng et al. 2018b) for short periods (e.g. $\leq$ 2

348 years). This method may, however, leads to inconsistent SM time series for a long-term period due to the fact

that the number of available sites is different in distinct periods (see Tables A2 and A4 in the Appendix A).

Trend analyses (see Section 3.2) are applied to both $SM_{AA-min}$ and $SM_{AA-valid}$ to investigate the impact of

changes of available SM monitoring sites on the long-term (i.e. 10-year) trend.

Fig. 8a shows the time series of $SM_{AA-min}$ and $SM_{AA-valid}$ along with the daily precipitation data for the Maqu

network during the period between May 2009 and May 2019. Both two time series of the SM show similar

seasonality with low values in winter due to frozen soils and high values in summer due to rainfall (see

subplot of Fig. 8a). Deviations can be found between the $SM_{AA-min}$ and $SM_{AA-valid}$ especially for the period

between 2014 and 2019, whereby the $SM_{AA-valid}$ tends to produce smaller SM values in the warm season. Fig.

9a shows further the Mann Kendall trend test and Sen's slope estimate for the $SM_{AA-min}$, $SM_{AA-valid}$, and

precipitation of the Maqu network area for the full year, warm seasons, and cold seasons in a 10-year period.

As described in Section 3.2, the time series would present a monotonous trend if the absolute value of

statistics $Z$ is greater than a critical value, i.e. $Z_{0.05}$ = 1.96 in this study. The results show that there is not

significant trend found for both precipitation and $SM_{AA-min}$ time series, while the $SM_{AA-valid}$ shows a drying

trend with a Sen's slope of -0.008 for warm seasons. The drying trend of the $SM_{AA-valid}$ is caused by the

change of available SM monitoring sites (see Table A2). Specifically, several monitoring sites (e.g. NST11-

NST15) located in the wetter area were damaged since 2013, and four new monitoring sites (i.e. NST21-

NST25) were installed in the drier area in 2015 (see Table 2), which affects the trend of the $SM_{AA-valid}$.

Fig. 8b shows the time series of the $SM_{AA-min}$ and $SM_{AA-valid}$ along with the daily precipitation data for the

Shiquanhe network during the period between August 2010 and August 2019. Both time series of the SM

display a similar seasonality as found for the Maqu network (see subplot of Fig. 8b). However, obvious

deviations can be noticed for the inter-annual variations, and the $SM_{AA-valid}$ tends to produce lager values

before 2014 but smaller values since then. The Mann Kendall trend test and Sen's slope estimate for the

$SM_{AA-min}$, $SM_{AA-valid}$, and precipitation time series of the Shiquanhe network area are shown in Fig. 9b. The

$SM_{AA-min}$ demonstrates a wetting trend with a Sen's slope of 0.003, while an opposite drying trend is found

for the $SM_{AA-valid}$ due to a change in number of available SM monitoring sites (see Table A4) similar to the

results from the Maqu network. Specifically, several monitoring sites (e.g. SQ11 and SQ12) located in the

wetter area were damaged around 2014, and five new monitoring sites (i.e. SQ17-21) were installed in the

drier area in 2016 (see Table 3).

In summary, the $SM_{AA\text{-}valid}$ is likely affected by the change of available SM monitoring sites over time that

leads to inconsistent trend with the $SM_{AA\text{-}min}$. This indicates that the $SM_{AA\text{-}min}$ is superior to the $SM_{AA\text{-}valid}$ for

the production of the long-term consistent upscaled SM time series.

**4.3 Application of the long-term upscaled soil moisture to validate the model-based products**

In this section, the long-term upscaled SM time series (i.e. $SM_{AA\text{-}min}$) produced for the two networks are

applied to validate the reliability of three model-based SM products, i.e. ERA5-land, MERRA2, and GLDAS

Noah, to demonstrate the uniqueness of this dataset for validating existing reanalysis datasets for a long term

period (~10 years). Since the ERA5-land product provides only total volumetric soil water content, the period

when the soil is subject to freezing and thawing (i.e. November-April) is excluded for this evaluation.

Fig. 10a shows the time series of $SM_{AA\text{-}min}$ and daily average SM data derived from the three products for the

Maqu network during the period between May 2009 and May 2019. The error statistics, i.e. bias and RMSE,

computed between the three products and the $SM_{AA\text{-}min}$ for both warm (May-October) and cold seasons

(November-April) are given in Table 5. Although the three products generally capture the seasonal variations

of the $SM_{AA\text{-}min}$, the magnitude of the temporal SM variability is underestimated. Both GLDAS Noah and

MERRA2 products underestimate the SM measurements during the warm season leading to biases of about

392    -0.112 and -0.113 $m^3$ $m^{-3}$, respectively. This may be due to the fact that the LSMs adopted for producing

these products do not consider the impact of vertical soil heterogeneity caused by organic matter contents

that is widely present in the soil Tibetan surface (Chen et al., 2013; Zheng et al., 2015a). In addition, the

MERRA2 product overestimates the SM measurements during the cold season with bias of about 0.006 $m^3$

$m^{-3}$. The ERA5-land product is able to capture the magnitude of $SM_{AA\text{-}min}$ dynamics in the warm season but

has a larger volatility and yields a RMSE of about 0.067 $m^3$ $m^{-3}$. The trend analysis for the three model-based

SM products are shown in Fig. 9a as well. All three products do not show significant trend in warm seasons

as the $SM_{AA\text{-}min}$, while the GLDAS Noah and MERRA2 products show a wetting trend in cold seasons that

is in disagreement with the $SM_{AA\text{-}min}$ trend.

Fig. 10b shows the time series of $SM_{AA\text{-}min}$ and daily SM data derived from the three products for the

Shiquanhe network area during the period between August 2010 and August 2019, and the corresponding

error statistics are given in Table 5 as well. Although the three products generally capture the seasonal

variations of the $SM_{AA\text{-}min}$, both GLDAS Noah and MERRA2 products overestimate the $SM_{AA\text{-}min}$ during the

entire study period leading to positive biases, and also positive bias (about 0.002 $m^3$ $m^{-3}$( is found in the

ERA5-land product for the warm season. The trend analyses for the three SM products are also shown in Fig.

9b. Both the ERA5-land and MERRA2 products are able to reproduce the wetting trend found for the $SM_{AA\text{-}}$

$_{min}$, while the GLDAS Noah product is not able to capture the trend.

In summary, the currently model-based SM products do not provide a reliable representation of the trend and
the dynamics of measured SM on the long-term (~10 years) in the grassland and desert ecosystems that
dominate the Tibetan landscape.

**5 Discussion**

As shown in previous sections, the number of available SM monitoring sites in the Tibet-Obs generally
changes with time. For instance, several monitoring sites of the Maqu network located in the wetter area were
damaged since 2013, and four new monitoring sites were installed in the drier area in 2015 that affects the
trend of SM time series (i.e. $SM_{AA-valid}$ shown in Section 4.2). On the other hand, the 10-year upscaled SM
data (i.e. $SM_{AA-min}$) produced in this study utilizing three and four monitoring sites with long-term continuous
measurements would yield RMSEs of about 0.022 and 0.011 $m^3$ $m^{-3}$ for the Maqu and Shiquanhe networks,
respectively (see Section 4.1). Therefore, to provide a higher-quality continuous SM time series for the future,
it is necessary to find an appropriate strategy to maintain the monitoring sites of Tibet-Obs. This section
discusses the possible strategies with the Maqu and Shiquanhe networks as examples.
At first, a sensitivity analysis is conducted to quantify the impact of the number of monitoring sites on the
regional SM estimate. The SM time series described in Section 4.1 (i.e. 11/2009-10/2010 for the Maqu
network and 8/2018-7/2019 for the Shiquanhe network) is used to test the sensitivity, and there are in total
17 and 12 available monitoring sites for the Maqu and Shiquanhe networks, respectively. Taking the Maqu
network as an example, we randomly pick different numbers of sites from 1 to 16 of the 17 sites to make up
different combinations, and then compute the RMSEs of the averaged SM obtained with these combinations
(Famiglietti et al., 2008; Zhao et al., 2013). These RMSEs are further grouped into nine levels ranging from
0.004 to 0.02 $m^3$ $m^{-3}$, and the percentage of the combinations falling into each level is summarized in Table
6. In general, the percentage increases with increasing number of monitoring sites at any RMSE levels. It can
be noted that more than 50% of combinations are able to comply with the RMSE requirement of 0.004 $m^3$
$m^{-3}$ if the number of available monitoring sites are 16 and 11 in the Maqu and Shiquanhe networks,
respectively. If the number of available monitoring sites are more than 13 and 6 in the Maqu and Shiquanhe
networks, there are about 60% of combinations with 13 sites (6 sites ) are able to comply with the RMSE
requirement of 0.01 $m^3$ $m^{-3}$. For an RMSE of 0.02 $m^3$ $m^{-3}$, more than 50% of combinations complies with
this requirement if the number of available monitoring sites is more than 7 and 3 for the two networks,
respectively. In summary, the number of monitoring sites required to maintain current networks depends on
the defined RMSE requirement.
As shown in Section 4.1, the usage of a minimum number of sites (i.e. three for Maqu and four for Shiquanhe)
with about 10-year continuous measurements yields RMSEs of 0.022 and 0.011 $m^3$ $m^{-3}$ for the Maqu and
Shiquanhe networks, respectively. Since there are still 12 monitoring sites providing SM measurements for
both networks until 2019 (see Tables 2 and 3), it is possible to decrease the RMSEs when the selected
permanent monitoring sites are appropriately determined. For the Shiquanhe network, the optimal strategy is
to keep the current 12 monitoring sites, which is exactly the combination used in Section 4.1. For the Maqu

network, it can be found that there is about 3.52% of combinations with 12 sites could yield the minimum

RMSE of 0.006 $m^3$ $m^{-3}$ (see Table 6). In order to find the optimal combination with 12 sites for the Maqu

network, all the possible combinations (i.e. the number of 6188) are ranked by RMSE values from the

smallest to largest, and Table 7 lists the examples of ranking 1-5th and 95-100th. It can be noted that the 100th

combination contains the largest number of currently available monitoring sites (i.e. 7 sites including CST03,

CST05, NST01, NST03, NST05, NST06, and NST10) with a RMSE of less than 0.006 $m^3$ $m^{-3}$. Therefore,

the 100th combination of 12 monitoring sites (as shown in Table 7) is suggested for the Maqu network.

In summary, it is suggested to maintain the current 12 monitoring sites for the Shiquanhe network, while for

the Maqu network it is suggested to restore five old monitoring sites, i.e. CST02, NST11, NST13, NST14,

and NST15.

**6 Data availability**

The 10-year (2009-2019) surface SM dataset is freely available from the 4TU.ResearchData repository at

https://doi.org/10.4121/12763700.v7 (Zhang et al., 2020). The original *in situ* SM data, the upscaled SM data,

and the supplementary data are stored in .xlsx files. A user guide document is given to introduce the content

of the dataset, the status of the Tibet-Obs, and the online dataset utilized in the study.

**7 Conclusions**

In this paper, we report on the status of the Tibet-Obs and present the long-term *in situ* SM and spatially

upscaled SM dataset for the period 2009-2019. In general, the number of available SM monitoring sites

decreased over time due to damage of sensors. Until 2019, there are only three and four sites that provide an

approximately 10-year consistent SM time series for the Maqu and Shiquanhe networks, respectively.

Comparisons between four upscaling methods, i.e. arithmetic averaging (AA), Voronoi diagram (VD), time

stability (TS), and apparent thermal inertia (ATI), show that the AA method with input of the maximum

number of available SM monitoring sites (AA-max) can be used to represent the actual areal SM conditions

($SM_{truth}$). The arithmetic average of the three and four monitoring sites with long-term continuous

measurements (AA-min) are found to be most suitable to produce the upscaled SM dataset for the period

2009-2019, which yields RMSEs of 0.022 and 0.011 $m^3$ $m^{-3}$ for the Maqu and Shiquanhe networks in

comparison to the $SM_{truth}$.

Trend analysis of the approximately 10-year upscaled SM time series produced by the AA-min ($SM_{AA-min}$)

shows that the Shiquanhe network in the western part of the TP is getting wet while no significant trend is

found for the Maqu network in the east. The usage of all the available monitoring sites each year leads to

inconsistent time series of SM that cannot capture the trend of $SM_{AA-min}$ reliably. Comparisons between the

$SM_{AA-min}$ and the model-based SM products from the ERA5-land, GLDAS Noah, and MERRA2 further

demonstrate that current model-based SM products still show deficiencies in representing the trend and the

dynamics of the SM measured on the TP. Moreover, strategies for maintaining the Tibet-Obs are provided,

and it is suggested to maintain currently 12 operational sites for the Shiquanhe network, while for the Maqu

network it is suggested to restore five old sites.

The 10-year (2009-2019) surface SM dataset presented in this paper includes the 15-min *in situ* measurements

taken at a depth of 5 cm collected from three regional-scale networks (i.e. Maqu, Naqu, and Ngari including

Ali and Shiquanhe) of the Tibet-Obs, and the spatially upscaled SM datasets produced by the AA-min for

the Maqu and Shiquanhe networks. This dataset is valuable for calibrating/validating long-term satellite- and

model-based SM products, evaluation of SM upscaling methods, development of data fusion methods, and

quantifying the coupling of SM with precipitation at 10-year scale.

## Author contribution

Pei Zhang, Donghai Zheng, Rogier van der Velde and Zhongbo Su designed the framework of this work. Pei

Zhang performed the computations and data analysis, and written the manuscript. Donghai Zheng, Rogier

van der Velde and Zhongbo Su supervised the progress of this work and provided critical suggestions, and

revised the manuscript. Zhongbo Su and Jun Wen designed the setup of Tibet-Obs, Yijian Zeng, XinWang

and Zuoliang Wang involved in maintaining the Tibet-Obs and downloading the original measurements. Pei

Zhang, Zuoliang Wang, and Jiali Chen organized the data.

## Competing interests

The authors declare that they have no conflict of interest.

## Acknowledgments

This study was supported by the Strategic Priority Research Program of Chinese Academy of Sciences (Grant

No. XDA20100103) and National Natural Science Foundation of China (Grant No. 41971308, 41871273).

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

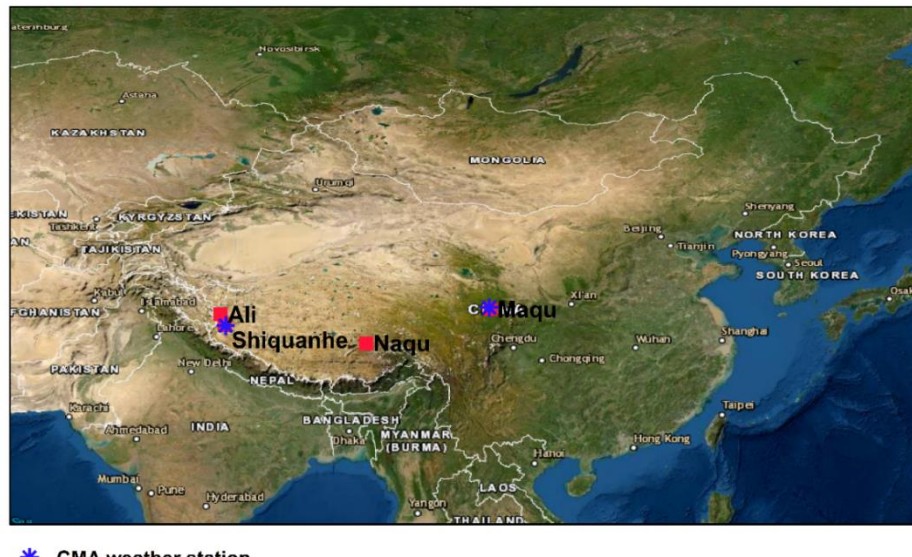

**Fig. 1. Locations of the Tibet-Obs including Maqu, Naqu, and Ngari (including Ali and Shiquanhe) soil moisture**
**monitoring networks. The weather stations of Maqu and Shiquanhe operated by the China Meteorological**
**Administration (CMA) are also shown. (Base map is from Esri, Copyright: © Esri)**

(a) Maqu

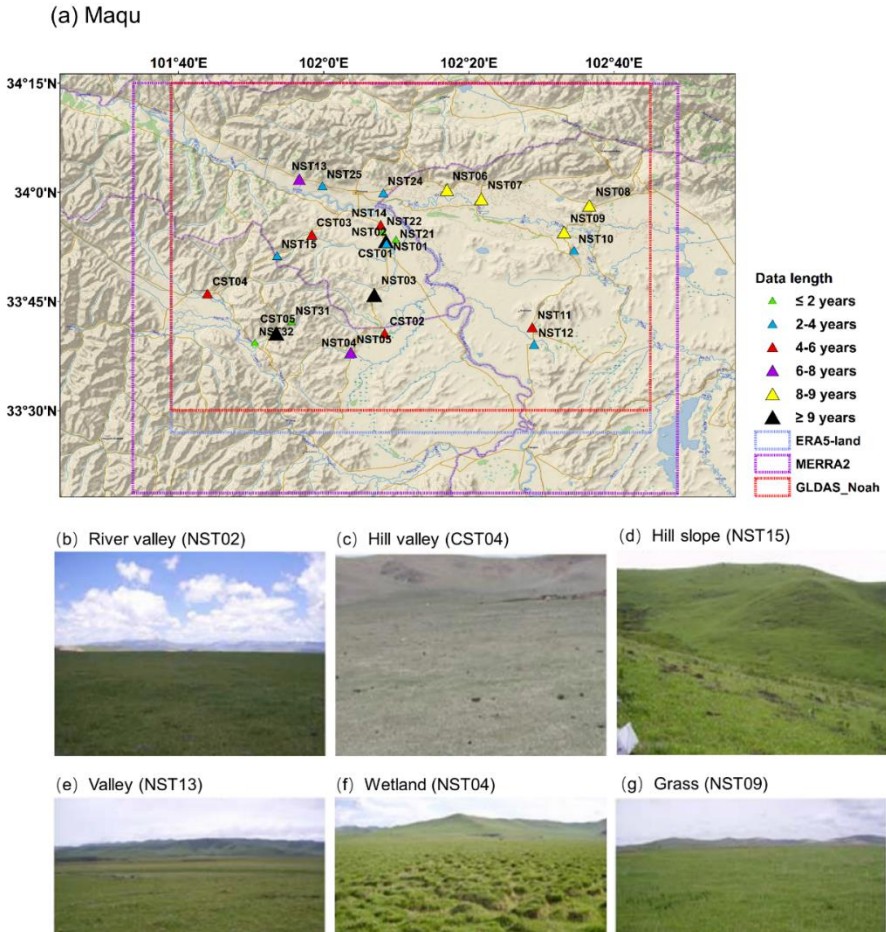

(b) River valley (NST02)    (c) Hill valley (CST04)    (d) Hill slope (NST15)

(e) Valley (NST13)    (f) Wetland (NST04)    (g) Grass (NST09)

**Fig. 2. (a) Overview of the Maqu monitoring network, and typical characteristics of topography and land cover**
**within the network: (b) river valley, (c) hill valley, (d) hill slope, (e) valley, (f) wetland and (g) grass. The colored**
**triangles in (a) represent different data lengths of surface SM measurements for each site, and the colored boxes**
**represent the coverage of selected model-based products. The site name in the bracket in (b)-(g) indicates the site**
**location for which the photograph is selected. (Base map copyright: ©2018 Garmin)**

(a) Maqu    (b) Ngari

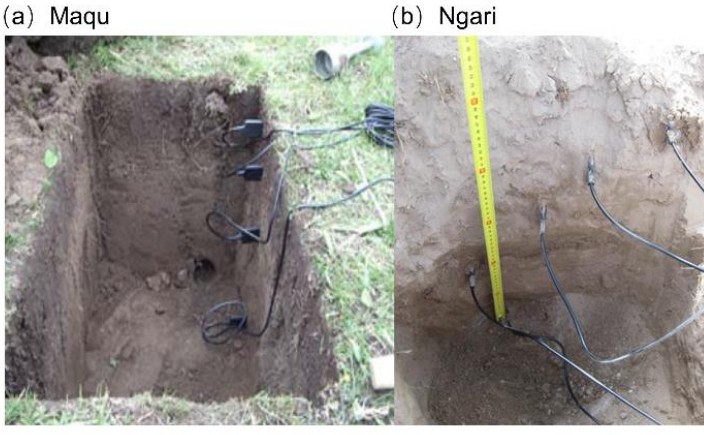

**Fig. 3. Examples of typical installation of sensors in monitoring sites of (a) Maqu and (b) Ngari networks.**

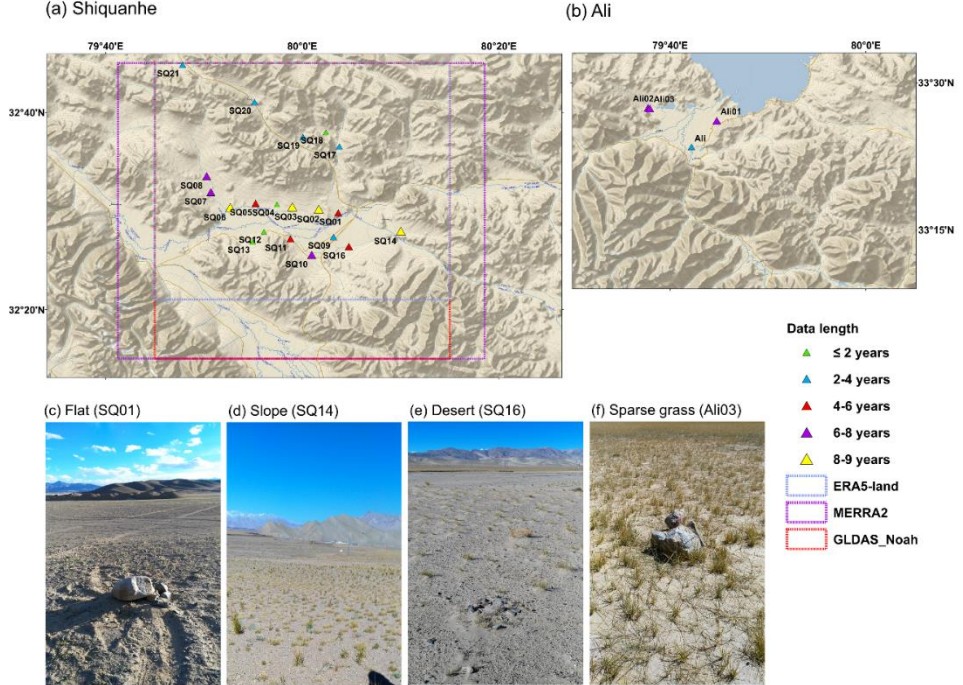

**Fig. 4. Overview of the Ngari monitoring network including (a) Shiquanhe and (b) Ali networks, and typical characteristics of topography and land cover within the network: (c) flat, (d) slope, (e) desert, and (f) sparse grass. The colored triangles in (a) and (b) represent different data lengths of surface SM measurements for each site, and the colored boxes represent the coverage of selected model-based products. The site name in the bracket in (c)-(f) indicates the site location for which the photograph is selected. (Base map copyright: ©2018 Garmin)**

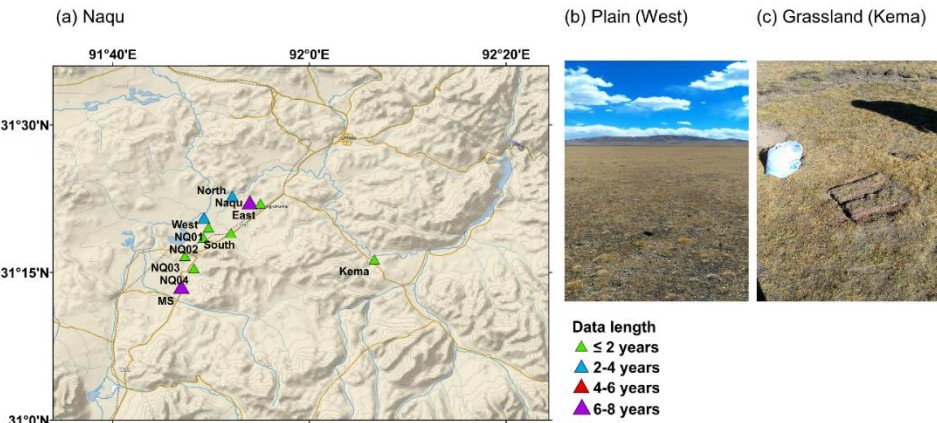

**Fig. 5. (a) Overview of the Naqu monitoring network, and typical characteristics of topography and land cover within the network: (b) plain and (c) grassland. The colored triangles in (a) represent different data lengths of surface SM measurements for each site. The site name in the bracket in (b) and (c) indicates the site location for which the photograph is selected. (Base map copyright: ©2018 Garmin)**

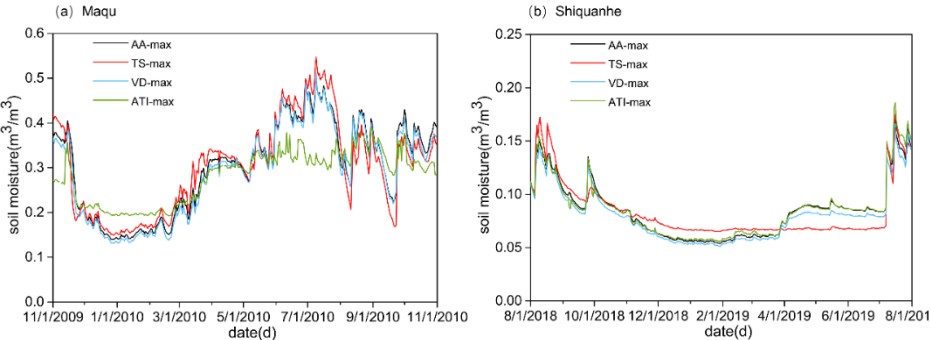

**Fig. 6. Comparisons of daily average SM for the (a) Maqu and (b) Shiquanhe networks produced by four upscaling methods with input of the maximum number of available SM monitoring sites.**

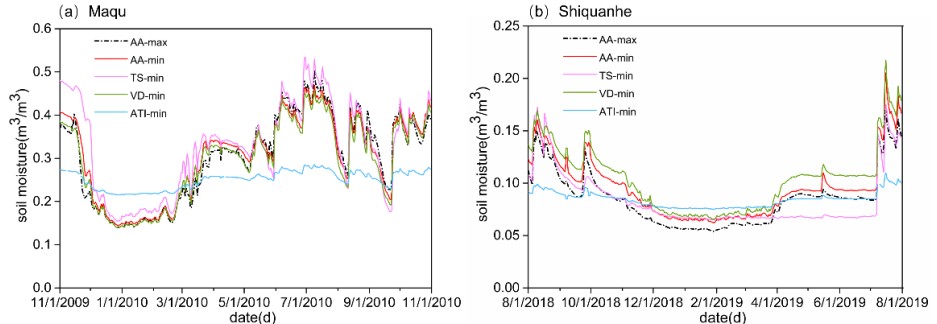

**Fig. 7. Comparisons of daily average SM for the (a) Maqu and (b) Shiquanhe networks produced by four upscaling methods with input of the minimum number of available SM monitoring sites.**

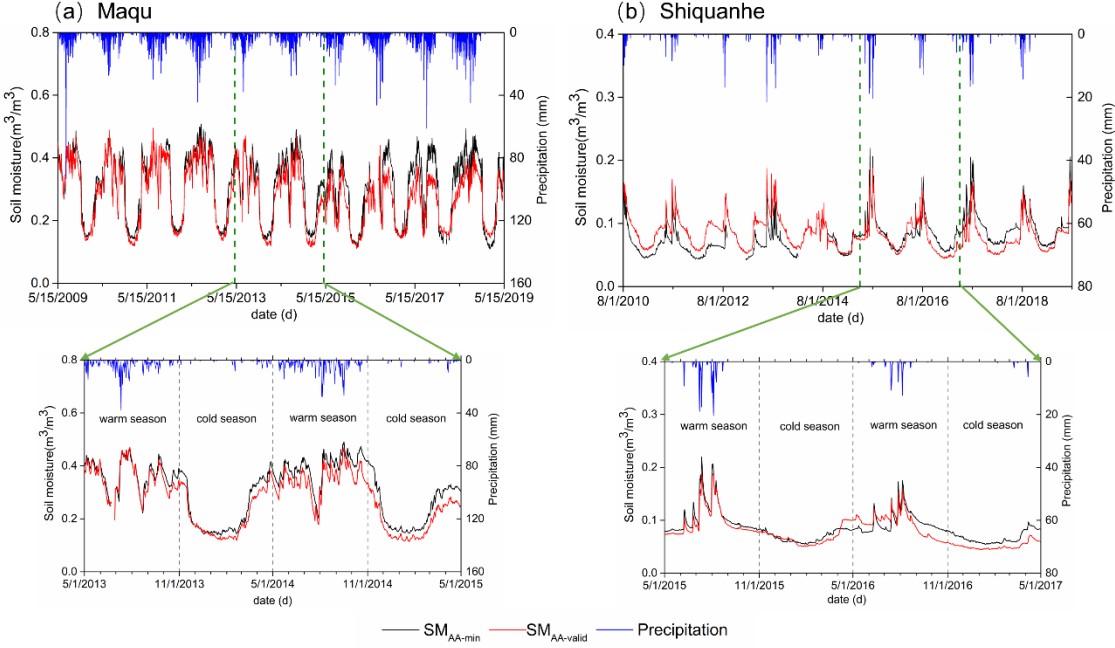

**Fig. 8. Time series of SM$_{AA-min}$, SM$_{AA-valid}$, and precipitation for the (a) Maqu and (b) Shiquanhe networks for a 10-year period, the subplot highlights a 2-year period.**

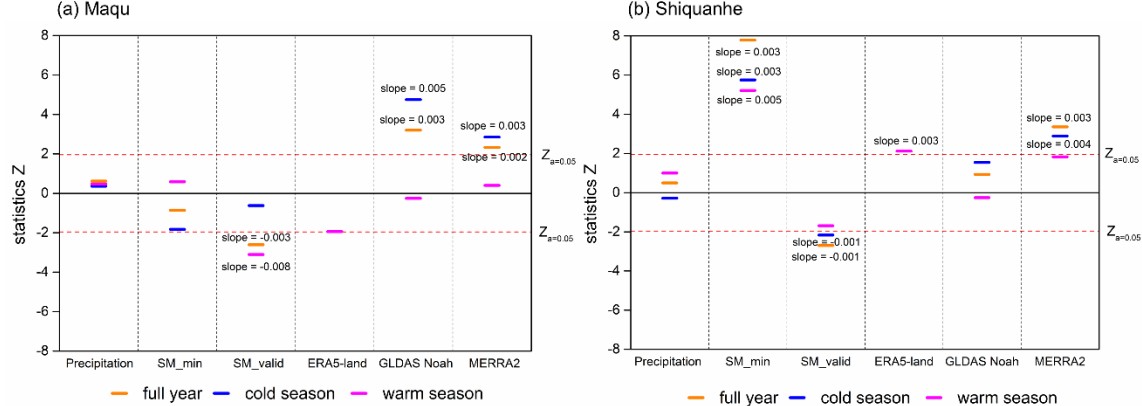

**Fig. 9. Mann Kendall trend test and Sen's slope estimate for precipitation, SM$_{AA-min}$, SM$_{AA-valid}$, and model-based**
**SM derived from the ERA5-land, GLDAS Noah, and MERRA2 for a 10-year period for the (a) Maqu and (b)**
**Shiquanhe networks.**

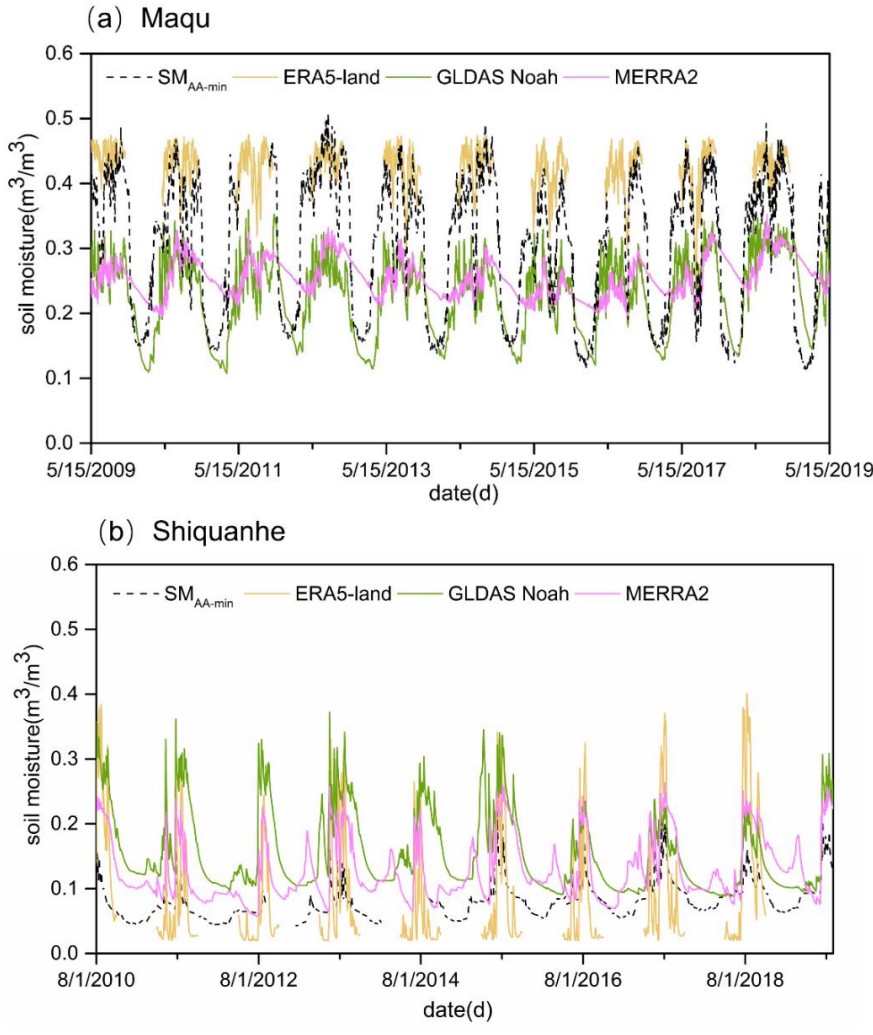

**Fig. 10. A 10-year time series of model-based SM derived from the ERA5-land, MERRA2, and GLDAS Noah**
**products and the upscaled SM (SM$_{AA-min}$) for the (a) Maqu and (b) Shiquanhe networks.**

**Table 1. Summary of the main Tibet-Obs applications and corresponding findings.**

| Literature | *In situ* data | Satellite- and/or model-based products | Key findings |
|---|---|---|---|
| Dente et al. (2012a) | Maqu network, period between 2008 and 2009 | LPRM AMSR-E SM product, ASCAT SM product | i) The weighted average of SM depended on the percentage spatial coverage strata can be regarded as the ground reference.<br>ii) The AMSR-E and ASCAT products are able to provide reasonable area SM during monsoon seasons. |
| Dente et al. (2012b) | Maqu network, period of 2010 | Soil Moisture and Ocean Salinity (SMOS) Level 2 SM product | The SMOS product exhibits a systematic dry bias (0.13 $m^3$ $m^{-3}$) at the Maqu network. |
| Zeng et al. (2015) | Maqu network, period between 2008 and 2010 | SMOS Level 3 SM product (version 2.45), Advanced Microwave Scanning Radiometer for Earth Observation System SM products (AMSR-E) SM products developed by National Aeronautics and Space Administration (NASA version 6), Land Parameter Retrieval Model (LPRM version 2), and Japan Aerospace Exploration Agency (JAXA version 700), AMSR2 Level 3 SM product (version 1.11), Advanced Scatterometer SM product (ASCAT version TU-Wien-WARP 5.5), ERA-Interim SM product (version 2.0), and Essential Climate Variable SM product (ECV version 02.0) | i) The ECV and ERA products give the best performance, and all products are able to capture the SM dynamic except for the NASA product.<br>ii) The JAXA AMSR-E/AMSR2 products underestimate SM, while the ASCAT product overestimates it.<br>iii) The SMOS product exhibits big noise and bias, and the LPRM AMSR-E product shows a significantly larger seasonal amplitude. |
| Zheng et al. (2015a) | Maqu network, period between 2009 and 2010 | Noah LSM (land surface model) simulations | The modified hydraulic parameterization is able to resolve the SM underestimation in the upper soil layer under wet conditions, and it also leads to better capture for SM profile dynamics combined with the modified root distribution. |
| Bi & Ma (2015) | Maqu network, period between 2008 and 2011 | GLDAS SM products produced by Noah, Mosaic CLM and Variable Infiltration Capacity (VIC) models | The SM simulated by the four LSMs can give reasonable SM dynamics but still show negative biases probably resulted from the high soil organic carbon content. |
| Li et al. (2018) | Maqu network, period between 2015 and 2016 | Soil Moisture Active Passive (SMAP) Level 3 standard (36km) and enhanced (9km) passive SM products (version 3), Community Land Model (CLM4.5) simulations | i) The standard and enhanced SMAP products have similar performance for SM spatial distributions.<br>ii) The SM of enhanced SMAP product exhibits good agreement with the CLM4.5 SM simulation. |
| Zhao et al. (2017) | Maqu network, period between 2008 and 2010 | Downscaled SM from five typical triangle-based empirical SM relationship models | The model treating the surface SM as a second-order polynomial with LST, vegetation indices, and surface albedo outperforms other models. |
| Ju et al. (2019) | Maqu network, period of 2012 | VIC LSM simulations | The IEPFM (immune evolution particle filter with Markov chain Monte Carlo simulation) is able to mitigate particle impoverishment and provide better assimilation results. |
| Zheng et al. (2018b) | Ngari network, period between 2015 and 2016 | SMAP Level 2 radiometer SM product | Modifying surface roughness and employing soil temperature and texture information can improve the SMAP SM retrievals for the desert ecosystem of the TP. |
| Zhang et al. (2018) | Maqu and Ngari networks, period between 2010 and 2013 | ERA-Interim SM product, MERRA SM product, GLDAS_Noah SM product (version2.0 and version2.1) | All these products exhibit overestimation at the Ngari network while underestimation at the Maqu network except for the ERA-Interim product. |

| Zheng et al. (2018a) | Maqu and Ngari networks, period between 2015 and 2016 | SMAP Level 1C radiometer brightness temperature products (version 3) | i) The SMAP algorithm underestimates the significance of surface roughness while overestimates the impact of vegetation. ii) The modified brightness temperature simulation can result in better SM retrievals. |
|---|---|---|---|
| Wei et al. (2019) | Maqu and Ngari networks, period between 2015 and 2016 | SMAP Level 3 SM passive product | The downscaled SM still can keep accuracy compared to the SM of original SMAP product. |
| Liu et al. (2019) | Maqu and Ngari networks, period between 2012 and 2016 | SMAP Level 3 SM products (version 4.00), SMOS-IC SM products (version 105), Fengyun-3B Microwave Radiation Image SM product (FY3B MWRI), JAXA AMSR2 Level 3 SM product, LPRM AMSR2 Level 3 SM product (version 3.00) | i) The JAXA AMSR2 product underestimates area SM while the LPRM AMSR2 product overestimates it. ii) The SMOS-IC product exhibits some noise of SM temporal variation. iii) The SMAP product has the highest accuracy among the five products while FY3B shows relatively lower accuracy. |
| Yang et al. (2020) | Maqu and Ngari network, period between 2008 and 2011 | AMSR-E brightness temperature product | The assimilated SM products exhibit higher accuracy than the AMSR-E product and LSM simulations for wet areas, whereas their accuracy is similar for dry areas. |
| Su et al. (2013) | Maqu and Naqu networks, period between 2008 and 2009. | AMSR-E SM product, ASCAT Level 2 SM product, ECMWF SM analyses i.e. optimum interpolation and extended Kalman filter products | i) The Naqu area SM is overestimated by the ECMWF products in monsoon seasons, while the Maqu area SM produced by the ECMWF is comparable to previous studies. ii) The SM estimate cannot be considerably improved by assimilating ASCAT data due to the CDF matching approach and the data quality. |
| Zeng et al. (2016) | Maqu, Naqu and Ngari networks, period between 2010 and 2011 | LPRM AMSR-E SM product, ERA-Interim SM product | The blended SM is able to capture temporal variations across different climatic zones over the TP. |
| Cheng et al. (2019) | Maqu, Naqu and Ngari networks, period of 2010 | European Space Agency Climate Change Initiative Soil Moisture SM product (ESA CCISM version 4.4), ERA5 SM product | i) The seasonal variation and spatial distribution of SM can be captured by all four products i.e., ESA CCI_active, ESA CCI_passive, ESA CCI_combined, and ERA5. ii) The ESA CCI_active and ESA CCI_combined products exhibit narrower magnitude than the ESA CCI passive and ERA5 products. iii) The SM uptrend across the TP can be found from the ERA5 product. |

**Table 2. Data records of all the SMST monitoring sites performed for the Maqu network. Blank cells represent that there are no measurements performed. Cells with hyphen represent that data is available. The number in cells represents the month(s) when the data is missing during a year.**

| | 2009 | 2010 | 2011 | 2012 | 2013 | 2014 | 2015 | 2016 | 2017 | 2018 | 2019 | Data length (months) |
|---|---|---|---|---|---|---|---|---|---|---|---|---|
| CST01 | — | — | 10~12 | 1~6 10~12 | | | | | | | | 36 |
| CST02 | — | — | 5~12 | 1~10 | 6 | 7~12 | | | | | | 46 |
| CST03 | — | — | — | — | 6~12 | 1~10 | 7~12 | | | 1~9 | 5~12 | 68 |
| CST04 | 1~5 | — | 12 | 1~3 11~12 | 1~2 6 | 8~10 | 7~12 | | 1~6 | 7~12 | | 73 |
| CST05 | — | — | — | — | 6 | — | — | 5~7 | — | 1~2 | 6~12 | 119 |
| NST01 | 1~5 | — | — | — | 6 | — | — | 5~7 | — | — | 6~12 | 116 |
| NST02 | 1~3 | — | — | 7~8 10~12 | | | | | | | | 40 |
| NST03 | — | — | 5~10 | — | 6 | — | — | 5~7 | — | — | 6~12 | 115 |
| NST04 | — | — | 10~12 | | | | | | | | | 33 |
| NST05 | 3~5 | — | — | — | 6~12 | 1~7 | — | 5~7 | 7~12 | 1~7 | 6~12 | 92 |
| NST06 | — | 1~3 12 | 1~3 | — | 6 | — | — | 6~7 | 8~12 | 1~7 | 6~12 | 104 |
| NST07 | — | — | 3 | — | 6, 12 | 1 | 12 | 1~2 7,12 | 1~2 12 | 1~3 9~12 | | 101 |
| NST08 | — | 2, 4 9~12 | 1~5 | — | 6~10 | 1~10 | — | 6~7 | — | — | 6~12 | 95 |
| NST09 | 1, 12 | 1~4 12 | 1~3 | — | 1~2 6 | 7~10 | 12 | 1~3 7, 12 | 1~2 7 | — | 6~12 | 99 |
| NST10 | — | 11~12 | 1~5 7~12 | 1~6 | 6~12 | | | | | 1~7 | 6~12 | 44 |
| NST11 | — | — | — | 7~8 | 6 | 7~12 | | | | | | 63 |
| NST12 | 10~12 | 1~9 | — | — | 6~12 | 1~10 | 7~12 | | | | | 49 |
| NST13 | — | — | — | — | 6 | — | 7~12 | | | | | 77 |
| NST14 | 6~9 | — | — | — | 6 | 10~12 | | | | | | 64 |
| NST15 | — | 10~12 | 1~5 | 6~12 | | | | | | | | 33 |
| NST21 | | | | | | 1~7 | 7~12 | | | | | 11 |
| NST22 | | | | | | 1~7 | 7~12 | | | | | 11 |
| NST24 | | | | | | 1~7 | 2~12 | 1~7 | — | — | 6~12 | 40 |
| NST25 | | | | | | 1~7 | — | 2~12 | 1~8 | — | 6~12 | 39 |
| NST31 | | | | | | | | | 1~8 | 7~12 | | 10 |
| NST32 | | | | | | | | | | 1~5 | 6~12 | 12 |

**Table 3. Same as the Table 2 but for the Ngari network.**

| | 2010 | 2011 | 2012 | 2013 | 2014 | 2015 | 2016 | 2017 | 2018 | 2019 | Data length (months) |
|---|---|---|---|---|---|---|---|---|---|---|---|
| | | | | | | Shiquanhe network | | | | | |
| SQ01 | 1~7 | — | — | — | 9~12 | 1~9 | | | | | 52 |
| SQ02 | 1~7 | — | — | — | 5~9 | — | — | — | — | 9~12 | 104 |
| SQ03 | 1~7 | — | — | — | 8~9 | — | — | — | — | 9~12 | 107 |
| SQ04 | 1~7 | — | 9~12 | | | | | | | | 25 |
| SQ05 | 1~7 | — | — | — | 5~12 | | | | | | 45 |
| SQ06 | 1~7 | — | 9~12 | 1 | 2~9 | — | — | — | — | 9~12 | 96 |
| SQ07 | 1~7 | — | — | 9~12 | 1~8 | — | 7~8 | 7~8 | — | 9~12 | 93 |
| SQ08 | 1~7 | 8~12 | | 1~8 | 8~9 | — | — | — | — | 9~12 | 82 |
| SQ09 | 1~7 | — | 9~12 | 1~8 | 9~12 | | | | | | 37 |
| SQ10 | | 1~8 | — | — | 7~12 | 1~9 | 7~12 | 1~8 | — | 9~12 | 67 |
| SQ11 | 1~7 | — | — | 9~12 | | | | | 1~8 | 9~12 | 49 |
| SQ12 | 1~7 | — | 9~12 | | | | | | | | 25 |
| SQ13 | 1~7 | 8~12 | | | | | | | | | 12 |
| SQ14 | 1~7 | — | — | — | 6 8~9 | — | — | — | — | 9~12 | 106 |
| SQ16 | 1~7 | 7~8 | — | — | 3~8 | 9~12 | | | | | 53 |
| SQ17 | | | | | | | 1~8 | — | — | 9~12 | 36 |
| SQ18 | | | | | | | 1~8 | 1 | 9~12 | | 23 |
| SQ19 | | | | | | | 1~8 | — | — | 9~12 | 36 |
| SQ20 | | | | | | | 1~8 | — | — | 9~12 | 36 |
| SQ21 | | | | | | | 1~8 | — | — | 9~12 | 36 |
| | | | | | | Ali network | | | | | |
| Ail | 1~7 | — | 9~12 | 1~8 | | | | 1~8 | 8~12 | | 40 |
| Ali01 | 1~7 | 8~12 | 1~8 | — | 8 | — | — | — | 8~12 | | 82 |
| Ali02 | 1~7 11~12 | 1~8 | — | — | 8 | — | — | — | 8~12 | | 85 |
| Ali03 | 1~7 | — | — | 3~12 | 1~8 | — | — | — | 8~12 | | 78 |

**Table 4. Same as the Table 2 but for the Naqu network.**

| | 2010 | 2011 | 2012 | 2013 | 2014 | 2015 | 2016 | 2017 | 2018 | 2019 | Data length (months) |
|---|---|---|---|---|---|---|---|---|---|---|---|
| Naqu | 1~7 | — | — | 8~9 | 6~8 | 6~9 | — | 9~12 | 1~8 | 9~12 | 88 |

| | | | | | | | | | | | |
|---|---|---|---|---|---|---|---|---|---|---|---|
| **East** | | 1~8 | — | 9~12 | | | | | | | 24 |
| **West** | 1~7 | 1~8 | — | 1~9 | 7~12 | 1~7 | 8~12 | | | | 42 |
| **North** | | 1~8 11~12 | 1~3 9 | 9~12 | | | 1~8 | 9~12 | 1~8 | 9~12 | 42 |
| **South** | | 1~8 | 9~12 | | | | | | | | 12 |
| **Kema** | | | | 1~9 | 3~9 | — | 8~12 | | | | 26 |
| **MS** | 1~7 | — | 10~12 | 1~9 | 8~9 11~12 | 1~5 | — | 9~12 | 1~8 | 9~12 | 76 |
| **NQ01** | | | | | | | | | 1~8 | 9~12 | 12 |
| **NQ02** | | | | | | | | | 1~8 | 9~12 | 12 |
| **NQ03** | | | | | | | 1~8 | 9~12 | 1~8 | 9~12 | 24 |
| **NQ04** | | | | | | | | | 1~8 | 9~12 | 12 |

**Table 5. Error statistics computed between the $SM_{AA\text{-}min}$ and the three model-based SM products for the Maqu and Shiquanhe networks.**

| | Bias ($m^3\,m^{-3}$) | RMSE ($m^3\,m^{-3}$) | Bias ($m^3\,m^{-3}$) | RMSE ($m^3\,m^{-3}$) |
|---|---|---|---|---|
| | Warm season | | Cold season | |
| | Maqu | | | |
| ERA5-land | 0.050 | 0.067 | - | - |
| GLDAS Noah | -0.112 | 0.125 | -0.049 | 0.088 |
| MERRA2 | -0.113 | 0.124 | 0.006 | 0.097 |
| | Shiquanhe | | | |
| ERA5-land | 0.002 | 0.079 | - | - |
| GLDAS Noah | 0.010 | 0.116 | 0.052 | 0.058 |
| MERRA2 | 0.054 | 0.069 | 0.049 | 0.053 |

**Table 6. Percentages of the site combinations that fall into an accuracy requirement in terms of RMSE.**

| RMSE | 0.004 | 0.006 | 0.008 | 0.010 | 0.012 | 0.014 | 0.016 | 0.018 | 0.020 |
|---|---|---|---|---|---|---|---|---|---|
| | | | | Maqu network | | | | | |
| n=1 (%) | | | | | | | | | |
| n=2 (%) | | | | | | | | 0.74 | 3.68 |
| n=3 (%) | | | | | | 0.44 | 1.32 | 3.97 | 7.79 |
| n=4 (%) | | | | | 0.21 | 1.05 | 3.74 | 9.16 | 16.93 |
| n=5 (%) | | | | 0.03 | 0.58 | 3.10 | 9.31 | 18.23 | 28.18 |
| n=6 (%) | | | | 0.09 | 1.87 | 8.27 | 19.18 | 31.22 | 42.36 |
| n=7 (%) | | | | 0.69 | 6.21 | 18.11 | 31.91 | 43.98 | 54.32 |
| n=8 (%) | | | 0.08 | 3.29 | 14.97 | 30.32 | 43.97 | 55.36 | 64.79 |
| n=9 (%) | | | 0.84 | 9.58 | 26.27 | 42.42 | 55.47 | 65.94 | 74.16 |
| n=10 (%) | | 0.01 | 3.91 | 19.74 | 38.94 | 54.41 | 66.13 | 75.21 | 82.23 |

| | | | | | | | | | |
|---|---|---|---|---|---|---|---|---|---|
| n=11 (%) | | 0.53 | 11.10 | 32.92 | 51.7 | 65.66 | 75.9 | 83.32 | 88.87 |
| n=12 (%) | | 3.52 | 23.95 | 47.3 | 64.03 | 75.87 | 84.45 | 90.14 | 94.30 |
| n=13 (%) | 0.29 | 13.82 | 39.87 | 61.81 | 75.67 | 85.38 | 91.55 | 95.38 | 97.77 |
| n=14 (%) | 3.68 | 32.35 | 57.79 | 74.85 | 86.47 | 92.79 | 96.91 | 98.82 | 99.41 |
| n=15 (%) | 21.32 | 56.62 | 75.00 | 88.97 | 95.59 | 98.53 | 99.26 | 100.00 | 100.00 |
| n=16 (%) | 52.94 | 82.35 | 94.12 | 94.12 | 100.00 | 100.00 | 100.00 | 100.00 | 100.00 |
| Shiquanhe network | | | | | | | | | |
| n=1 (%) | | | | | | | 8.33 | 16.67 | 25.00 |
| n=2 (%) | | 1.52 | 1.52 | 4.55 | 13.64 | 30.30 | 37.88 | 42.42 | 48.48 |
| n=3 (%) | | 6.82 | 21.36 | 25.45 | 33.18 | 42.73 | 53.18 | 59.55 | 65.00 |
| n=4 (%) | 1.62 | 11.31 | 29.7 | 41.41 | 51.11 | 57.37 | 63.23 | 70.51 | 77.58 |
| n=5 (%) | 3.66 | 23.11 | 36.87 | 49.12 | 60.23 | 68.18 | 76.14 | 82.32 | 88.26 |
| n=6 (%) | 11.36 | 30.95 | 44.37 | 59.85 | 70.24 | 79.11 | 85.28 | 90.15 | 93.29 |
| n=7 (%) | 20.20 | 39.77 | 56.06 | 68.31 | 77.90 | 86.87 | 93.43 | 96.84 | 98.48 |
| n=8 (%) | 29.29 | 50.51 | 62.63 | 77.58 | 89.09 | 96.57 | 97.98 | 98.99 | 99.60 |
| n=9 (%) | 33.64 | 59.55 | 82.73 | 91.36 | 96.36 | 98.18 | 99.55 | 99.55 | 100.00 |
| n=10 (%) | 48.48 | 78.79 | 92.42 | 96.97 | 96.97 | 100.00 | 100.00 | 100.00 | 100.00 |
| n=11 (%) | 83.33 | 91.67 | 100.00 | 100.00 | 100.00 | 100.00 | 100.00 | 100.00 | 100.00 |

**Table 7. The combinations of monitoring sites ranked by RMSE values of average SM at the Maqu network.**

| Rank | Site1 | Site2 | Site3 | Site4 | Site5 | Site6 | Site7 | Site8 | Site9 | Site10 | Site11 | Site12 | **RMSE** |
|---|---|---|---|---|---|---|---|---|---|---|---|---|---|
| 1 | CST01 | CST02 | NST02 | NST03 | NST04 | NST05 | NST06 | NST07 | NST10 | NST13 | NST14 | NST15 | 0.00402 |
| 2 | CST01 | CST02 | CST04 | NST01 | NST02 | NST03 | NST04 | NST05 | NST06 | NST07 | NST13 | NST15 | 0.00417 |
| 3 | CST02 | NST01 | NST02 | NST03 | NST04 | NST05 | NST06 | NST07 | NST10 | NST13 | NST14 | NST15 | 0.00450 |
| 4 | CST01 | CST02 | NST01 | NST02 | NST03 | NST04 | NST05 | NST06 | NST07 | NST13 | NST14 | NST15 | 0.00450 |
| 5 | CST01 | CST02 | CST03 | NST02 | NST03 | NST04 | NST05 | NST06 | NST07 | NST10 | NST14 | NST15 | 0.00451 |
| 96 | CST01 | CST02 | CST03 | CST04 | CST05 | NST03 | NST06 | NST10 | NST11 | NST13 | NST14 | NST15 | 0.00555 |
| 97 | CST01 | CST02 | CST03 | NST01 | NST02 | NST04 | NST05 | NST06 | NST11 | NST13 | NST14 | NST15 | 0.00555 |
| 98 | CST01 | CST02 | CST03 | CST04 | CST05 | NST01 | NST02 | NST05 | NST06 | NST10 | NST11 | NST15 | 0.00556 |
| 99 | CST03 | NST02 | NST03 | NST04 | NST05 | NST06 | NST07 | NST10 | NST11 | NST13 | NST14 | NST15 | 0.00557 |
| **100** | **CST02** | **CST03** | **CST05** | **NST01** | **NST03** | **NST05** | **NST06** | **NST10** | **NST11** | **NST13** | **NST14** | **NST15** | **0.00557** |

**Appendix A. Basic information of the Tibet-Obs**

725 **Table A1. Site information of the Maqu network (site name, elevation, topography (TPG), land cover (LC), soil texture at 5-15 cm depth (STX), soil bulk density at 5cm depth (BD), soil organic matter content at 5-15cm depth (OMC), Not Available (NA), BD and OMC values are measured in the laboratory).**

| Site name | Elevation (m) | TPG | LC | STX | BD (kg m$^{-3}$) | OMC (g/kg) |
|---|---|---|---|---|---|---|
| CST01 | 3431 | River valley | Grass | NA | NA | NA |
| CST02 | 3449 | River valley | Grass | NA | NA | NA |
| CST03 | 3507 | Hill valley | Grass | NA | NA | NA |
| CST04 | 3504 | Hill valley | Grass | NA | NA | NA |
| CST05 | 3542 | Hill valley | Grass | NA | NA | NA |
| NST01 | 3431 | River valley | Grass | Silt loam | 0.96 | 18 |
| NST02 | 3434 | River valley | Grass | Silt loam | 0.81 | 18 |
| NST03 | 3513 | Hill slope | Grass | Silt loam | 0.63 | 49 |
| NST04 | 3448 | River valley | Wetland | Silt loam | 0.26 | 229 |
| NST05 | 3476 | Hill slope | Grass | Silt loam | 0.75 | 22 |
| NST06 | 3428 | River valley | Grass | Silt loam | 0.81 | 23 |
| NST07 | 3430 | River valley | Grass | Silt loam | 0.58 | 23 |
| NST08 | 3473 | Valley | Grass | Silt loam | 1.06 | 34 |
| NST09 | 3434 | River valley | Grass | Sandy loam | 0.91 | 17 |
| NST10 | 3512 | Hill slope | Grass | Loam-silt loam | 1.05 | 24 |
| NST11 | 3442 | River valley | Wetland | Organic soil | 0.24 | 136 |
| NST12 | 3441 | River valley | Grass | Silt loam | 1.02 | 39 |
| NST13 | 3519 | Valley | Grass | Silt loam | 0.67 | 29 |
| NST14 | 3432 | River valley | Grass | Silt loam | 0.68 | 30 |
| NST15 | 3752 | Hill slope | Grass | Silt loam | 0.78 | 56 |
| NST21 | 3428 | River valley | Grass | Silt loam | NA | NA |
| NST22 | 3440 | River valley | Grass | Silt loam | NA | NA |
| NST24 | 3446 | River valley | Grass | Silt loam | NA | NA |
| NST25 | 3600 | Hill slope | Grass | Silt loam | NA | NA |
| NST31 | 3490 | NA | NA | NA | NA | NA |
| NST32 | 3490 | Hill valley | Grass | NA | NA | NA |

**Table A2. Soil moisture with temporal persistence for the Maqu network. Cells with hyphen represent that no**
730 **data is missing, cells with "M" indicate data is missing with little influence.**

| Time | 2009.11~ 2010.11 | 2010.11.~ 2011.11 | 2011.11~ 2012.11 | 2012.11~ 2013.11 | 2013.11~ 2014.11 | 2014.11~ 2015.11 | 2015.11~ 2016.11 | 2016.11~ 2017.11 | 2017.11~ 2018.11 |
|---|---|---|---|---|---|---|---|---|---|
| CST05 | — | — | — | — | — | — | — | — | — |
| NST01 | — | — | — | — | — | — | M | — | — |
| NST03 | — | M | — | — | — | — | M | — | — |
| NST06 | — | M | — | — | — | — | — | | |
| NST07 | — | — | — | — | — | — | | | |
| NST13 | — | — | — | — | — | — | | | |
| NST01 | — | — | — | — | | | | | |
| NST14 | — | — | — | — | | | | | |
| CST03 | — | — | — | | | | | | |
| NST05 | — | — | — | | | | | | |
| CST01 | — | — | | | | | | | |
| CST04 | — | — | | | | | | | |
| NST02 | — | — | | | | | | | |
| NST04 | — | — | | | | | | | |
| CST02 | — | | | | | | | | |
| NST10 | — | | | | | | | | |
| NST15 | — | | | | | | | | |

**Table A3. Same as the Table A1 but for the Ngari network (BD and OMC data are not available).**

| Site name | Elevation (m) | TPG | LC | STX |
|---|---|---|---|---|
| | | | Shiquanhe network | |
| SQ01 | 4306 | Flat | Desert | Loamy sand |
| SQ02 | 4304 | Gentle slope | Desert | Sand |
| SQ03 | 4278 | Gentle slope | Desert (with sparse bushes) | Sand |
| SQ04 | 4269 | Edge of a wetland | Sparse grass | Loamy sand |
| SQ05 | 4261 | Edge of a marsh | Sparse grass | Sand |
| SQ06 | 4257 | Flat | Sparse grass | Loamy Sand |
| SQ07 | 4280 | Flat | Desert (with sparse bushes) | Sand |
| SQ08 | 4306 | Flat | Desert | Sand |
| SQ09 | 4275 | Flat | Desert/river bed | Sand |
| SQ10 | 4275 | Flat | Grassland | Fine sand with some thick roots |
| SQ11 | 4274 | Flat | Grassland with bushes | Loamy sand |
| SQ12 | 4264 | Flat | Edge of riverbed | Sandy loam |
| SQ13 | 4292 | Flat | Valley bottom | Sand |

| SQ14 | 4368 | Slope | Desert | Sandy loam |
|------|------|-------|--------|------------|
| SQ16 | 4288 | Flat | Desert/river bed | Loam |
| SQ17 | 4563 | NA | NA | NA |
| SQ18 | 4634 | NA | NA | NA |
| SQ19 | 4647 | NA | NA | NA |
| SQ20 | 4695 | NA | NA | NA |
| SQ21 | 4606 | NA | NA | NA |
| Ali network | | | | |
| Ali | 4288 | Flat | Grass | Loamy sand |
| Ali01 | 4262 | Flat | Sparse grass | Sand |
| Ali02 | 4266 | Flat | Sparse grass | Sand |
| Ali03 | 4261 | Edge of a wetland | Grass | Sand |

**Table A4. Same as Table A2 but for the Shiquanhe network.**

| Time | 2010.8~ 2011.8 | 2011.8~ 2012.8 | 2012.8~ 2013.8 | 2013.8~ 2014.8 | 2014.8~ 2015.8 | 2015.8~ 2016.8 | 2016.8~ 2017.8 | 2017.8~ 2018.8 | 2018.8~ 2019.8 |
|------|------|------|------|------|------|------|------|------|------|
| SQ02 | — | — | — | M | — | — | — | — | — |
| SQ03 | — | — | — | — | — | — | — | — | — |
| SQ06 | — | — | M | M | — | — | — | — | — |
| SQ14 | — | — | — | — | — | — | — | — | — |
| SQ08 | | | | — | — | — | — | — | — |
| SQ07 | | | | | — | — | — | — | — |
| SQ17 | | | | | | | — | — | — |
| SQ19 | | | | | | | — | — | — |
| SQ20 | | | | | | | — | — | — |
| SQ21 | | | | | | | — | — | — |
| SQ10 | | | | | | | | — | — |
| SQ11 | | | | | | | | | — |

**Table A5. Same as the Table A1 but for the Naqu network (BD and OMC data are not available).**

| Site name | Elevation (m) | TPG | LC | STX |
|-----------|---------------|-----|-----|-----|
| Naqu | 4509 | Plain | Grassland | Loamy sand |
| East | 4527 | Flat hill top | Grassland | Loamy sand |
| West | 4506 | Plain | Grassland | Loamy sand |
| North | 4507 | Slope on riverbank | Grassland | Loamy sand |
| South | 4510 | Slope of wetland | Wetland | Loamy sand |

| Kema | 4465 | River valley | Grass | Silt loam |
| MS | 4583 | NA | NA | NA |
| NQ01 | 4517 | NA | NA | NA |
| NQ02 | 4552 | NA | NA | NA |
| NQ03 | 4638 | NA | NA | NA |
| NQ04 | 4632 | NA | NA | NA |

## Appendix B. Spatial upscaling methods

### B.1 Arithmetic averaging

The arithmetic averaging method assigns an equal weight coefficient to each SM monitoring site of the network, which can be formulated as:

$$\overline{\theta}_t^{ups} = \frac{1}{N}\sum_{i=1}^{N}\theta_{t,i}^{obs} \tag{B1}$$

where $i$ represents the $i$th SM monitoring site.

### B.2 Voronoi diagram

The Voronoi diagram method divides the network area into several parts according to the distances between each SM monitoring site. This approach determines the weight of each site ($w_i$ [-]) based on the geographic distribution of all the SM monitoring sites within the network area, which can be formulated as:

$$\overline{\theta}_t^{ups} = \frac{\sum_{i=1}^{N}w_i\theta_{t,i}^{obs}}{\sum_{i=1}^{N}w_i} \tag{B2}$$

### B.3 Time stability

The time stability method is based on the assumption that the spatial SM pattern over time tends to be consistent (Vachaud et al., 1985), and the most stable site can be regarded as the representative site of the network. For each SM monitoring site $i$ within the time window ($M$ days in total), the mean relative difference $MRD_i$ [-] and standard deviation of the relative difference $\sigma(RD_i)$ [-] are estimated as:

$$\sigma(RD_i) = \sqrt{\frac{1}{M-1}\sum_{t=1}^{M}(RD_{t,i} - MRD_i)^2} \tag{B3}$$

$$MRD_i = \frac{1}{M}\sum_{t=1}^{M}\frac{\theta_{t,i}^{obs} - \overline{\theta_t^{obs}}}{\overline{\theta_t^{obs}}} \tag{B4}$$

$$RD_{t,i} = \frac{\theta_{t,i}^{obs} - \overline{\theta_t^{obs}}}{\overline{\theta_t^{obs}}} \tag{B5}$$

where $\theta_{t,i}^{obs}$ [m³ m⁻³] represents the SM measured on the $t$th day at the $i$th monitoring site, $\overline{\theta_t^{obs}}$ [m³ m⁻³] represents the mean SM measured at all available monitoring sites on the $t$th day. $MRD_i$ quantifies the bias of each SM monitoring site to identify a particular location is wetter or drier than regional mean, and $\sigma(RD_i)$

characterizes the precision of the SM measurement. Jacobs et al., (2004) combined above two statistical metrics as a comprehensive evaluation criterion ($CEC_i$ [-]):

$$CEC_i = \sqrt{(MRD_i)^2 + \sigma(RD_i)^2} \tag{B6}$$

The most stable site is identified by the lowest $CEC_i$ value.

**B.4 Apparent thermal inertia**

The apparent thermal inertia (ATI) method is based on the close relationship between apparent thermal inertia ($\tau$ [K$^{-1}$]) and SM ($\theta$ [m$^3$ m$^{-3}$]) (Van doninck et al., 2011; Veroustraete et al., 2012). If the true areal SM ($\bar{\theta}_t^{tru}$ [m$^3$ m$^{-3}$]) is available, then the weight vector $\beta$ can be derived by the ordinary least-squares (OLS) method that minimizes the cost function $J$ as:

$$J = \sum_{t=1}^{M}(\theta_t^{tru} - \beta^T \theta_t^{obs})^2 \tag{B7}$$

However, the $\theta_t^{tru}$ [m$^3$ m$^{-3}$] is usually not available in practice, and the representative SM ($\bar{\theta}_t^{rep}$ [m$^3$ m$^{-3}$]) is thus introduced that contains random noise but with no bias. Since the OLS method may results in overfitting with usage of the $\bar{\theta}_t^{rep}$, a regularization term is introduced and Eq. (B7) can be re-formulated as (Tarantola, 2005):

$$J = \sum_{t=1}^{M}(\bar{\theta}_t^{rep} - \beta^T \theta_t^{obs})\sigma^{-2}(\bar{\theta}_t^{rep} - \beta^T \theta_t^{obs}) + R\beta^T\beta \tag{B8}$$

where $\sigma$ [m$^3$ m$^{-3}$] represents the standard deviation of $\bar{\theta}_t^{rep}$, R [-] is the regularization parameter.

The core issue of the ATI approach is to obtain the $\bar{\theta}_t^{rep}$ and minimize the cost function of Eq. (B8) to obtain β and R. The $\bar{\theta}_t^{rep}$ can be retrieved from the apparent thermal inertia $\tau$ via empirical regression g($\tau$), and $\tau$ has strong connection with the surface status, e.g. land surface temperature and albedo, which is defined as:

$$\tau = C\frac{1-a}{A} \tag{B9}$$

where $C$ [-] represents the solar correction factor, $a$ [-] represents the surface albedo, and $A$ [K] represents the amplitude of the diurnal temperature cycle. The albedo and land surface temperature data obtained from the MODIS MCD43A3 and MYD11A1/MOD11A1 Version 6 products are used to derive the ATI according to Eq. (B9) in this study.

The solar correlation factor $C$ in Eq. (B9) is computed as:

$$C = \sin\varphi\sin\delta(1 - tan^2\varphi tan^2\delta)^{1/2} + cos\varphi cos\delta arccos(-tan\varphi tan\delta) \tag{B10}$$

with

$$\delta = 0.00691 - 0.399912\cos(\gamma) + 0.070257\sin(\gamma) - 0.006758\cos(2\gamma) + 0.000907\sin(2\gamma) - 0.002697\cos(3\gamma) + 0.00148\sin(3\gamma) \tag{B11}$$

and

$$\gamma = \frac{2\pi(n_d-1)}{365.25} \tag{B12}$$

where $\varphi$ represents the latitude [rad], $\delta$ represents the solar declination [rad], and $n_d$ represents the Julian day number.

The amplitude of the diurnal LST $A$ is estimated as $LST_{max}$ - $LST_{min}$ for a single day. Finally, we use the regression analysis between *in situ* SM measurements ($\theta$) at each monitoring site and corresponding ATI ($\tau$) to obtain the g(·) form.

There are 17 and 12 monitoring sites participate in the regression analysis for the Maqu and Shiquanhe networks during the periods of 11/2009-10/2010 and 8/2018-7/2019, respectively. The ATI cannot be obtained for each monitoring site in every day since the satellite-based LST data are contaminated by clouds. In order to make full use of the data, we make the ATI-SM pair for the 1st monitoring site on the 1st day as No. 1, the pair for the 17th (or 12th) monitoring site in the Maqu (or Shiquanhe) network on the 1st day as the

No. 17 (or No. 12), the pair for the 1st monitoring site at the 2nd day as the No. 18 (No. 13), and so on. Later on, we select a certain number of ATI-SM pairs (e.g. 40, 50, 60, 70, 80, 90, and 100) as a group to compute the averaged ATI and SM and construct the most reliable (i.e. with the maximum $R^2$) regression relationship between them. If the ATI or SM data at one day is missing, this pair is ignored. As shown in Fig. B1, the empirical relationship is generated from 80 pairs ATI and SM averaged for the Maqu and Shiquanhe

networks.

When the empirical relationship g(·) is determined, the regional-average SM can be derived from grid-averaged ATI by the function g(·), which it is regarded as $\bar{\theta}_t^{rep}$ in Eq. (B8). Finally, the optimal $\beta$ ($\hat{\beta}$) is obtained by minimizing the cost function (i.e. Eq. (B8)), and the upscaled SM can be estimated as:

$$\bar{\theta}_t^{ups} = \hat{\beta}\theta_t^{obs} \tag{B13}$$

The detailed description of the ATI method is referred to Qin et al. (2013).

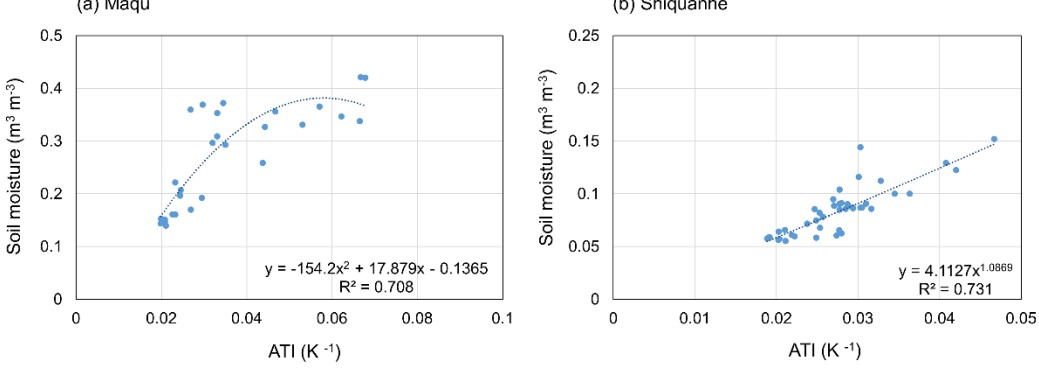

**Fig. B1 Empirical relationship between 80 pair of ATI and SM averaged for the (a) Maqu and (b) Shiquanhe networks.**

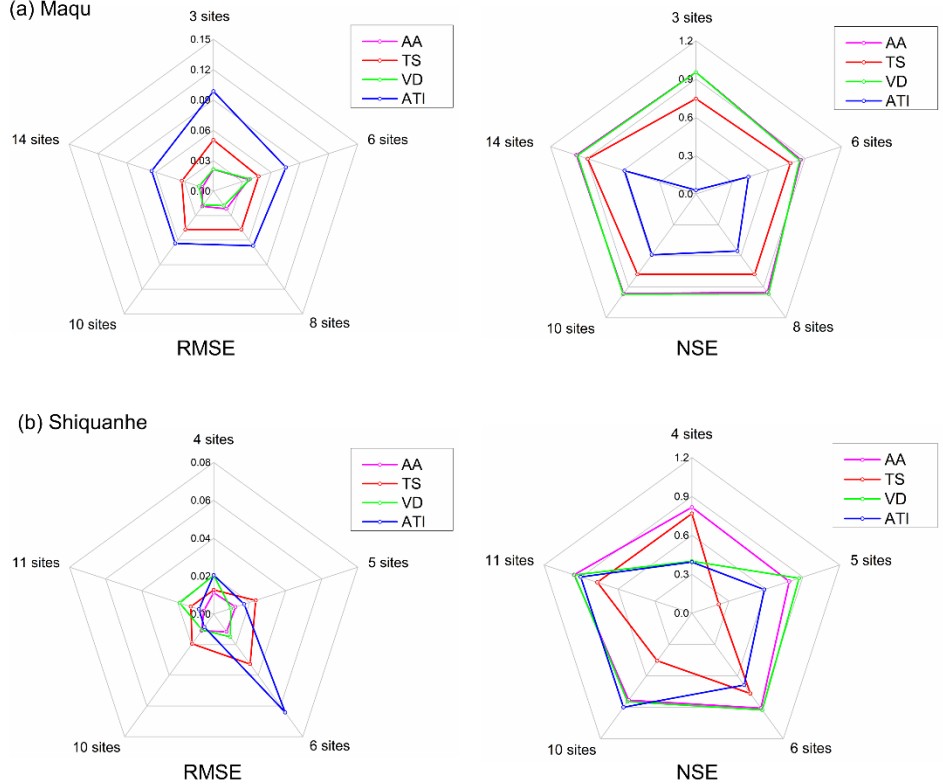

**Fig. B2. Radar diagram of error statistics (i.e. RMSE and NSE) computed between the SM_truth produced by the AA-max and the upscaled SM produced by the four upscaling methods with input of different number of available monitoring sites for the (a) Maqu and (b) Shiquanhe networks.**

**Table B1. Evaluation metrics computed for the upscaled SM produced with four methods with input of the maximum available monitoring sites.**

| Methods | Maqu | | | Shiquanhe | | |
|---------|------|------|-----|-----------|------|-----|
| | MRD | σ(RD) | CEC | MRD | σ(RD) | CEC |
| AA-max | 0.009 | 0.054 | **0.055** | 0.012 | 0.046 | **0.047** |
| TS-max | 0.022 | 0.089 | 0.092 | 0.011 | 0.114 | 0.114 |
| VD-max | -0.026 | 0.064 | 0.069 | -0.042 | 0.033 | 0.053 |
| ATI-max | -0.005 | 0.145 | 0.145 | 0.016 | 0.068 | 0.070 |

**Table B2. Error statistics computed between the SM obtained by the four upscaling methods with input of the minimum available monitoring sites, and the SM_truth produced by the AA-max for the Maqu and Shiquanhe networks.**

| | Bias ($m^3\,m^{-3}$) | RMSE($m^3\,m^{-3}$) | ubRMSE ($m^3\,m^{-3}$) | NSE |
|---------|------|------|------|-----|
| | | Maqu | | |
| AA-min | 0.005 | 0.022 | 0.021 | 0.954 |
| TS-min | 0.025 | 0.050 | 0.044 | 0.747 |
| VD-min | -0.007 | 0.022 | 0.020 | 0.954 |
| ATI-min | -0.052 | 0.099 | 0.084 | 0.030 |
| | | Shiquanhe | | |

| | | | | |
|---|---|---|---|---|
| AA-min | 0.010 | 0.011 | 0.005 | 0.816 |
| TS-min | -0.001 | 0.013 | 0.013 | 0.768 |
| VD-min | 0.019 | 0.020 | 0.006 | 0.400 |
| ATI-min | -0.001 | 0.021 | 0.021 | 0.393 |

825