# Peer review of "Status of the Tibetan Plateau observatory (Tibet-Obs) and a 2 10-year (2009-2019) surface soil moisture dataset"

_Earth System Science Data, 2020_

## Referee Comment (RC1) · Mirko Mälicke (Referee) · 6 Nov 2020

The presented manuscript *"Status of the Tibetan Plateau observatory (Tibet-Obs) and a 10-year (2009-2019) surface soil moisture dataset"* by Zhang et al. describes an interesting dataset of *in situ* soil moisture measurements and an upscaled data product.

Long-term soil moisture datasets are important to foster our understanding of above-ground and below-ground hydrological processes that take place on a multitude of temporal and spatial scales. The actual dataset is available in a data repository with DOI and presented in a clear structure with a lot of auxiliary tables and descriptions. But, from my point of view, the manuscript is also showing some moderate to major

issues that need to be resolved before publication. The methods need to shift their focus from upscaling methods to more complete data and study site descriptions. In addition, the referenced data source seems to be incomplete. Detailed comments are listed below, grouped into my main points, comments on the data source itself, minor comments, and a few technical points.

With these issues addressed, I am looking forward to see this really interesting work published.

With kind regards,

Mirko Mälicke

**main comments**

These are my main comments in the order of importance:

1. The authors reference different data sources in the manuscript: precipitation data from the China Meteorological Data Service Center (p.5 l.153) and three different model-based soil moisture products (p.5 l.163; p.5 l.171; p.5 l.175). In all three cases, the data is referenced by an URL, without a proper entry in the bibliography or footnotes. Important information, like the access date, is missing.
Beyond this, I have some very serious issues with referencing the landing page of a national authority as a data source. It is not obvious, where data can be downloaded. Further, the search for the given station *Maqu* did not return any result and I am missing crucial information like the dataset name or aggregation level, which can be specified on the website. The precipitation data is absolutely necessary to reuse the dataset and apply the methods as presented. Thus I

would strongly recommend adding a copy of the data that has actually be used to the data repository. Neither the data repository nor the manuscript gives any metadata about precipitation (except a rough location).

The same applies to the model-based soil moisture products. In all three cases, the given URL references the landing page of the respective product, not the data. The GLDAS website still had a maintenance notice from September 2020 (writing on 05.11.2020) and is down. This highlights the importance of adding all the needed data to the repository.

2. From my point of view, a proper study site description and measurement setup is missing and should be added to the manuscript. The dataset itself gives some very broad information like *hill top* or *desert*, but that does not really foster my understanding of the conditions at the sites. Maybe there are even images of the study sites available?

The following questions might be helpful to be considered:

How does the terrain (especially aspect and slope) look like at each site?

What are the meteorological conditions at each site?

What is the landuse?

How were the sensors calibrated?

Which steps have been undertaken to check the data quality?

Were there corrections applied to ensure data quality?

Was the measuring interval set to 15 minutes or was the raw data aggregated to 15-minute intervals?

Does the device integrate over the given period, or does it sample the soil moisture every 15 minutes? What are the soil types at the sites, or the geology?

What are the general hydrologic conditions on the sites? Are there rivers nearby?

Are the sites in a wet, semi-arid or even arid climate?

Is there possible influence groundwater on the SM measurements?

Is there something else measured at or nearby the sites? Are any helpful atmospheric data available that could be included?
Have there ever been any laboratory measurements of probes from the sites, like
soil density, tension, porosity, or water content measurements?
Is soil profile data available? Maybe even images?
Why was that data recorded in the first place?
What does the specific setup at one site look like?

The same applies to all other data products that are used.

From my understanding, the manuscript should, as a dataset description paper,
put more focus on a description of the study sites and the data itself, and less
focus on a comparison of upscaling methods.

3. The authors present the AA method (p.6 l.193-196) as the prime method to up-
scale soil moisture measurements. It is hard to believe for me personally, that
this approach is really used in recent studies to create high-quality upscaled soil
moisture products. A few references of recent studies would be most helpful for
me.
Further, the manuscript compares different upscaling methods to each other, but
not all upscaled data products are included in the dataset. From my understand-
ing, a method comparison can be published in a scientific method development
paper (and then be referenced here). In this dataset description paper, I find
it only useful as auxiliary or additional information to the actual dataset as long
as all the data (of the other methods) along with the necessary metadata is in-
cluded.
Finally, if I didn't misunderstand the method, I cannot see how the arithmetic
mean of about 20 single values per timestep is useful as a dataset of its own.
This processing step can easily be reproduced without any effort. It is presented
as the main and core data product of this publication. I would strongly recom-
mend to clarify this point and put the focus on the actual measured data. This

includes shifting the focus from upscaling methods to data and study site descriptions.

4. I would like to comment on the part p. l.267-270. The authors use the CEC, which is only described in the appendix, to chose the best upscaling method. The selected upscaling product is then used as ground truth to assess all upscaling methods. On the one hand, the CEC is itself the core of the TS upscaling method, which is problematic as the CEC as an objective function for selection is then not objective anymore. On the other hand, the AA upscaling results are selected as *ground truth*, which is problematic as all four upscaled data products, including the AA results, are assessed using the ground truth dataset (p.9 l.285-287). At the same time, only the AA upscaled data is included in the dataset, not the others. My personal expectation of *ground truth* included in a data publication, would be independent measurements, that can be used as an objective reference.
I would strongly recommend clarifying the process of upscaling the dataset and include all data.

5. Most of section 4.3 (p.10 l.343 - p.11 l.363) compares the different model-based soil moisture products with the upscaling result that is considered to be the ground truth. Unfortunately, it was not possible to download all products due to server maintenance and unclear sources (as described in point 1) to directly compare any of the products.
The model-based products come at different spatial and temporal scales than the measuring network. To me, it is not clear, which of both over- or underestimates the SM variability. Maybe a comparison of all upscaled products, or even better: SM measurements, to all model-based products and less a validation of one through the other, would be more insightful to better classify the presented data in terms of their expressiveness for SM variability.
In any case, the trend analysis (section 4.2) and its application to model-based products (section 4.3) need to be better connected to the actual measurements presented in the dataset. For me, it would be helpful to better point out, how this information is useful for understanding or using the data.

**The dataset**

These comments concern only the dataset itself given at
https://doi.org/10.4121/uuid:21220b23-ff36-4ca9-a08f-ccd53782e834

1. The most important part is already mentioned in the main comments, point 1. I would strongly encourage the authors to include all used data into the data repository.

2. I think the dataset should be enriched with some more metadata. E.g. which sensor was used? What is the unit? How precise are the measurements? How does the study site look like? How does the terrain look like?
   Please also refer to my main comment 2.

3. It would be most helpful to include the soil temperature and electric conductivity data recorded by the 5TE into the dataset. That would tremendously foster the re-usability of the dataset. In addition, is there a specific reason why the other depths are not included?

4. Please consider extending the dataset description on the source. E.g. include the upscaling method used, name the sensors used or give more details on the context and catchment the sites are operated in. The data source itself should contain all necessary information to use the data. The authors could reuse many parts of this paper here.

As a personal remark, I found Table 1 of this publication really useful. Together with an overview plot of all measurements, it could be added to the user information PDF as well.

5. The manuscript reports on four different upscaling methods, but only one upscaled result is included in the source. I would suggest to include the other upscaled products and model-based products, as well.

6. minor: Please consider using an open data format like ODS, CSV or HDF5 instead of XLSX. The relevant software needed to open XLSX is neither free nor open.

**minor comments**

These are my minor comments, in order of appearance:

1. p.2 l.73-74: *"apparent-thermal-inertia (ATI) method."*; It would be most helpful to reference the actual implementation of this method, as the application ranges from surface heterogeneity analysis on Mars to soil moisture monitoring.

2. p.3 l.76: *"different spatial upscaling methods has been scarcely assessed"*; Without being an expert on that field, I would like to name studies like Qui et al. (2013), Wang et al. (2014, 2015), Gao et al. (2017) or Moghaddam et al. (2014), which to my understanding, assess upscaling methods for *in situ* soil moisture measurements. I would suggest clarifying if and why the named approaches might not be suitable for an application in the Tibet-Obs.

3. p.3 l.84-86: Why exactly were the named methods (AA, VD, TS, ATI) chosen? Are there any specific reasons, that make them most suitable? Is there a reason to exclude geostatistical approaches for upscaling?

[Figure]

4. p.3 l l.85-86: *"In addition, the Mann-Kendall (M-K) analysis is adopted to analyse the trend of both regional-scale SM and precipitation time series."*; For me, this sentence falls a bit out of context and is neither connected to the previous nor the following sentences. Why a trend analysis? Why precipitation? What kind of precipitation data? Why a M-K test that only tests on monotone trends?

5. section 2.1: throughout the whole section 2.1, the authors introduce the different sites and networks. I was asking myself, what exactly is a *site*, what makes up a *network*? Is a site a single soil profile? Are networks just sites close to each other, or do they share common ground like a catchment, landuse or similar? I would suggest adding a short definition of these terms.

6. p.7 l.229: *"$\Theta_t^{tru}$ represents the reference SM that is considered as the ground truth"*; What is the reference SM here? To my understanding, ground truth is an independent measurement of the modeled variable, which has not been described as part of the networks so far. Please clarify.

7. p.8 l.263-l.270: This paragraph references the method MRD, CEC, and $\sigma(RD)$, which are not described in the method part. Please describe, define, and add them.

8. p.10 l.321-324: The authors describe that some sites where damaged and new ones were built in an area of distinctly different conditions. I would like the authors to elaborate on how this is expected to affect a **monotonic** trend analysis, like M-K, and what exactly the reader can thus learn from this analysis.
Taking p.10 .329-334 into account, to me it seems like the long-term trends in the data are very much dependent on the locations of the sites and not caused by any other driver, like changing meteorological conditions. I think the trend analysis is helpful for understanding the dataset, but more from a data quality perspective, than to describe the study site (like on p.10 l.338-341). I would consider a rework

of the whole section 4.2, to better highlight why the trend analysis was done and what the conclusions and implications for using the dataset are.

9. Discussion: From my point of view, major parts of the discussion need substantial revisions. The whole paragraph p.12 l.387 - 403 remains quite unclear to me. It is referring to *Nine levels of RMSE standard* (p.12 l.393), which are not further described in the methods. Please clarify what that is. Subsequently, different percentages of these levels are reported and discusses. From these numbers, it is concluded to shut a number of sites down. Why?
In the following paragraph p.12 l.404-418 this is applied to the networks but I honestly don't get the point of this paragraph. Why would you want to decrease the number of sites? What is a *"Best level of RMSE"* (l.412)? What does *"all the possible combinations"* (l.413-414) refer to and what is $C_{17}^{12} = 6188$? What is *"the 100$^{th}$ combination of monitoring sites"* (l.417-418) ?

10. Figure 1 (p.19): I would strongly recommend reworking this figure. The overview of all sites is of crucial importance to understand the dataset. Please consider including an overview map of Eastern Asia or China, to let the reader better locate the networks. Please consider using larger symbols and especially enlarge the CMA weather station symbols. It would also be helpful to include some larger cities into the maps if there are any. That will make the orientation easier. Maybe it's even a good idea to split this figure into multiple maps and add information like geology or landuse to the maps, if available.

11. Finally, I really liked Table 1. Here the authors collected studies that were conducted on the different study sites, the data used, and key findings. I would consider briefly describing some of the most interesting findings in the introduction.

**technical comments**

1. I would suggest to actually include the caption to Fig.6, instead of referencing another figure's caption.

2. p.2 l.66: *"Studies have also reported..."*; It is a bit unclear for me, which studies the authors are referring to. The two by Su (sentence before), or the ones named in the following paragraph (Colliander, Quin, Dente..)?

3. p.2 l. 74-75: *"several particular or alternative methods [...] were adopted in some investigations."*; I would suggest to either specify the actual methods and investigations or to drop the sentence.

4. p.3 l.79-83: The authors introduce their SM network here. I would suggest referencing Fig. 1 here, to point the reader to the overview map. This makes the whole section way clearer.

5. p.4 l.111-112; l.126-127; l.141: here, the authors specify the soil texture for the different SM networks. Please clarify why only the soil texture is specified and not other soil properties? How is that expected to influence the measurements and the quality of the dataset? Where does this information come from?

6. p.4 .143: Just out of curiosity: How can you install a 5TE at 2.5cm depth? Can you comment on data quality here?

7. p.5 l.146 *"only the Naqu and MS have collected"*; what is MS referring to here?

8. p.5. l.151 *"The daily precipitation from two weather stations (Fig. 1)"*; I think the symbols on the map are way too small. I can't really see the weather stations in Fig.1

9. p.5 l.158 What is the ECMWF?

[Figure]

10. p.5 l.161 What is H-TESSEL and CY45R1?

11. p.7 l.210: *"the average SM data"*; What is the average SM data? I am not sure what *average* is referring to here.

12. p.8 l.267: If Eq. (A3) gives the *bias* of the upscaled product, what does Eq. (3a) then calculate? Is it the same, or another bias? Please clarify.

13. p.9. l.311 - p.10 l.312: With an extent of 10 years on the x-axis, it is very hard for me to see the described differences in value. I can hardly see the data. Please consider adding another plot or subplot with a shorter time span.

**References**

Gao, Shengguo, et al. "Upscaling of sparse in situ soil moisture observations by integrating auxiliary information from remote sensing." International Journal of Remote Sensing 38.17 (2017): 4782-4803.

Moghaddam, Mahta, et al. "The SoilSCAPE network multiscale in-situ soil moisture measurements: Innovations in network design and approaches to upscaling in support of SMAP." AGUFM 2014 (2014): IN11A-3599.

Qin, Jun, et al. "Spatial upscaling of in-situ soil moisture measurements based on MODIS-derived apparent thermal inertia." Remote Sensing of Environment 138 (2013): 1-9.

Wang, Jianghao, et al. "Upscaling in situ soil moisture observations to pixel averages with spatio-temporal geostatistics." Remote Sensing 7.9 (2015): 11372-11388.

Wang, Jianghao, et al. "A geostatistical approach to upscale soil moisture with unequal precision observations." IEEE Geoscience and Remote Sensing Letters 11.12 (2014): 2125-2129.

---

## Referee Comment (RC2) · Anonymous Referee #2 · 6 Nov 2020

**OVERVIEW**

The study describes the status of the Tibetan Plateau observatory and propose the development of a 10 years surface soil moisture dataset obtained from in situ observations.

**GENERAL COMMENTS**

The paper is fairly well written and clear. The described dataset is very important both for the relevance of the study area and the importance of obtaining long-term soil moisture time series to be used as benchmark for modelling and satellite products. However, I believe some points need to be addressed before the publication. I have

listed below my general comments with the indication of their relevance.

1) MAJOR: The description of the dataset needs some improvements. I am aware that the Tibetan Plateau observatory has been already described in several previous papers (Table 1 is great), but in the online dataset more details should be added (soil, land use, climatology, pictures, . . .).

2) MAJOR: I have found the paper a compromise between a data paper and a scientific paper describing and testing upscaling procedures. I believe that in the dataset all the upscaled soil moisture time series should be available. Indeed, I have found weak the reasons for selecting the arithmetic average as reference method (ground truth). I guess that its better performance is very related to the employed metric (CEC). Of course, a reference does not exist, and hence I would suggest to publish and to make the analysis (e.g., comparison with modelled data), for all upscaled time series.

3) MODERATE: The ATI upscaling method is not clear to me. Where did you get the data for computing ATI? Why is its performance so different from the other upscaling techniques? Why is the range of soil moisture with ATI approach much smaller than other approaches? Please, provide more details.

4) MODERATE: The results of trend analysis are not clear to me. What do the authors want to highlight with this analysis? Why showing both UF and UB metrics? Also, the interpretation of results is unclear. By looking at figures, the trend of SM and precipitation is not consistent for Maqu site (line 319). For the same site, positive and negative values of UF are present, therefore I would not conclude for a drying trend (line 338). All trends are not significant (see line 320) as UF and UB values are lower than critical values. All these points should be addressed.

5) MAJOR: I believe that the SM AA-valid time series is wrong and should not been used. Of course, by averaging different sensors varying over time is not a correct procedure. I suggest removing this time series.

[Figure]

6) MODERATE: In the results and discussion sections several small errors (typo) are present that should be corrected. A quick reread of these sections will fix these errors.

**RECOMMENDATION**

Based on the above comments, I suggest a major revision before the possible publication on Hydrology and Earth System Sciences.

---

## Author Response (AR1)

**Summary of Major Changes**

Dear Editor and Reviewers,

Thank you very much for carefully reviewing our manuscript and providing detailed and constructive comments. This has helped us to improve our manuscript significantly. The manuscript has been thoroughly revised and strengthened based on all the comments and suggestions made by both reviewers.

Major changes made to the manuscript are summarized as follows:

1. Additional information and detailed descriptions of the Tibet-Obs have been provided in Section 2.1 and Appendix A. In addition, several figures and tables related to the comparison between four upscaling methods have been moved to Appendix B to make the manuscript more focused on the description of generated 10-year dataset itself.

2. The descriptions of Mann Kendall trend test and Sen's slope estimate in Section 3.2 have been rewrote, and the 10-year trend analysis in Sections 4.2 and 4.3 have also been thoroughly revised.

3. Section 5 has been thoroughly revised to make it more readable.

4. Detailed descriptions of the ATI (Apparent Thermal Inertia) upscaling method have been provided in the Appendix B.4.

Supplementary data and document have been added to the dataset in the 4TU.ResearchData repository as summarized below:

1. The precipitation data (2009-2019) from two weather stations, i.e. Maqu (34°00'N, 102°05'E) and Shiquanhe (32°30'N, 80°50'E), has been included.

2. The upscaled soil moisture data produced by the four upscaling methods for a single year have been included.

3. The soil moisture data from three model-based products (i.e. ERA5-land, GLDAS Noah, and MERRA2) at the Maqu and Shiquanhe network areas for a 10-year period have been included.

4. Document of user information has been revised to include descriptions of monitoring sites and summary of previous applications using the Tibet-Obs data.

Please find also below our detailed response to each comment made by the reviewers.

We think that the revised manuscript has appropriately addressed all the reviewers' concerns and we hope that you can consider it for publication in Earth System Science Data.

Sincerely,
Pei Zhang
On behalf of all co-authors

We would like to thank the reviewer for carefully reading our manuscript and providing detailed and constructive comments. This has helped us to improve our manuscript significantly. In the text below we provide our response to each comment point by point.

Reviewer's comments are in **bold**.

Author's responses are in regular.

Author's additions/modifications in the text are in blue.

**Main comments**

**1.      The authors reference different data sources in the manuscript: precipitation data from the China Meteorological Data Service Center (p.5 l.153) and three different model-based soil moisture products (p.5 l.163; p.5 l.171; p.5 l.175). In all three cases, the data is referenced by an URL, without a proper entry in the bibliography or footnotes. Important information, like the access date, is missing.**

**Beyond this, I have some very serious issues with referencing the landing page of a national authority as a data source. It is not obvious, where data can be downloaded. Further, the search for the given station Maqu did not return any result and I am missing crucial information like the dataset name or aggregation level, which can be specified on the website. The precipitation data is absolutely necessary to reuse the dataset and apply the methods as presented. Thus would strongly recommend adding a copy of the data that has actually be used to the data repository. Neither the data repository nor the manuscript gives any metadata about precipitation (except a rough location). The same applies to the model-based soil moisture products. In all three cases, the given URL references the landing page of the respective product, not the data. The GLDAS website still had a maintenance notice from September 2020 (writing on 05.11.2020) and is down. This highlights the importance of adding all the needed data to the repository.**

Thanks for the comments and suggestions. The detailed download links of the precipitation data and model-based products have been updated in the revised manuscript.

The link of the precipitation data is provided on Page 5 Line 163:

"https://data.cma.cn/dataService/cdcindex/datacode/SURF_CLI_CHN_MUL_DAY.html"

The download link of the ERA5-land product is provided on Page 5 Line 172:

"https://cds.climate.copernicus.eu/cdsapp#!/dataset/reanalysis-era5-land?tab=form"

The download link of the MERRA2 product is provided on Page 6 Line 181:

"https://search.earthdata.nasa.gov/search?q=M2T1NXINT"

The download link of the GLDAS Noah product is provided on Page 6 Line 190:

"https://search.earthdata.nasa.gov/search?q=GLDAS_NOAH025_3H_2.0"

The precipitation data, the upscaled soil moisture (SM) data produced by the four upscaling methods for a single year, and the 10-year SM data from three model-based products (i.e. ERA5-land, GLDAS Noah, and MERRA2) at the Maqu and Shiquanhe network areas have been added to the dataset in the 4TU.ResearchData repository, as shown in the following table.

| Folder | File/subfolder | Sheet/script |
|---|---|---|
| \*In situ* soil moisture\ | \Maqu.xlsx | - Information: This sheet contains the information of all the monitoring sites including location, elevation, topography, land cover, soil texture and soil organic matter content.
- 2009-2019: These 11 sheets contain the available original 15-min/30-min *in situ* measurements at 5 cm depth for each year. |
|  | \Shiquanhe.xlsx | - Information: This sheet contains the information of all the monitoring sites including location, elevation, topography, land cover, soil texture and soil organic matter content.
- 2010-2019: These 10 sheets contain the available original 15-min *in situ* measurements at 5 cm depth for each year. |
|  | \Ali.xlsx | - Information: This sheet contains the information of monitoring sites including location, elevation, topography, land cover, soil texture and soil organic matter content.
- 2010-2018: These 9 sheets contain the available original 15-min *in situ* measurements at 5 cm depth for each year. |
|  | \Naqu.xlsx | - Information: This sheet contains the information of monitoring sites including location, elevation, topography, land cover, soil texture and soil organic matter content.
- 2010-2019: These 10 sheets contain the available original 15-min *in situ* measurements at 5/2.5 cm depth for each year. |
| \Upscaled soil moisture\ | \Maqu upscaled.xlsx | - 2009-2019: These 11 sheets contain spatial upscaled soil moisture with the input of original *in situ* measurement between 5/15/2009 and 5/15/2019. |
|  | \ Maqu upscaled-daily average.xlsx | - 2009-2019: These 11 sheets contain spatial upscaled soil moisture with the input of daily average measurement between 5/15/2009 and 5/15/2019. |
|  | \Shiquanhe upscaled.xlsx | - 2010-2019: These 10 sheets contain spatial upscaled soil moisture with the input of original *in situ* measurement between 8/1/2010 and 8/1/2019 |
|  | \Shiquanhe upscaled-daily average.xlsx | - 2010-2019: These 10 sheets contain spatial upscaled soil moisture with the input of daily average measurement between 8/1/2010 and 8/1/2019 |
| \ Supplementary data \ | \ Upscaled -daily average.xlsx | - Maqu: This sheet contains spatial upscaled soil moisture with the input of daily mean from 17 sites (3 sites) between 11/1/2009 and 10/31/2010
- Shiquanhe: This sheet contains spatial upscaled soil moisture with the input of daily mean from 12 sites (4 sites) between 8/1/2018 and 7/31/2019 |
|  | \Model-based products.xlsx | - Maqu: This sheet contains daily regional mean soil moisture of Maqu network area (Fig. 2) from the ERA5-land, GLDAS Noah, and MERRA2 products between 5/15/2009 and 5/15/2019
- Shiquanhe: This sheet contains daily regional mean soil moisture of Shiquanhe network area (Fig. 3) from the ERA5-land, GLDAS Noah, and MERRA2 products between 8/1/2010 and 8/1/2019 |
|  | \Precipitation.xlsx | - Maqu: This sheet contains daily precipitation of Maqu weather station (Fig. 1) between 5/15/2009 and 5/15/2019
- Shiquanhe: This sheet contains daily precipitation of Shiquanhe weather station (Fig. 1) between 8/1/2010 and 8/1/2019 |

**2.      From my point of view, a proper study site description and measurement setup is missing and should be added to the manuscript. The dataset itself gives some very broad information like hill top or desert, but that does not really foster my understanding of the conditions at the sites. Maybe there are even images of the study sites available?**

**The following questions might be helpful to be considered:**

**How does the terrain (especially aspect and slope) look like at each site?**

**What are the meteorological conditions at each site?**

**What is the land use?**

**How were the sensors calibrated?**

**Which steps have been undertaken to check the data quality?**

**Were there corrections applied to ensure data quality?**

**Was the measuring interval set to 15 minutes or was the raw data aggregated to 15-minute intervals?**

**Does the device integrate over the given period, or does it sample the soil moisture every 15 minutes? What are the soil types at the sites, or the geology?**

**What are the general hydrologic conditions on the sites? Are there rivers nearby?**

**Are the sites in a wet, semi-arid or even arid climate?**

**Is there possible influence groundwater on the SM measurements?**

**Is there something else measured at or nearby the sites? Are any helpful atmospheric data available that could be included?**

**Have there ever been any laboratory measurements of probes from the sites, like soil density, tension, porosity, or water content measurements?**

**Is soil profile data available? Maybe even images?**

**Why was that data recorded in the first place?**

**What does the specific setup at one site look like?**

**The same applies to all other data products that are used. From my understanding, the manuscript should, as a dataset description paper, put more focus on a description of the study sites and the data itself, and less focus on a comparison of upscaling methods.**

Thanks for the comments and suggestions. All available information of the monitoring sites is provided in the revised manuscript in Section 2.1 and Appendix A, including the elevation, topography, land cover, soil texture, soil bulk density, and soil organic matter content that are listed in Tables A1, A3, and A5 for the Maqu, Ngari, and Naqu networks, respectively. Photographs selected from some monitoring sites are shown in Figs. 2, 4, and 5 for the Maqu, Ngari, and Naqu networks, respectively. Examples of profile measurement setup are shown in Fig. 3.

[revised manuscript text omitted]

**3.     The authors present the AA method (p.6 l.193-196) as the prime method to upscale soil moisture measurements. It is hard to believe for me personally, that this approach is really used in recent studies to create high-quality upscaled soil moisture products. A few references of recent studies would be most helpful for me. Further, the manuscript compares different upscaling methods to each other, but not all upscaled data products are included in the dataset. From my understanding, a method comparison can be published in a scientific method development paper (and then be referenced here). In this dataset description paper, I find it only useful as auxiliary or additional information to the actual dataset as long as all the data (of the other methods) along with the necessary metadata is included. Finally, if I didn't misunderstand the method, I cannot see how the arithmetic mean of about 20 single values per time step is useful as a dataset of its own. This processing step can easily be reproduced without any effort. It is presented as the main and core data product of this publication. I would strongly recommend to clarify this point and put the focus on the actual measured data.**

Thanks for the comments and suggestions. The comparison of four upscaling methods in a single year is a key step for selecting the appropriate upscaling method for producing the 10-year dataset and quantifying the uncertainty of this dataset since the number of available SM monitoring sites in the Tibet-Obs generally changes with time (see Tables 2-4 and Tables A2 and A4 for the details). Based on the comparison (see Section 4.1 for the details), we found that the arithmetic averaging method (AA) is more suitable for the Tibet-Obs, and the AA method with input of the minimum number of available SM monitoring sites (i.e. AA$_{min}$, 3 and 4 sites for the Maqu and Shiquanhe networks) is more suitable for producing the 10-year dataset with RMSEs of about 0.022 and 0.011 m$^3$ m$^{-3}$ for the Maqu and Shiquanhe networks (see Figs. 7 and B2 and Table B2). As such, the AA$_{min}$ is further adopted to produce the 10-year upscaled SM dataset that is the main and core data product of this publication. Therefore, we think it is better to keep the comparison of four upscaling methods in this manuscript instead of presenting them in the other paper.

Relevant figures and table above mentioned are shown below:

[Figure]

**Fig. 7. Comparisons of daily average SM for the (a) Maqu and (b) Shiquanhe networks produced by the four upscaling methods with input of the minimum number of available SM monitoring sites.**

[Figure]

**Fig. B2. Radar graph of error statistics (i.e. RMSE and NSE) computed between the SM$_{truth}$ produced by the AA-max and the upscaled SM produced by the four upscaling methods with input of different number of available monitoring sites for the (a) Maqu and (b) Shiquanhe networks.**

**Table B2. Error statistics computed between the SM obtained by the four upscaling methods with input of the minimum available monitoring sites and the SM$_{truth}$ produced by the AA-max for the Maqu and Shiquanhe networks.**

|  | Bias (m$^3$ m$^{-3}$) | RMSE(m$^3$ m$^{-3}$) | ubRMSE(m$^3$ m$^{-3}$) | NSE |
|---|---|---|---|---|
|  |  | Maqu |  |  |
| AA-min | 0.005 | 0.022 | 0.021 | 0.954 |
| TS-min | 0.025 | 0.050 | 0.044 | 0.747 |
| VD-min | -0.007 | 0.022 | 0.020 | 0.954 |
| ATI-min | -0.052 | 0.099 | 0.084 | 0.030 |
|  |  | Shiquanhe |  |  |
| AA-min | 0.010 | 0.011 | 0.005 | 0.816 |
| TS-min | -0.001 | 0.013 | 0.013 | 0.768 |
| VD-min | 0.019 | 0.020 | 0.006 | 0.400 |
| ATI-min | -0.001 | 0.021 | 0.021 | 0.393 |

According to the reviewer's suggestion, the reference about the arithmetic average in other studies are provided on Page 9 Line 300-301:

"The arithmetic average of the dense *in situ* measurements was also used as the ground truth in other studies (Qin et al., 2013; Su et al., 2013)."

And the upscaled SM data produced by the four upscaling methods for a single year has been added to the dataset in the 4TU.ResearchData repository. In addition, several figures and tables were moved to Appendix B to make the manuscript more focused on the description of generated 10-year dataset.

**4.      I would like to comment on the part p. l.267-270. The authors use the CEC, which is only described in the appendix, to chose the best upscaling method. The selected upscaling product is then used as ground truth to assess all upscaling methods. On the one hand, the CEC is itself the core of the TS upscaling method, which is problematic as the CEC as an objective function for selection is then not objective anymore. On the other hand, the AA upscaling results are selected as ground truth, which is problematic as all four upscaled data products, including the AA results, are assessed using the ground truth dataset (p.9 l.285-287). At the same time, only the AA upscaled data is included in the dataset, not the others. My personal expectation of ground truth included in a data publication, would be independent measurements, that can be used as an objective reference. I would strongly recommend clarifying the process of upscaling the dataset and include all data.**

Thanks for the comments and suggestions. In this manuscript, four upscaling methods with the input of the maximum number of available SM monitoring sites in a single year (17 and 12 sites for the Maqu and Shiquanhe networks) is firstly compared, and the comprehensive evaluation criterion (*CEC*) value is computed for the four upscaled SM time series to determine which one can best represent the areal conditions (i.e. ground truth, $SM_{truth}$). Since the number of available SM monitoring sites in the Tibet-Obs generally decreases with time (see Tables 2-4 and Tables A2 and A4 for the details), we further compare the performance of four upscaling methods with input of reducing number of available SM monitoring sites with the $SM_{truth}$ as reference to select the most suitable method for producing the 10-year dataset. It is found that the AA method with input of the minimum number of available SM monitoring sites (i.e. $AA_{min}$, 3 and 4 sites for the Maqu and Shiquanhe networks) is more suitable for producing the 10-year dataset (see Figs. 7 and B2 and Table B2).

Relevant information about above mentioned process is provided on Page 9 Line 278-283:

"In this section, four upscaling methods (see Section 3.1) are inter-compared first with the input of the maximum number of available SM monitoring sites for a single year in the Maqu and Shiquanhe networks to find the most suitable upscaled SM that can best represent the areal conditions (i.e. ground truth, $SM_{truth}$). Later on, the performance of the four upscaling methods is further investigated with the input of reducing number of SM monitoring sites to find the most suitable method for producing long-term (~10 year) upscaled SM for the Maqu and Shiquanhe networks."

It should be also noted that the *CEC* value is computed for the four upscaled SM dataset to determine the $SM_{truth}$, while the *CEC* is adopted by the TS method with input of available SM monitoring sites to produce the upscaled SM dataset. In addition, the upscaled SM data produced by the four methods in a single year has been added to the dataset in the 4TU.ResearchData repository.

**5.      Most of section 4.3 (p.10 l.343 - p.11 l.363) compares the different model-based soil moisture products with the upscaling result that is considered to be the ground truth. Unfortunately, it was not possible to download all products due to server maintenance and unclear sources (as described in point 1) to directly compare any of the products. The model-based products come at different spatial and temporal scales than the measuring network. To me, it is not clear, which of both over- or underestimates the SM variability. Maybe a comparison of all upscaled products, or even better: SM measurements, to all model-based products and less a validation of one through the other, would be more insightful to better classify the presented data in terms of their expressiveness for SM variability. In any case, the trend analysis (section 4.2) and its application to model-based products (section 4.3) need to be better connected to the actual measurements presented in the dataset. For me, it would be helpful to better point out, how this information is useful for understanding or using the data.**

Thanks for the comments and suggestions. Since the number of available SM monitoring sites in the Tibet-Obs generally changes with time (see Tables 2-4 and Tables A2 and A4 for the details), which will lead to inconsistent trend as the $SM_{AA-min}$ shown in Fig. 9 in Section 4.2. To address this issue, we first compare four upscaling methods in a single year to select the appropriate upscaling method for producing the 10-year dataset and quantifying the uncertainty of this dataset. It's found that the AA method with input of the minimum number of available SM monitoring sites (i.e. $AA_{min}$, 3 and 4 sites for the Maqu and Shiquanhe networks) is more suitable for producing the 10-year dataset with RMSEs of about 0.022 and 0.011 $m^3$ $m^{-3}$ for the Maqu and Shiquanhe networks (see Figs. 7 and B2 and Table B2).

To demonstrate the uniqueness of the generated dataset with $AA_{min}$ for validating existing reanalysis datasets for a long term period (~10 years), we further compare the generated dataset with three model-based products. Relevant information is provided on Page 11 Line 371-374:

"In this section, the long-term upscaled SM time series (i.e. $SM_{AA-min}$) produced for the two networks are applied to validate the reliability of three model-based SM products, i.e. ERA5-land, MERRA2, and GLDAS Noah, to demonstrate the uniqueness of this dataset for validating existing reanalysis datasets for a long term period (~10 years)."

The processing of the model-based products is provided on Page 8-9 Line 267-275:

"The performance of the ERA5-land, MERRA2, and GLDAS Noah SM products are assessed using the upscaled SM data of the Maqu and Shiquanhe networks for a 10-year period. The corresponding regional-scale SM for each product can be obtained by averaging the grid data falling in the network areas. The numbers of grids covering the Maqu and Shiquanhe networks are 77 and 20 for the ERA5-land product, 12 and 4 for the GLDAS Noah product, and only one for the MERRA2 product. Moreover, the ERA5-land and MERRA2 products with the temporal resolution of hourly and 3-hourly are averaged to daily-scale, and the unit of GLDAS Noah SM is converted from kg $m^{-2}$ to $m^3$ $m^{-3}$. The uppermost layer of the ERA5-land (0-7 cm), MERRA2 (0-5 cm), and GLDAS Noah SM products (0–10 cm) are considered to match the in situ observations at depth of 5 cm."

In addition, the 10-year SM data from three model-based products (i.e. ERA5-land, GLDAS Noah, and MERRA2) at the Maqu and Shiquanhe network areas have been added to the dataset in the 4TU.ResearchData repository.

**Minor comments**

These are my minor comments, in order of appearance:

1.      p.2 l.73-74: "apparent-thermal-inertia (ATI) method."; It would be most helpful to reference the actual implementation of this method, as the application ranges from surface heterogeneity analysis on Mars to soil moisture monitoring.

Thanks for the suggestion. The reference is provided on Page 2 Line 72-74:

"Qin et al. (2013, 2015) derived the weights by minimizing a cost function between *in situ* SM of individual monitoring site and a representative SM of the network that is estimated using the apparent-thermal-inertia-based (ATI) method (Gao et al., 2017)."

2.      p.3 l.76: "different spatial upscaling methods has been scarcely assessed"; Without being an expert on that field, I would like to name studies like Qui et al. (2013), Wang et al. (2014, 2015), Gao et al. (2017) or Moghaddam et al. (2014), which to my understanding, assess upscaling methods for in situ soil moisture measurements. I would suggest clarifying if and why the named approaches might not be suitable for an application in the Tibet-Obs.

Thanks for the suggestion. The references mentioned above have been cited in the revised manuscript and the sentence is revised on Page 2-3 Line 74-80:

"Alternative methods, such as time stability and ridge regression, have been adopted in other investigations (i.e. Zhao et al., 2013, Kang et al., 2017). While a large number of studies have assessed the performance of different upscaling methods in other areas such as the Tonzi Ranch network in California and the Heihe watershed (Moghaddam et al., 2014, Wang et al., 2014), only few investigations have been done for the TP (Gao et al., 2017, Qin et al., 2015). Since the number of monitoring sites changes over time due to damage of SM sensors in the Tibet-Obs, it is essential to evaluate and select an appropriate upscaling method for a limited number of sites."

3.      p.3 l.84-86: Why exactly were the named methods (AA, VD, TS, ATI) chosen? Are there any specific reasons, that make them most suitable? Is there a reason to exclude geo-statistical approaches for upscaling?

Thanks for the comments and suggestions. The four upscaling methods selected in this study are also widely used in previous other studies with reasonable results. Relevant information is provided on Page 2 Line 68-74:

"For instance, Colliander et al. (2017) employed Voronoi diagrams for the worldwide validation of the Soil Moisture Active/Passive (SMAP) SM products to determine the weights of individual monitoring sites within core regional-scale networks based on the geographic location; Dente et al. (2012a) determined the weights based on the topography and soil texture for the Maqu SM monitoring network of the Tibet-Obs; Qin et al. (2013, 2015) derived the weights by minimizing a cost function between *in situ* SM of individual monitoring site and a representative SM of the network that is estimated using the apparent-thermal-inertia-based (ATI) method (Gao et al., 2017)."

We also tested the geo-statistical method such as Block Kriging (BK). It is observed from Fig.R1 that there is not suitable theoretical model (e.g. bounded linear, exponent, spherical, and Gaussian) can be fitted with the experimental variogram distribution. As such, the geo-statistical method may be not suitable for the Tibet-Obs due

to the low autocorrelation of SM resulted from the sparse monitoring sites and complex topography. Therefore, we did not include the geo-statistical methods in this study.

[Figure]

**Fig. R1. Experimental variogram distribution using Surfer for the Maqu network.**

**4.        p.3 l.85-86: "In addition, the Mann-Kendall (M-K) analysis is adopted to analyse the trend of both regional-scale SM and precipitation time series."; For me, this sentence falls a bit out of context and is neither connected to the previous nor the following sentences. Why a trend analysis? Why precipitation? What kind of precipitation data? Why a M-K test that only tests on monotone trends?**

Thanks for the suggestions. This sentence is revised on Page 3 Line 87-88:

"Moreover, the variation and trend of the regional-scale SM time series are analyzed, and this 10-year SM dataset is used to validate the performance of three model-based SM products."

**5.        section 2.1: throughout the whole section 2.1, the authors introduce the different sites and networks. I was asking myself, what exactly is a site, what makes up a network? Is a site a single soil profile? Are networks just sites close to each other, or do they share common ground like a catchment, landuse or similar? I would suggest adding a short definition of these terms.**

Thanks for the suggestions. A short definition is provided on Page 3 Line 102-105:

"The Tibet-Obs consists of the Maqu, Naqu, and Ngari (including Shiquanhe and Ali) regional-scale SMST monitoring networks (Fig. 1) that cover the cold humid climate, cold semiarid climate, and cold arid climate, respectively. Each network includes different number of monitoring sites that measure the SMST at different soil depths."

In addition, additional information such as topography and land cover are provided for every monitoring site in each network in Table A1 (Maqu), A3 (Ngari), A5 (Naqu).

**6.        p.7 l.229: "_trut represents the reference SM that is considered as the ground truth"; What is the reference SM here? To my understanding, ground truth is an independent measurement of the modeled variable, which has not been described as part of the networks so far. Please clarify.**

Thanks for the comments. In this manuscript, four upscaling methods with input of the maximum number of available SM monitoring sites in a single year (17 and 12 sites for the Maqu and Shiquanhe networks) is firstly

compared, and the *CEC* value is computed for the four upscaled SM time series to determine which one can best represent the areal conditions (i.e. ground truth, SM$_{truth}$). It is found that the SM$_{AA-max}$ yields the lowest *CEC* values for both networks, indicating that the SM$_{AA-max}$ can be used to represent actual areal conditions, which will thus be regarded as the ground truth.

**7.      p.8 l.263-l.270: This paragraph references the method MRD, CEC, and _(RD), which are not described in the method part. Please describe, define, and add them.**

Thanks for the suggestion. These descriptions of the error metrics are added to Section 3.3 on Page 8 Line 259-266:

"The metrics used to define the most representative SM time series (i.e. the best upsclaed SM) is the comprehensive evaluation criterion (*CEC* [-]) combined by two statistical metrics including relative difference (*MRD* [-]) and standard deviation of the relative difference ($\sigma(RD)$ [-]) (Jacobs et al., 2004). Detailed description of above mentioned three metrics are given in Appendix B.3. It should be noted that the $\theta_{t,i}^{obs}$ and $\overline{\theta_t^{obs}}$ in Eqs. (B4) and (B5) represent the upscaled SM using four different methods and their average here when using the *CEC* to determine the best upscaled SM. The most representative time series is identified by the lowest *CEC* value."

**8.      p.10 l.321-324: The authors describe that some sites where damaged and new ones were built in an area of distinctly different conditions. I would like the authors to elaborate on how this is expected to affect a monotonic trend analysis, like M-K, and what exactly the reader can thus learn from this analysis.**

**Taking p.10 .329-334 into account, to me it seems like the long-term trends in the data are very much dependent on the locations of the sites and not caused by any other driver, like changing meteorological conditions. I think the trend analysis is helpful for understanding the dataset, but more from a data quality perspective, than to describe the study site (like on p.10 l.338-341). I would consider a rework of the whole section 4.2, to better highlight why the trend analysis was done and what the conclusions and implications for using the dataset are.**

Thanks for the comments and suggestions. The trend analysis in Section 4.2 has been thoroughly revised as shown below.

"Fig. 9a shows further the Mann Kendall trend test and Sen's slope estimate for the SM$_{AA-min}$, SM$_{AA-valid}$, and precipitation of the Maqu network area for the full year, warm seasons, and cold seasons in a 10-year period. As described in Section 3.2, the time series would present a monotonous trend if the absolute value of statistics *Z* is greater than a critical value, i.e. $Z_{0.05}=1.96$ in this study. The results show that there is not significant trend found for both precipitation and SM$_{AA-min}$ time series, while the SM$_{AA-valid}$ shows a drying trend with a Sen's slope of -0.008 for warm seasons. The drying trend of the SM$_{AA-valid}$ is caused by the change of available SM monitoring sites (see Table A2). Specifically, several monitoring sites (e.g. NST11- NST15) located in the wetter area were damaged since 2013, and four new monitoring sites (i.e. NST21- NST25) were installed in the drier area in 2015 (see Table 2) that affect the trend of the SM$_{AA-valid}$.

The Mann Kendall trend test and Sen's slope estimate for the SM$_{AA-min}$, SM$_{AA-valid}$, and precipitation time series of the Shiquanhe network area are shown in Fig. 9b. The SM$_{AA-min}$ demonstrates a wetting trend with a Sen's slope of 0.003, while an opposite drying tendency is found for the SM$_{AA-valid}$ due to the change of available SM monitoring sites (see Table A4) as the Maqu network. Specifically, several monitoring sites (e.g. SQ11 and SQ12)

located in the wetter area were damaged around 2014, and five new monitoring sites (i.e. SQ17-21) were installed in the drier area in 2016 (see Table 3).

[Figure]

**Fig. 9. Mann Kendall trend test and Sen's slope estimate for precipitation, SM_AA-min, SM_AA-valid, and model-based SM derived from the ERA5-land, GLDAS Noah, and MERRA2 for the (a) Maqu and (b) Shiquanhe networks in a 10-year period. "**

**9.      Discussion: From my point of view, major parts of the discussion need substantial revisions. The whole paragraph p.12 l.387 - 403 remains quite unclear to me. It is referring to Nine levels of RMSE standard (p.12 l.393), which are not further described in the methods. Please clarify what that is. Subsequently, different percentages of these levels are reported and discusses. From these numbers, it is concluded to shut a number of sites down. Why?**

**In the following paragraph p.12 l.404-418 this is applied to the networks but I honestly don't get the point of this paragraph. Why would you want to decrease the number of sites? What is a "Best level of RMSE" (l.412)? What does "all the possible combinations" (l.413-414) refer to and what is C12 17 = 6188? What is "the 100th combination of monitoring sites" (l.417-418) ?**

Thanks for the comments and suggestions. Section 5 has been thoroughly revised to make it more readable.

[revised manuscript text omitted]

In summary, it is suggested to maintain well current 12 monitoring sites for the Shiquanhe network, while for the Maqu network it is suggested to restore five old monitoring sites, i.e. CST02, NST11, NST13, NST14, and NST15."
We decrease the number of monitoring sites because some of them have been damaged. The monitoring sites described in Section 2.1 are all sites during the 10 years, however, until 2019 there are only 12 monitoring sites still working. So we need to select the appropriate strategies to maintain the networks that take into account data quality and practical feasibility.

**10.      Figure 1 (p.19): I would strongly recommend reworking this figure. The overview of all sites is of crucial importance to understand the dataset. Please consider including an overview map of Eastern Asia or China, to let the reader better locate the networks. Please consider using larger symbols and especially enlarge the CMA weather station symbols. It would also be helpful to include some larger cities into the maps if there are any. That will make the orientation easier. Maybe it's even a good idea to split this figure into multiple maps and add information like geology or landuse to the maps, if available.**
Thanks for the suggestions. This figure is divided into 4 figures in the revised manuscript, i.e. Figs. 1, 2, 4, and 5.

**11.      Finally, I really liked Table 1. Here the authors collected studies that were conducted on the different study sites, the data used, and key findings. I would consider briefly describing some of the most interesting findings in the introduction**

Thanks for the comments and suggestions. The descriptions related to Table 1 are provided on Page 2 Line 44-53: "The SM data collected from the Tibet-Obs has been widely used in past decade to calibrate/validate satellite- and model-based SM products (e.g. Su et al., 2013; Zheng et al., 2015a; Colliander et al., 2017), and to evaluate and develop SM upscaling methods (e.g. Qin et al., 2013; 2015), SM retrieval algorithms for microwave remote sensing (e.g. van der Velde et al., 2012; 2014a; Zheng et al., 2018a; 2018b; 2019) and fusion methods to merge in situ SM and satellite- or model-based products (e.g. Yang et al., 2020; Zeng et al., 2016).

Key information and outcomes of the main scientific applications using the Tibet-Obs SM data are summarized in Table 1. As shown in Table 1, the state-of-the-art satellite- and model-based products are useful but still show deficiencies to different degrees in different hydrometeorological conditions on the TP, and further evaluation and improvement of the latest versions of these products remain imperative."

The purpose of providing Table 1 is to highlight the importance of producing a long-term dataset and re-evaluating the satellite- and model-based products in a 10-year period.

**Technical comments**

**1. I would suggest to actually include the caption to Fig.6, instead of referencing another figure's caption.**

Thanks for the suggestion. This figure is merged into Fig. 8 with the caption of "Temporal variation of $SM_{AA-min}$, $SM_{AA-valid}$ and precipitation for the (a) Maqu and (b)Shiquanhe networks in a 10-year period as well as the subplot with a 2-year period."

**2. p.2 l.66: "Studies have also reported..."; It is a bit unclear for me, which studies the authors are referring to. The two by Su (sentence before), or the ones named in the following paragraph (Colliander, Quin, Dente..)?**

Thanks for the comment. The sentence is changed to "Many other studies also adopted the weighted averaging methods, whereby the weights are assigned to account for spatial heterogeneity within the network areas covered by *in situ* monitoring sites." on Page 2 Line 66-68, and the specific studies are made examples in the following sentences.

**3. p.2 l. 74-75: "several particular or alternative methods [...] were adopted in some investigations.";
I would suggest to either specify the actual methods and investigations or to drop the sentence.**

Thanks for the comment. The sentence is changed to "Alternative methods, such as time stability and ridge regression, have been adopted in other investigations (i.e. Zhao et al., 2013, Kang et al., 2017). While a large number of studies have assessed the performance of different upscaling methods in other areas such as the Tonzi Ranch network in California and the Heihe watershed ( Moghaddam et al., 2014, Wang et al., 2014), only few investigations have been done for the TP (Gao et al., 2017, Qin et al., 2015)." on Page 2-3 Line 74-78.

**4. p.3 l.79-83: The authors introduce their SM network here. I would suggest referencing Fig. 1 here, to point the reader to the overview map. This makes the whole section way clearer.**

Thanks for the suggestion. Fig. 1 is referenced here on Page 3 Line 103.

**5. p.4 l.111-112; l.126-127; l.141: here, the authors specify the soil texture for the different SM networks. Please clarify why only the soil texture is specified and not other soil properties? How is that expected to influence the measurements and the quality of the dataset? Where does this information come from?**

Thanks for the comments. This sentence about the soil texture is deleted. And the site information including soil texture, soil bulk density, and soil organic matter content is provided in the Table A1.The values of soil bulk and soil organic matter content are measured in the lab.

**6. p.4 .143: Just out of curiosity: How can you install a 5TE at 2.5cm depth? Can you comment on data quality here?**

Thanks for the comments. The photographs of probes installation are provided in Fig. 3.

At the Naqu site EC-10 sensors were installed in 2006 at depths of 2.5, 7.5., 15, 30 and 60 cm. In 2010, 5TM probe were installed at depths of 5, 10, 20, 40 and 80 cm. The performance of the 2.5 cm probes were verified using on-site gravimetric samples, which yield an RMSE of 0.029 $m^3$ $m^{-3}$.

The data quality is describe on Page 4 Line 121-124:

"The accuracy of the 5TM output was further improved via a soil-specific calibration performed for each soil type found in the Maqu network area (Dente et al. 2012a), leading to a decrease in the root mean square error (RMSE) from 0.06 to 0.02 $m^3$ $m^{-3}$ (Dente et al. 2012a)."

And on Page 5 Line 152-154:

"The Decagon 5TM ECH$_2$O probes were installed at depths of 5/2.5, 10/7.5, 15, 30, and 60 cm to measure the SMST, and the soil-specific calibration was also performed by van der Velde (2010) that yields a RMSE of about 0.029 $m^3$ $m^{-3}$."

Please note that 5TM and EC-10 probes have been installed as part of Tibet-Obs. 5TE are not included in the network.

**7. p.5 l.146 "only the Naqu and MS have collected"; what is MS referring to here?**

Thanks for the comment. Naqu and MS are the monitoring site names, it is changed to "Naqu and MS sites" on Page 5 Line 155.

**8. p.5. l.151 "The daily precipitation from two weather stations (Fig. 1)"; I think the symbols on the map are way too small. I can't really see the weather stations in Fig.1**

Thanks for the comment. Fig. 1 is revised to make the symbols clearer.

[Figure]

✷ CMA weather station
■ Tibet-Obs network

**Fig. 1. Locations of the Tibet-Obs including Maqu, Naqu, and Ngari (including Ali and Shiquanhe) soil moisture monitoring networks. The weather stations of Maqu and Shiquanhe operated by the China Meteorological Administration (CMA) are also shown. (Base map is from Esri, Copyright: © Esri)**

**9.     p.5 l.158 What is the ECMWF?**

Thanks for the comment. The full name of ECMWF is "European Centre for Medium-Range Weather Forecasts" provided on Page 5 Line 168.

**10.     p.5 l.161 What is H-TESSEL and CY45R1?**

Thanks for the comment. These two abbreviations are deleted.

**11.     p.7 l.210: "the average SM data"; What is the average SM data? I am not sure what average is referring to here.**

Thanks for the comment. We rewrote this section of "3.2 Trend analysis" to make the Mann Kendall trend test and Sen's slope estimate clearer.

**12.     p.8 l.267: If Eq. (A3) gives the bias of the upscaled product, what does Eq. (3a) then calculate? Is it the same, or another bias? Please clarify.**

Thanks for the comment. Eq. (A3) is the equation of standard deviation of the relative difference, while Eq. (3a) is the equation of bias. This sentence is deleted in the revised manuscript.

**13.     p.9. l.311 - p.10 l.312: With an extent of 10 years on the x-axis, it is very hard for me to see the described differences in value. I can hardly see the data. Please consider adding another plot or subplot with a shorter time span.**

Thanks for the comment. The subplots with a 2-year time span are provided in Fig. 8.

[Figure]

**Fig. 8.  Temporal variation of $SM_{AA-min}$, $SM_{AA-valid}$, and precipitation for the (a) Maqu and (b) Shiquanhe networks in a 10-year period as well as the subplot with a 2-year period.**

**The dataset**

**These comments concern only the dataset itself given at**

**https://doi.org/10.4121/uuid:21220b23-ff36-4ca9-a08f-ccd53782e834**

**1.        The most important part is already mentioned in the main comments, point 1. I would strongly encourage the authors to include all used data into the data repository.**

**2.        I think the dataset should be enriched with some more metadata. E.g. which sensor was used? What is the unit? How precise are the measurements? How does the study site look like? How does the terrain look like? Please also refer to my main comment 2.**

**3.        It would be most helpful to include the soil temperature and electric conductivity data recorded by the 5TE into the dataset. That would tremendously foster the re-usability of the dataset. In addition, is there a specific reason why the other depths are not included?**

**4.        Please consider extending the dataset description on the source. E.g. include the upscaling method used, name the sensors used or give more details on the context and catchment the sites are operated in. The data source itself should contain all necessary information to use the data. The authors could reuse many parts of this paper here. As a personal remark, I found Table 1 of this publication really useful. Together with an overview plot of all measurements, it could be added to the user information PDF as well.**

**5.        The manuscript reports on four different upscaling methods, but only one upscaled result is included in the source. I would suggest to include the other upscaled products and model-based products, as well.**

**6.        minor: Please consider using an open data format like ODS, CSV or HDF5 instead of XLSX. The relevant software needed to open XLSX is neither free nor open.**

Thanks for the comments and suggestions for the dataset. Most of the suggestions are adopted in the updated dataset to make it more useful.

Document of user information has been revised to include descriptions of monitoring sites and summary of previous applications using the Tibet-Obs data. The precipitation data, the upscaled soil moisture (SM) data produced by the four upscaling methods for a single year, and the 10-year SM data from three model-based products (i.e. ERA5-land, GLDAS Noah, and MERRA2) at the Maqu and Shiquanhe network areas have been added to the dataset (as shown in the response to Main Comment 1).

We are still working on controlling the data quality of temperature and SM data at other depths, as well as selecting appropriate method to upscale them for a 10-year period. Nevertheless, the surface SM dataset produced in this study is valuable for calibrating/validating long-term satellite-based SM products, evaluation of SM upscaling methods, and development of data fusion methods. Therefore, we choose to report this dataset first.

**Response to Reviewer #2**

We would like to thank the reviewer for carefully reading our manuscript and providing detailed and constructive comments. This has helped us to improve our manuscript significantly. In the text below we provide our response to each comment point by point.

Reviewer's comments are in **bold**.

Author's responses are in regular.

Author's additions/modifications in the text are in blue.

**1.       MAJOR: The description of the dataset needs some improvements. I am aware that the Tibetan Plateau observatory has been already described in several previous papers (Table 1 is great), but in the online dataset more details should be added (soil, land use, climatology, pictures, ······).**

Thanks for the comments and suggestions. All available information of the monitoring sites is provided in the revised manuscript in Section 2.1 and Appendix A, including the elevation, topography, land cover, soil texture, soil bulk density, and soil organic matter content that are listed in Tables A1, A3, and A5 for the Maqu, Ngari, and Naqu networks, respectively. Photographs selected from some monitoring sites are shown in Figs. 2, 4, and 5 for the Maqu, Ngari, and Naqu networks, respectively. Examples of profile measurement setup are shown in Fig. 3.

[revised manuscript text omitted]

**2.      MAJOR: I have found the paper a compromise between a data paper and a scientific paper describing and testing upscaling procedures. I believe that in the dataset all the upscaled soil moisture time series should be available. Indeed, I have found weak the reasons for selecting the arithmetic average as reference method (ground truth). I guess that its better performance is very related to the employed metric (CEC). Of course, a reference does not exist, and hence I would suggest to publish and to make the analysis (e.g., comparison with modelled data), for all upscaled time series.**

The precipitation data, the upscaled soil moisture (SM) data produced by the four upscaling methods for a single year, and the 10-year SM data from three model-based products (i.e. ERA5-land, GLDAS Noah, and MERRA2) at the Maqu and Shiquanhe network areas have been added to the dataset in the 4TU.ResearchData repository, as shown in the following table.

| Folder | File/subfolder | Sheet |
|---|---|---|
| \\*In situ* soil moisture\\ | \\Maqu.xlsx | - Information: This sheet contains the information of all the monitoring sites including location, elevation, topography, land cover, soil texture and soil organic matter content.
- 2009-2019: These 11 sheets contain the available original 15-min/30-min *in situ* measurements at 5 cm depth for each year. |
| | \\Shiquanhe.xlsx | - Information: This sheet contains the information of all the monitoring sites including location, elevation, topography, land cover, soil texture and soil organic matter content.
- 2010-2019: These 10 sheets contain the available original 15-min *in situ* measurements at 5 cm depth for each year. |
| | \\Ali.xlsx | - Information: This sheet contains the information of monitoring sites including location, elevation, topography, land cover, soil texture and soil organic matter content.
- 2010-2018: These 9 sheets contain the available original 15-min *in situ* measurements at 5 cm depth for each year. |
| | \\Naqu.xlsx | - Information: This sheet contains the information of monitoring sites including location, elevation, topography, land cover, soil texture and soil organic matter content.
- 2010-2019: These 10 sheets contain the available original 15-min *in situ* measurements at 5/2.5 cm depth for each year. |
| \\Upscaled soil moisture\\ | \\Maqu upscaled.xlsx | - 2009-2019: These 11 sheets contain spatial upscaled soil moisture with the input of original *in situ* measurement between 5/15/2009 and 5/15/2019. |
| | \\ Maqu upscaled-daily average.xlsx | - 2009-2019: These 11 sheets contain spatial upscaled soil moisture with the input of daily average measurement between 5/15/2009 and 5/15/2019. |

| | | |
|---|---|---|
| | \Shiquanhe upscaled.xlsx | - 2010-2019: These 10 sheets contain spatial upscaled soil moisture with the input of original *in situ* measurement between 8/1/2010 and 8/1/2019. |
| | \Shiquanhe upscaled-daily average.xlsx | - 2010-2019: These 10 sheets contain spatial upscaled soil moisture with the input of daily average measurement between 8/1/2010 and 8/1/2019 |
| \ Supplementary data \ | \ Upscaled -daily average.xlsx | - Maqu: This sheet contains spatial upscaled soil moisture with the input of daily mean from 17 sites (3 sites) between 11/1/2009 and 10/31/2010
- Shiquanhe: This sheet contains spatial upscaled soil moisture with the input of daily mean from 12 sites (4 sites) between 8/1/2018 and 7/31/2019 |
| | \Model-based products.xlsx | - Maqu: This sheet contains daily regional mean soil moisture of Maqu network area (Fig. 2) from the ERA5-land, GLDAS Noah, and MERRA2 products between 5/15/2009 and 5/15/2019
- Shiquanhe: This sheet contains daily regional mean soil moisture of Shiquanhe network area (Fig. 3) from the ERA5-land, GLDAS Noah, and MERRA2 products between 8/1/2010 and 8/1/2019 |
| | \Precipitation.xlsx | - Maqu: This sheet contains daily precipitation of Maqu weather station (Fig. 1) between 5/15/2009 and 5/15/2019
- Shiquanhe: This sheet contains daily precipitation of Shiquanhe weather station (Fig. 1) between 8/1/2010 and 8/1/2019 |

The comparison of four upscaling methods in a single year is a key step for selecting the appropriate upscaling method for producing the 10-year dataset and quantifying the uncertainty of this dataset since the number of available SM monitoring sites in the Tibet-Obs generally changes with time (see Tables 2-4 and Tables A2 and A4 for the details). Based on the comparison (see Section 4.1 for the details), we found that the arithmetic averaging method (AA) is more suitable for the Tibet-Obs, and the AA method with input of minimum number of available SM monitoring sites (i.e. $AA_{min}$, 3 and 4 sites for the Maqu and Shiquanhe networks) is more suitable for producing the 10-year dataset with RMSEs of about 0.022 and 0.011 $m^3$ $m^{-3}$ for the Maqu and Shiquanhe networks (see Figs. 7 and B2 and Table B2). As such, the $AA_{min}$ is further adopted to produce the 10-year upscaled SM dataset that is the main and core data product of this publication. Therefore, we think it is better to keep the comparison of four upscaling methods in this manuscript instead of presenting them in the other paper.

Relevant figures and table above mentioned are shown below:

[Figure]

**Fig. 7. Comparisons of daily average SM for the (a) Maqu and (b) Shiquanhe networks produced by the four upscaling methods with input of the minimum number of available SM monitoring sites.**

[Figure]

**Fig. B2. Radar graph of error statistics (i.e. RMSE and NSE) computed between the SM_truth produced by the AA-max and the upscaled SM produced by the four upscaling methods with input of different number of available monitoring sites for the (a) Maqu and (b) Shiquanhe networks.**

**Table B2. Error statistics computed between the SM obtained by the four upscaling methods with input of the minimum available monitoring sites and the SM_truth produced by the AA-max for the Maqu and Shiquanhe networks.**

|  | Bias ($m^3\,m^{-3}$) | RMSE($m^3\,m^{-3}$) | ubRMSE($m^3\,m^{-3}$) | NSE |
|---|---|---|---|---|
|  | | Maqu | | |
| AA-min | 0.005 | 0.022 | 0.021 | 0.954 |
| TS-min | 0.025 | 0.050 | 0.044 | 0.747 |
| VD-min | -0.007 | 0.022 | 0.020 | 0.954 |
| ATI-min | -0.052 | 0.099 | 0.084 | 0.030 |
|  | | Shiquanhe | | |
| AA-min | 0.010 | 0.011 | 0.005 | 0.816 |
| TS-min | -0.001 | 0.013 | 0.013 | 0.768 |
| VD-min | 0.019 | 0.020 | 0.006 | 0.400 |
| ATI-min | -0.001 | 0.021 | 0.021 | 0.393 |

According to the reviewer's suggestion, the reference about the arithmetic average in other study is provided on Page 9 Line 300-301:

"The arithmetic average of the dense *in situ* measurements was also used as the ground truth in other studies (Qin et al., 2013; Su et al., 2013)."

And the upscaled SM data produced by the four upscaling methods for a single year has been added to the dataset in the 4TU.ResearchData repository. In addition, several figures and tables were moved to Appendix B to make the manuscript more focused on the description of generated 10-year dataset.

**3. MODERATE: The ATI upscaling method is not clear to me. Where did you get the data for computing ATI? Why is its performance so different from the other upscaling techniques? Why is the range of soil moisture with ATI approach much smaller than other approaches? Please, provide more details.**

Thanks for the comments and suggestions. More details of the ATI method are provided in the Appendix B.4.

[revised manuscript text omitted]

**4.      MODERATE: The results of trend analysis are not clear to me. What do the authors want to highlight with this analysis? Why showing both UF and UB metrics? Also, the interpretation of results is unclear. By looking at figures, the trend of SM and precipitation is not consistent for Maqu site (line 319). For the same site, positive and negative values of UF are present, therefore I would not conclude for drying trend (line 338). All trends are not significant (see line 320) as UF and UB values are lower than critical values. All these points should be addressed.**

Thanks for the comments and suggestions. The trend analysis is an aspect to evaluate the SM data/products. In the revised manuscript, the 10-year Mann Kendall trend test and Sen's slope estimate is thoroughly revised. The description of the method is rewrote in Section 3.2.

"The Mann-Kendall test and Sen's slope estimate (Gilbert, 1987; Mann, 1945; Smith et al., 2012) are adopted in this study to analyze the trend of 10-year upscaled SM time series and model-based products (i.e. ERA5-land, GLDAS Noah, and MERRA2). Specifically, the trend analysis is based on the monthly average SM, and all the missing data is regarded as an equal value smaller than other valid data. The test consists of calculating the seasonal statistics S and its variance VAR(S) separately for each month during the 10-year period, and the seasonal statistics are then summed to obtain the Z statistics.

For the month $i$ (e.g. January), the statistics $S_i$ can be computed as:

$$S_i = \sum_{k=1}^{9} \sum_{l=k+1}^{10} sgn(SM_{i,l} - SM_{i,k}) \tag{2a}$$

$$sgn(SM_{i,l} - SM_{i,k}) = \begin{cases} 1 & SM_{i,l} > SM_{i,k} \\ 0 & SM_{i,l} = SM_{i,k} \\ -1 & SM_{i,l} < SM_{i,k} \end{cases}$$

where $k$ and $l$ represent the different year and $l > k$, $SM_{i,l}$ and $SM_{i,k}$ represent the monthly average SM for the month $i$ of the year $k$ and $l$, respectively.

The $VAR(S_i)$ is computed as:

$$VAR(S_i) = \frac{1}{18}[N_i(N_i - 1)(2N_i + 5) - \sum_{p=1}^{g_i} t_{i,p}(t_{i,p} - 1)(2t_{i,p} + 5)] \tag{2b}$$

where $N_i$ is the length of the record for the month $i$ (e.g. the 10 year data record in this study with $N_i=10$), $g_i$ is the number of equal-value data in month $i$, $t_{i,p}$ is the number of equal-value data in the $p^{th}$ group for month $i$.

After obtaining the $S_i$ and $VAR(S_i)$, the statistic $S'$ and $VAR(S')$ for the selected season (e.g. warm season between May and October and cold season between November and April in this study) can be summed as:

$$S' = \sum_{i=1}^{M} S_i \tag{2c}$$

$$VAR(S') = \sum_{i=1}^{M} VAR(S_i) \tag{2d}$$

where M represents the number of months in the selected season, e.g. M = 12 for the full year, while M = 6 for the warm and cold season, respectively.

Then the statistics Z can be computed as:

$$Z = \begin{cases} \frac{S'-1}{\sqrt{Var(S')}} & if \ S' > 0 \\ 0 & if \ S' = 0 \\ \frac{S'+1}{\sqrt{Var(S')}} & if \ S' < 0 \end{cases} \tag{2e}$$

If the statistics $Z$ is positive (negative) and its absolute value is greater than $Z_{1-\alpha/2}$ (here $\alpha = 0.05$, $Z_{1-\alpha/2} = 1.96$), the trend of the SM time series is regarded as upward (downward) at the significance level of $\alpha$. Otherwise, we accept the hypothesis that there is not significant trend found for the SM time series.

If the trend is monotonous, we will further estimate the slope (change per unit time) with Sen's method (Sen, 1968). The slopes of each month can be calculated as:

$$Q_i = \frac{SM_{i,l} - SM_{i,k}}{l-k} \tag{2f}$$

Then rank all the individual slopes ($Q_i$) for all months and find the median which is considered as the seasonal Kendall slope estimate."

The results of Mann Kendall trend test and Sen's slope estimate for 10-year SM and precipitation time series at the Maqu and Shiquanhe networks area are presented in Fig. 9.

[Figure]

**Fig. 9. Mann Kendall trend test and Sen's slope estimate for precipitation, SM$_{AA-min}$, SM$_{AA-valid}$, and model-based SM derived from the ERA5-land, GLDAS Noah, and MERRA2 for the (a) Maqu and (b) Shiquanhe networks in a 10-year period.**

The trend analysis in Section 4.2 has been thoroughly revised as shown below:

"Fig. 9a shows further the Mann Kendall trend test and Sen's slope estimate for the SM$_{AA-min}$, SM$_{AA-valid}$, and precipitation of the Maqu network area for the full year, warm seasons, and cold seasons in a 10-year period. As described in Section 3.2, the time series would present a monotonous trend if the absolute value of statistics $Z$ is greater than a critical value, i.e. $Z_{0.05} = 1.96$ in this study. The results show that there is not significant trend found for both precipitation and SM$_{AA-min}$ time series, while the SM$_{AA-valid}$ shows a drying trend with a Sen's slope of -0.008 for warm seasons. The drying trend of the SM$_{AA-valid}$ is caused by the change of available SM monitoring sites (see Table A2). Specifically, several monitoring sites (e.g. NST11- NST15) located in the wetter area were damaged since 2013, and four new monitoring sites (i.e. NST21- NST25) were installed in the drier area in 2015 (see Table 2) that affect the trend of the SM$_{AA-valid}$.

The Mann Kendall trend test and Sen's slope estimate for the SM$_{AA-min}$, SM$_{AA-valid}$, and precipitation time series of the Shiquanhe network area are shown in Fig. 9b. The SM$_{AA-min}$ demonstrates a wetting trend with a Sen's slope of 0.003, while an opposite drying tendency is found for the SM$_{AA-valid}$ due to the change of available SM monitoring sites (see Table A4) as the Maqu network. Specifically, several monitoring sites (e.g. SQ11 and SQ12) located in the wetter area were damaged around 2014, and five new monitoring sites (i.e. SQ17-21) were installed in the drier area in 2016 (see Table 3)."

The trend analysis of model-based products in Section 4.3 has been revised as shown below:

"The trend analysis for the three model-based SM products are shown in Fig. 9a as well. Both the GLDAS Noah and MERRA2 products do not show significant trend in the warm season as the SM$_{AA-min}$, while a drying trend is observed for the ERA-land product. In addition, both the GLDAS Noah and MERRA2 products show a wetting trend in the cold season that disagree with the SM$_{AA-min}$.

The trend analysis for the three SM products are also shown in Fig. 9b. Both the ERA5-land and MERRA2 products are able to reproduce the wetting trend found for the SM$_{AA-min}$, while the GLDAS Noah product cannot capture well the trend."

**5.      MAJOR: I believe that the SM AA-valid time series is wrong and should not been used. Of course, by averaging different sensors varying over time is not a correct procedure. I suggest removing this time series.**

Thanks for the comments and suggestions. The purpose of providing SM$_{AA-valid}$ time series is to highlight the uncertainties caused by the inconsistent measurements from the data quality perspective. The inconsistent trend produced by the SM$_{AA-valid}$ as shown in Fig. 9 indicates that it is not reasonable to make full use of the available inconsistent measurements for production of the long-term consistent upscaled SM time series. In addition, the variation of SM$_{AA-valid}$ time series can provide a reference for making the maintaining strategy of Tibet-Obs as discussed.

Relevant information about above mentioned idea is provided in Section 4.2 on Page 10 Line 341-342:

"Trend analysis (see Section 3.2) are applied to both SM$_{AA-min}$ and SM$_{AA-valid}$ to investigate the impact of change of available SM monitoring sites over time on the long-term (i.e. 10-year) trend."

and on Page 11 Line 353-356:

"The drying trend of the $SM_{AA\text{-valid}}$ is caused by the change of available SM monitoring sites (see Table A2). Specifically, several sites (e.g. NST11- NST15) located in the wetter area were damaged since 2013, and four new sites (i.e. NST21- NST25) were installed in the drier area in 2015 (see Table 2) that affect the trend of the $SM_{AA\text{-valid}}$."

and on Page 11 Line 362-366:

"The $SM_{AA\text{-min}}$ demonstrates a wetting trend for both seasons, while an opposite drying tendency is found for the $SM_{AA\text{-valid}}$ in cold season due to the change of available SM monitoring sites (see Table A4) as the Maqu network. Specifically, several sites (e.g. SQ11 and SQ12) located in the wetter area were damaged around 2014, and five new sites (i.e. SQ17-21) were installed in the drier area in 2016 (see Table 3). "

and on Page 11 Line 367-369:

"In summary, the $SM_{AA\text{-valid}}$ are likely affected by the change of available SM monitoring sites over time that leads to inconsistent trend as the $SM_{AA\text{-min}}$. This indicates that the $SM_{AA\text{-min}}$ is superior to the $SM_{AA\text{-valid}}$ for the production of the long-term consistent upscaled SM time series."

**6.      MODERATE: In the results and discussion sections several small errors (typo) are present that should be corrected. A quick reread of these sections will fix these errors.**

Thanks for your suggestions, several parts of these two sections are rewritten and the errors/typos are corrected in the revised manuscript.

---

## Referee Report (RR1)

**Comment to Author's Response**

**Feb. 2021**

The authors made substantial efforts to improve the presented manuscript. I found most comments to be taken into account and major parts of the manuscript be revised. Especially figures were enhanced and their readability was substantially improved. Numerous tables were added, completing the research site and data descriptions. I am especially thankful for the photographs that were added to Fig. 2, 3, 4 and 5.

I would like to highlight, that the data repository was updated, as well. From my point of view it does now contain all needed data. However, the authors included more specific download links into the revised manuscript. My concerns regarding using URLs to reference data sources still hold. Additionally, the URLs still lack vital bibliographic information like the access date, authors or institution. I would still suggest to further describe the downloaded data properly (e.g. origin, units, aggregation level, quality control, support, licenses) and reference the source properly, so that users do not rely on URLs. Most of this info is already present in the manuscript and adding it to the data repository as well, should be straightforward.

I tried to retrace the download procedure. For the link on p.5 line 163, I ended up on a Chinese page, which redirected to the authority landing page, once switched to English language and thus the issues with URL persists. It would be great, as permanent URLs or DOIs don't seem to exists, if the authors could add the metadata about the downloaded data to the data repository, as well. From my point of view it doesn't make sense to forward to the original publisher of the data to find metadata in this case.

The URL on p. 5 line 172 now leads to the download page of the data product, however there are still literally hundreds of filter possibilities that one has to or can specify. At the same time, the overall ERA5 product has a DOI linked on that page, which resolves to a landing page, that presents a lot of metadata about the entire ERA5 dataset. My suggestion here would be to specify, which parts of ERA5 were exactly used (and how) and extract the metadata from the ERA5 landing page that applies and add it to the data repository.

For the link on page 6 line 181, the correct data product is found, however, almost 15 thousand satellite images are offered for download, which requires

authentication. Therefore, I would again suggest to add necessary metadata from the NASA website to the data repository.

All in all, I think it's just an additional table, or something comparable, that is missing in the data repository. It should give all the available, necessary metadata for the newly added datasets, making the repository usable on its own. The descriptions of the networks are already detailed and helpful. Just chapter 4 of the user information needs to be raised to the same level. Then, the URLs can be removed from the manuscript and replaced by a simple reference to the issuing institution. From my point of view, that would turn the already good data repository into a great one, just like the manuscript.

With kind regards,

Mirko Mälicke

---

## Author Response (AR2)

Dear Editor and Reviewers,

Thank you very much for the second round of reviewing our revised manuscript and providing detailed comments regarding to the dataset. The data publication and manuscript have been revised based on the comments made by both reviewers.

Please find below our detailed response to each comment made by the reviewers.

We think that the revised manuscript and data publication have appropriately addressed all the reviewers' concerns and we hope that you can consider it for publication in Earth System Science Data.

Sincerely,

Pei Zhang

On behalf of all co-authors

**Response to Reviewer #1**

We would like to thank the reviewer for carefully reading our manuscript and providing detailed and constructive comments. In the text below we provide our response to each comment point by point.

Reviewer's comments are in **bold**.

Author's responses are in regular.

Author's additions/modifications in the text are in blue.

**I would like to highlight, that the data repository was updated, as well. From my point of view it does now contain all needed data. However, the authors included more specific download links into the revised manuscript. My concerns regarding using URLs to reference data sources still hold. Additionally, the URLs still lack vital bibliographic information like the access date, authors or institution. I would still suggest to further describe the downloaded data properly (e.g. origin, units, aggregation level, quality control, support, licenses) and reference the source properly, so that users do not rely on URLs. Most of this info is already present in the manuscript and adding it to the data repository as well, should be straightforward.**

Thanks for the comments and suggestions. The links of the three model-based products and the last verified date are updated in the revised manuscript.

On Page 5 Line 168-172:

"ERA5-land is a reanalysis dataset produced by running land component of the ECMWF (European Centre for Medium-Range Weather Forecasts) ERA5 climate reanalysis (Albergel et al., 2018). ERA5-land provides SM data currently available from 1981 to present at hourly time interval with a spatial resolution of 0.1°, and the data is available from https://cds.climate.copernicus.eu/cdsapp#!/dataset/reanalysis-era5-land?tab (last verified on 11 March 2021)."

On Page 5 Line 176-180:

"MERRA2 is an atmospheric reanalysis dataset produced by NASA using the Goddard Earth Observing System Model version 5 (GEOS-5) and atmospheric data assimilation system (ADAS), version 5.12.4. MERRA2 provides SM data currently available from 1980 to present at hourly time interval with a spatial resolution of 0.5° (latitude) by 0.625° (longitude), and the data is available from https://disc.gsfc.nasa.gov/datasets/M2T1NXLND_5.12.4 /summary (last verified on 11 March 2021)."

On Page 6 Line 184-188:

"GLDAS-2.1 Noah is forced by a combination of model-based and observation data including Global Precipitation Climatology Project (GPCP) version 1.3, and simulated with the Noah Model 3.6 in Land Information System (LIS) version 7. GLDAS-2.1 Noah provides SM data currently available from 2000 to present at 3-hourly time interval with a spatial resolution of 0.25°, and the data is available from https://disc.gsfc.nasa.gov/datasets/ GLDAS_NOAH025_3H_2.1/summary (last verified on 11 March 2021)."

The descriptions of the dataset catalog entries on the link interface (e.g. Overview, Documentation, Quality assessment, data citation, etc.) are introduced in the user guide of the dataset in the 4TU.ResearchData repository.

**4.1 ERA5-land soil moisture product**

ERA5-land is a reanalysis dataset produced by running land component of the ECMWF (European Centre for Medium-Range Weather Forecasts) ERA5 climate reanalysis (Albergel et al., 2018). ERA5-land provides SM data currently available from 1981 to present at hourly time interval with a spatial resolution of 0.1°. More information about the ERA5-land product can be referred to Muñoz-Sabater et al., (2018). The data (2009-2019) of volumetric total soil water content for the top soil layer (0-7 cm) in Maqu and Shiquanhe network areas is put in our dataset (Table 8).

Downloading online ERA5-land data through CDS (climate data store) website interface:

1. Register a Copernicus account.
2. Go to the https://cds.climate.copernicus.eu/cdsapp#!/dataset/reanalysis-era5-land?tab
3. The dataset catalogue entries include the following tabs:
   - **Overview:** It gives a description of the dataset and metadata information (e.g. data description and main variables).
   - **Download data:** It is a download web form.
   - **Quality assessment:** It is a new feature, work in progress (The CDS datasets are assessed by the Evaluation and Quality Control (EQC) function of C3S independently of the data supplier).
   - **Documentation:** It provides links to details documentation about the dataset.
4. Go to the download data tab to select the required data.
   - **Variable:** Select the Soil Water, Volumetric soil water layer 1
   - **Year:** Select 2009-2019
   - **Month:** Select all
   - **Day:** Select all
   - **Time:** Select all
   - **Geographical area:** Select sub-region extraction, Maqu (33.5°-34.25° N, 101.63°-102.75° E), Shiquanhe (32.36°-32.76° N, 79.75°-80.25° E)
   - **Format:** NetCDF
5. Click the **Submit Form** and wait for the request processing (about several hours), until the green button **download** appears, you can click it and download the data.

**4.2 MERRA2 soil moisture product**

MERRA2 is an atmospheric reanalysis dataset produced by NASA using the Goddard Earth Observing System Model version 5 (GEOS-5) and atmospheric data assimilation system (ADAS), version 5.12.4. MERRA2 provides SM data currently available from 1980 to present at hourly time interval with a spatial resolution of 0.5° (latitude) by 0.625° (longitude). More information about the MERRA2 product can be referred to GMAO (2015). The data

(2009-2019) of volumetric liquid soil water content for the surface layer (0-5 cm) in Maqu and Shiquanhe network areas is put in our dataset (Table 8).

Downloading online MERRA2 data through GES DISC (Goddard Earth Sciences Data and Information Service Center) website interface:

1. Register an EARTHDATA account.
2. Go to the https://disc.gsfc.nasa.gov/datasets/M2T1NXLND_5.12.4/summary
3. The dataset catalogue entries include the following tabs:
   - **Product Summary:** It gives a description of the dataset and metadata information (e.g. temporal spatial, file format, etc.).
   - **Data citation:** To cite the data in publications.
   - **Documentation:** It provides links to details documentation about the dataset.
   - **Reference:** It is data collection reference.
4. Click the button of **Subset/ Get data** on the right of the interface to select the required data.
   - **Download Method:** Select the Get File Subsets using the GES DISC Subsetter
   - **Refine Date Range:** 2009-01-01 to 2019-12-31
   - **Refine Region:** Maqu (101.63, 33.5, 102.75, 34.25), Shiquanhe (79.75, 32.36, 80.25, 32.76)
   - **Variable:** Select SFMC = water surface layer
   - **Time of Day:** Get complete time span
   - **Grid:** bilinear interpolation on GLDAS-2_0.25
   - **Output format:** NetCDF
5. Click the **Get Data** and **Down load links list**, and then the tool like **Chrono Download Manager** can be used to download the data via the links list.

**4.3 GLDAS Noah soil moisture product**

GLDAS-2.1 Noah is forced by a combination of model-based and observation data including Global Precipitation Climatology Project (GPCP) version 1.3, and simulated with the Noah Model 3.6 in Land Information System (LIS) version 7. GLDAS-2.1 Noah provides SM data currently available from 2000 to present at 3-hourly time interval with a spatial resolution of 0.25°. More information about the GLDAS Noah product can be referred to Rodell et al. (2004). The data (2009-2019) of soil water content for the top soil layer (0-10 cm) in Maqu and Shiquanhe network areas is put in our dataset (Table 8).

Downloading online GLDAS Noah data through GES DISC (Goddard Earth Sciences Data and Information Service Center) website interface:

1. Register an EARTHDATA account.
2. Go to the https://disc.gsfc.nasa.gov/datasets/GLDAS_NOAH025_3H_2.1/summary
3. The dataset catalogue entries include the following tabs:
   - **Product Summary:** It gives a description of the dataset and metadata information (e.g. temporal spatial, file format, etc.).
   - **Data citation:** To cite the data in publications.

- **Documentation:** It provides links to details documentation about the dataset.
- **Reference:** It is data collection reference.

4. Click the button of **Subset/ Get data** on the right of the interface to select the required data.

- **Download Method:** Select the Get File Subsets using the GES DISC Subsetter
- **Refine Date Range:** 2009-01-01 to 2019-12-31
- **Refine Region:** Maqu (101.63, 33.5, 102.75, 34.25), Shiquanhe (79.75, 32.36, 80.25, 32.76)
- **Variable:** Select SoilMoi0_10cm_inst = Soil moisture content (0-10 cm underground) (kg m$^{-2}$)
- **Time of Day:** Get complete time span
- **Grid:** bilinear interpolation on GLDAS-2_0.25
- **Output format:** NetCDF

5. Click the **Get Data** and **Down load links list**, and then the tool like **Chrono Download Manager** can be used to download the data via the links list.

**Table 8. Specification of the model-based soil moisture products.**

| Data set | Variable | Unit | Spatial | Temporal | Period | Reference |
|---|---|---|---|---|---|---|
| ERA5-land | Volumetric soil water layer 1 (swvl1) | m$^3$ m$^{-3}$ | 0.1° | Hourly | 1981~ | Muñoz-Sabater. et al., (2018) |
| GLDAS Noah | SoilMoi0_10cm_inst | kg m$^{-2}$ | 0.25° | 3-hourly | 2000 ~ | Rodell et al., (2004) |
| MERRA2 | Water surface layer (SFMC) | m$^3$ m$^{-3}$ | 0.5° ×0.625° | Hourly | 1980 ~ | GMAO (2015) |

**I tried to retrace the download procedure. For the link on p.5 line 163, I ended up on a Chinese page, which redirected to the authority landing page, once switched to English language and thus the issues with URL persists. It would be great, as permanent URLs or DOIs don't seem to exists, if the authors could add the metadata about the downloaded data to the data repository, as well. From my point of view it doesn't make sense to forward to the original publisher of the data to find metadata in this case.**

Thanks for the comments. We asked the official website (https://data.cma.cn/) and be told that the online precipitation data is only available to agreement users and does not support sharing currently. Thus the corresponding part in the user guide and manuscript has been revised to explain this issue, and the link still remains for users who have downloading permission.

In the revised manuscript On Page 5 Line 160-165:

"The precipitation data is from two weather stations, i.e. Maqu (34°00'N, 102°05'E) and Shiquanhe (32°30'N, 80°05'E), operated by the China Meteorological Administration (CMA) which provides the near-surface meteorological data of about 700 weather stations in China. The daily precipitation data can be downloaded from https://data.cma.cn/dataService/cdcindex/datacode/SURF_CLI_CHN_MUL_DAY.html that is in Chinese. The data is only available to agreement users, which is not allowed to be shared without permission from the CMA."

In the user guide:

**4.4 Precipitation data**

The precipitation data is from two weather stations, i.e. Maqu (34°00'N, 102°05'E) and Shiquanhe (32°30'N, 80°05'E), operated by the China Meteorological Administration (CMA) which provides the near-surface meteorological data of about 700 weather stations in China. The daily precipitation data can be downloaded from https://data.cma.cn/dataService/cdcindex/datacode/SURF_CLI_CHN_MUL_DAY.html that is in Chinese. The data is only available to agreement users, which is not allowed to be shared without permission from the CMA.

**The URL on page. 5 line 172 now leads to the download page of the data product, however there are still literally hundreds of filter possibilities that one has to or can specify. At the same time, the overall ERA5 product has a DOI linked on that page, which resolves to a landing page, that presents a lot of metadata about the entire ERA5 dataset. My suggestion here would be to specify, which parts of ERA5 were exactly used (and how) and extract the metadata from the ERA5 landing page that applies and add it to the data repository.**

**For the link on page 6 line 181, the correct data product is found, however, almost 15 thousand satellite images are offered for download, which requires authentication. Therefore, I would again suggest to add necessary metadata from the NASA website to the data repository.**

Thanks for the comments and suggestions. The details of filtering and downloading required data and description of the metadata are introduced in the user guide as shown above.

**All in all, I think it's just an additional table, or something comparable, that is missing in the data repository. It should give all the available, necessary metadata for the newly added datasets, making the repository usable on its own. The descriptions of the networks are already detailed and helpful. Just chapter 4 of the user information needs to be raised to the same level. Then, the URLs can be removed from the manuscript and replaced by a simple reference to the issuing institution. From my point of view, that would turn the already good data repository into a great one, just like the manuscript.**

Thanks for the comments and suggestions. Chapter 4 of the user guide and related parts in the manuscript have been revised as shown above.

**Response to Reviewer #2**

We would like to thank the reviewer for carefully reading our manuscript and providing detailed and constructive comments. In the text below we provide our response to each comment point by point.

Reviewer's comments are in **bold**.

Author's responses are in regular.

Author's additions/modifications in the text are in blue.

**The authors have addressed most of the reviewers' comments and I have one minor comments remaining.**

**I have asked for more details in the description of the ONLINE dataset. Currently onky few lines. Please improve it for making it more appewling to the users (e.g., adding the table given in the replies to reviewers).**

Thanks for the comments, the descriptions of online dataset have been revised in Chapter 4 of the user guide:

**4. Online dataset**

**4.1 ERA5-land soil moisture product**

ERA5-land is a reanalysis dataset produced by running land component of the ECMWF (European Centre for Medium-Range Weather Forecasts) ERA5 climate reanalysis (Albergel et al., 2018). ERA5-land provides SM data currently available from 1981 to present at hourly time interval with a spatial resolution of 0.1°. More information about the ERA5-land product can be referred to Muñoz-Sabater et al., (2018). The data (2009-2019) of volumetric total soil water content for the top soil layer (0-7 cm) in Maqu and Shiquanhe network areas is put in our dataset (Table 8).

Downloading online ERA5-land data through CDS (climate data store) website interface:

1. Register a Copernicus account.
2. Go to the https://cds.climate.copernicus.eu/cdsapp#!/dataset/reanalysis-era5-land?tab
3. The dataset catalogue entries include the following tabs:
   - **Overview:** It gives a description of the dataset and metadata information (e.g. data description and main variables).
   - **Download data:** It is a download web form.
   - **Quality assessment:** It is a new feature, work in progress (The CDS datasets are assessed by the Evaluation and Quality Control (EQC) function of C3S independently of the data supplier).
   - **Documentation:** It provides links to details documentation about the dataset.
4. Go to the download data tab to select the required data.
   - **Variable:** Select the Soil Water, Volumetric soil water layer 1
   - **Year:** Select 2009-2019
   - **Month:** Select all

- **Day:** Select all
- **Time:** Select all
- **Geographical area:** Select sub-region extraction, Maqu (33.5°-34.25° N, 101.63°-102.75° E), Shiquanhe (32.36°-32.76° N, 79.75°-80.25° E)
- **Format:** NetCDF

5. Click the **Submit Form** and wait for the request processing (about several hours), until the green button **download** appears, you can click it and download the data.

**4.2 MERRA2 soil moisture product**

MERRA2 is an atmospheric reanalysis dataset produced by NASA using the Goddard Earth Observing System Model version 5 (GEOS-5) and atmospheric data assimilation system (ADAS), version 5.12.4. MERRA2 provides SM data currently available from 1980 to present at hourly time interval with a spatial resolution of 0.5° (latitude) by 0.625° (longitude). More information about the MERRA2 product can be referred to GMAO (2015). The data (2009-2019) of volumetric liquid soil water content for the surface layer (0-5 cm) in Maqu and Shiquanhe network areas is put in our dataset (Table 8).

Downloading online MERRA2 data through GES DISC (Goddard Earth Sciences Data and Information Service Center) website interface:

1. Register an EARTHDATA account.
2. Go to the https://disc.gsfc.nasa.gov/datasets/M2T1NXLND_5.12.4/summary
3. The dataset catalogue entries include the following tabs:
   - **Product Summary:** It gives a description of the dataset and metadata information (e.g. temporal spatial, file format, etc.).
   - **Data citation:** To cite the data in publications.
   - **Documentation:** It provides links to details documentation about the dataset.
   - **Reference:** It is data collection reference.
4. Click the button of **Subset/ Get data** on the right of the interface to select the required data.
   - **Download Method:** Select the Get File Subsets using the GES DISC Subsetter
   - **Refine Date Range:** 2009-01-01 to 2019-12-31
   - **Refine Region:** Maqu (101.63, 33.5, 102.75, 34.25), Shiquanhe (79.75, 32.36, 80.25, 32.76)
   - **Variable:** Select SFMC = water surface layer
   - **Time of Day:** Get complete time span
   - **Grid:** bilinear interpolation on GLDAS-2_0.25
   - **Output format:** NetCDF
5. Click the **Get Data** and **Down load links list**, and then the tool like **Chrono Download Manager** can be used to download the data via the links list.

**4.3 GLDAS Noah soil moisture product**

GLDAS-2.1 Noah is forced by a combination of model-based and observation data including Global Precipitation Climatology Project (GPCP) version 1.3, and simulated with the Noah Model 3.6 in Land Information System (LIS) version 7. GLDAS-2.1 Noah provides SM data currently available from 2000 to present at 3-hourly time interval with a spatial resolution of 0.25°. More information about the GLDAS Noah product can be referred to Rodell et al. (2004). The data (2009-2019) of soil water content for the top soil layer (0-10 cm) in Maqu and Shiquanhe network areas is put in our dataset (Table 8).

Downloading online GLDAS Noah data through GES DISC (Goddard Earth Sciences Data and Information Service Center) website interface:

1. Register an EARTHDATA account.
2. Go to the https://disc.gsfc.nasa.gov/datasets/GLDAS_NOAH025_3H_2.1/summary
3. The dataset catalogue entries include the following tabs:
   - **Product Summary:** It gives a description of the dataset and metadata information (e.g. temporal spatial, file format, etc.).
   - **Data citation:** To cite the data in publications.
   - **Documentation:** It provides links to details documentation about the dataset.
   - **Reference:** It is data collection reference.
4. Click the button of **Subset/ Get data** on the right of the interface to select the required data.
   - **Download Method:** Select the Get File Subsets using the GES DISC Subsetter
   - **Refine Date Range:** 2009-01-01 to 2019-12-31
   - **Refine Region:** Maqu (101.63, 33.5, 102.75, 34.25), Shiquanhe (79.75, 32.36, 80.25, 32.76)
   - **Variable:** Select SoilMoi0_10cm_inst = Soil moisture content (0-10 cm underground) (kg m$^{-2}$)
   - **Time of Day:** Get complete time span
   - **Grid:** bilinear interpolation on GLDAS-2_0.25
   - **Output format:** NetCDF
5. Click the **Get Data** and **Down load links list**, and then the tool like **Chrono Download Manager** can be used to download the data via the links list.

**Table 8. Specification of the model-based soil moisture products.**

| Data set | Variable | Unit | Spatial | Temporal | Period | Reference |
|---|---|---|---|---|---|---|
| ERA5-land | Volumetric soil water layer 1 (swvl1) | m$^3$ m$^{-3}$ | 0.1° | Hourly | 1981~ | Muñoz-Sabater. et al., (2018) |
| GLDAS Noah | SoilMoi0_10cm_inst | kg m$^{-2}$ | 0.25° | 3-hourly | 2000 ~ | Rodell et al., (2004) |
| MERRA2 | Water surface layer (SFMC) | m$^3$ m$^{-3}$ | 0.5° ×0.625° | Hourly | 1980 ~ | GMAO (2015) |

**4.4 Precipitation data**

The precipitation data is from two weather stations, i.e. Maqu (34°00'N, 102°05'E) and Shiquanhe (32°30'N, 80°05'E), operated by the China Meteorological Administration (CMA) which provides the near-surface meteorological data of about 700 weather stations in China. The daily precipitation data can be downloaded from https://data.cma.cn/dataService/cdcindex/datacode/SURF_CLI_CHN_MUL_DAY.html that is in Chinese. The data is only available to agreement users, which is not allowed to be shared without permission from the CMA.

---

## Author Response (AR3)

Dear Editor,

Thank you very much for comments regarding to the precipitation data.

In this study, the monthly precipitation data is used to make the Mann Kendall test and Sen's slope estimate as described in Section 3.2, and the results are shown in Section 4.2. The trend analysis of the precipitation is merely used to compared with the soil moisture trend, and it is not a part of the upscaling data. The monthly precipitation used in this study has been included in the "Supplementary data" folder of the updated dataset in the 4TU.ResearchData repository.

The data publication (https://doi.org/10.4121/12763700.v5), corresponding part in user guide and manuscript have been revised based on the comments and we hope that you can consider it for publication in Earth System Science Data.

Sincerely,

Pei Zhang

On behalf of all co-authors

---

## Author Response (AR4)

Dear Editor,

Thank you very much for providing detailed comments regarding to the manuscript and dataset. The data publication and manuscript have been revised based on the comments.

Please find below our detailed response to each comment.

We think that the revised manuscript and data publication have appropriately addressed all the concerns and we hope that you can consider it for publication in Earth System Science Data.

Sincerely,

Pei Zhang

On behalf of all co-authors

In the text below we provide our response to each comment point by point.

Editor's comments are in **bold**.

Author's responses are in regular.

Author's additions/modifications in the text are in blue.

**The description of the hourly data still lacks metadata, additionally it is not clear which data exactly was used for the analysis. Could you please update the User guide in your repository and describe the data acquisition procedure in similar detail as you did for the soil moisture products (how to choose the station you used, which time frame, etc...).**

Thanks for the comments and suggestions. The precipitation data were mainly used for the trend analysis by the Mann Kendall trend test and Sen's slope estimate (see Section 3.2) based on the monthly cumulative values, and the results are shown in Fig. 9 of Section 4.2, while daily value is merely plotted in Fig. 8 as a reference for the soil moisture variation.

[Figure]

**Fig. 9. Mann Kendall trend test and Sen's slope estimate for precipitation, SM$_{AA-min}$, SM$_{AA-valid}$, and model-based SM derived from the ERA5-land, GLDAS Noah, and MERRA2 for a 10-year period for (a) Maqu and (b) Shiquanhe networks.**

The precipitation data part in user guide has been updated in the revised version. It should be pointed out that the precipitation data is not part of the upscaling soil moisture dataset, the objective of providing precipitation in this study is for the reference against the soil moisture seasonal dynamic and trend changes.

"**4.4 Precipitation data**

The precipitation data is from two weather stations, i.e. Maqu (34°00'N, 102°05'E) and Shiquanhe (32°30'N, 80°05'E), operated by the China Meteorological Administration (CMA) which provides the near-surface meteorological data of about 700 weather stations in China. The daily precipitation data can be downloaded from https://data.cma.cn/data/detail/dataCode/SURF_CLI_CHN_MUL_DAY.html. The monthly precipitation data that actually used in this study is contained in the "Supplementary data" folder.

Downloading precipitation data through CMDC (China Meteorological Data Service) website interface:

1. Register an CMA account.
2. Go to the https://data.cma.cn/data/detail/dataCode/SURF_CLI_CHN_MUL_DAY.html. The language of this web is Chinese.
3. The interface include the following parts:
   - **Product Summary:** It gives a description of the dataset (e.g. name, keywords, begin time, end time, frequency, share level, quality, etc.).
   - **Retrieve:** It is a retrieve web form for your require data.
   - **Metadata:** It gives the metadata of the dataset.
   - **Documentation:** It provides links to details documentation about the dataset.
4. Click the button of **Retrieve** to select the required data.
   - **TimeScope:** 2009.5.1-2019.4.31 (2010.8.1-2019.7.31)
   - **Select Station:** Select Station list, Gansu, [56074] Maqu (Select Station list, Qianghai, [55228] Shiquanhe)
   - **Element:** 20-20h precipitation
5. Click the bottom of **Retrieve** to submit the required data. The data is only available to agreement users."

**Manuscript section 3.2: You wrote in your response that the trend calculation for the precipitation data was done similarly to the soil moisture data. However, in the manuscript, there is no mention whatsoever about precipitation data in section 3.2. Please revise this so that it is clear that you did the trend analysis for both soil moisture and precipitation. An obvious way would be to describe the equations in general terms (x instead of SM as a variable) and then describe for which variables you did the analysis.**

Thanks for the comments and suggestions. The corresponding parts in Section 3.2 are revised:

Page 7 Line 220-222:

"The Mann-Kendall test and Sen's slope estimate (Gilbert, 1987; Mann, 1945; Smith et al., 2012) are adopted to analyze the trend of the 10-year time series for the upscaled SM, model-based SM products (i.e. ERA5-land, GLDAS Noah, and MERRA2), and precipitation."

Page 7 Line 226-230:

"For month $i$ (e.g. January), the statistics $S_i$ can be computed as:

$$S_i = \sum_{k=1}^{9} \sum_{l=k+1}^{10} sgn(X_{i,l} - X_{i,k}) \tag{2a}$$

$$sgn(X_{i,l} - X_{i,k}) = \begin{cases} 1 & X_{i,l} > X_{i,k} \\ 0 & X_{i,l} = X_{i,k} \\ -1 & X_{i,l} < X_{i,k} \end{cases}$$

where $k$ and $l$ represent the different year and $l > k$, $X_{i,l}$ and $X_{i,k}$ represent the monthly value of the variable for the month $i$ of the year $k$ and $l$, respectively."

Page 8 Line 247-249:

"If the trend shows upward or downward, we will further estimate the slope (change per unit time) with Sen's method (Sen, 1968). The slopes of each month can be calculated as:

$$Q_i = \frac{X_{i,l} - X_{i,k}}{l - k} \tag{2f}"$$

**Monthly precipitation data: It is a good idea to include a slightly aggregated data product if you can't supply the original data that you used in the analysis. However, it is not clear if monthly data was used at all and where, or if it is simply provided as an informative dataset because the daily values are not accessible. Please make this very clear in the text.**

Thanks for the comments and suggestions. The description for the utilization of the monthly precipitation data application in this study is revised on Page 5 Line 170-171:

"The monthly precipitation data for the period between 2009 and 2019 is used for the trend analysis by Mann Kendall trend test and Sen's slope estimate in this study (see Section 4.2)."

**Furthermore, there is only very little metadata about this dataset except the coordinates. What kind of instruments were used (maybe there's a standard in the CMA, but it would be good to have more information, similar to the level of detail you provide for soil moisture)?**

Thanks for the comments and suggestions. The specific information of the instruments are not provided from the official website, and all available information for the weather station is included to Section 2.2 in the revised manuscript on Page 5 Line 159-165:

"Precipitation data is available from the dataset of daily climate data from Chinese surface meteorological stations. This dataset is maintained by the China Meteorological Administration (CMA) and based on the measurements from 756 basic and reference surface meteorological observation and automatic weather stations (AWS) in China from 1951 to present. The online dataset mainly includes seven meteorological variables such as air pressure, air temperature, relative humidity, wind speed, evaporation, sunshine duration, and precipitation. The precipitation data from two weather stations (see Fig. 1), i.e. Maqu (34°00'N, 102°05'E) and Shiquanhe (32°30'N, 80°05'E) are used in this study."

**Additionally, it is not clear how you aggregated the data to monthly values. It seems like they are averages of each month (because the numbers are too low compared to your annual precipitation amounts in section 2.1), which would be very unusual for precipitation data. In any case, you should explain why you have done monthly values, give sufficient metadata (comparable to the soil moisture metadata), describe what kind of aggregation you used and why, and where you use this data.**

The monthly value is monthly cumulative precipitation that is the sum of the daily precipitation of each month. It has been updated in the dataset, and please find the details in the response to previous comment.

The corresponding description is also included in the revised manuscript on Page 5 Line 165-171:

"The available daily precipitation is the cumulative value for the period between 20h of previous day and 20h of current day at Beijing time, which is available from https://data.cma.cn/data/detail/dataCode/SURF_CLI_ CHN_ MUL_DAY.html (last access 11 March 2021). The daily precipitation is summed up for each month to obtain the monthly cumulative value in this study, which can be found at https://doi.org/10.4121/uuid:21220b23-ff36-4ca9-

a08f-ccd53782e834 (last access 16 April 2021). The monthly precipitation data for the period between 2009 and 2019 is mainly used in this study for the trend analysis (see Section 4.2)."

The reason for using the monthly precipitation value to conduct trend analysis is described in Section 3.2:

"The Mann-Kendall test and Sen's slope estimate (Gilbert, 1987; Mann, 1945; Smith et al., 2012) are adopted to analyze the trend of the 10-year time series for the upscaled SM, model-based SM products (i.e. ERA5-land, GLDAS Noah, and MERRA2), and precipitation. Specifically, the trend analysis is based on the monthly data, and all the missing data is regarded as an equal value smaller than other valid data. The test consists of calculating the seasonal statistics S and its variance VAR(S) separately for each month during the 10-year period, and the seasonal  statistics are then summed to obtain the Z metric."

**Also, while skimming through the manuscript again, I noticed a number of typos and phrasing issues. Please consult an English native speaker to check for those so that the manuscript can be read easily.**

Thanks for the suggestion. The revised manuscript has been checked by the co-author Rogier van der Vdelde who have stayed in the USA for about three years. We think the typos and phrasing issues has been addressed.

---

## Author Response (AR5)

Dear Editor,

Thank you very much for providing detailed comments regarding to the manuscript in this round. The manuscript and data publication have been revised based on the comments.

Please find below our detailed response to each comment.

We think that the revised manuscript has appropriately addressed all the concerns and we hope that you can consider it for publication in Earth System Science Data.

Sincerely,

Pei Zhang

On behalf of all co-authors

In the text below we provide our response to each comment point by point.

Editor's comments are in **bold**.

Author's responses are in regular.

Author's additions/modifications in the text are in blue.

**Please remove the "SM" in line 244, as by now the sentence refers to both SM and precip time series.**

Thanks for the suggestion. The "SM" has been removed in revised manuscript on Page 8 Line 245:

"the trend of the time series is regarded as upward (downward) at the significance level of $\alpha$."

**It would be good to also include the mean annual precipitation for the Maqu network in section 2.1.1 as you do it for the other two networks and also provide the monthly data for Maqu now. Could you please add this value?**

Thanks for the comment and suggestion. The mean annual precipitation for the Maqu network has been added on Page 4 Line 111-112:

"The annual precipitation is about 600 mm that falls mainly in the warm season (May-October)."

In addition, the monthly precipitation data for the Naqu network is also added in the data publication along with other two networks:

| \ Supplementary data \ | \Precipitation.xlsx | - Maqu: This sheet contains monthly precipitation of Maqu weather station (Fig. 1) between 5/2009 and 4/2019
- Shiquanhe: This sheet contains monthly precipitation of Shiquanhe weather station (Fig. 1) between 8/2010 and 7/2019
- Naqu: This sheet contains monthly precipitation of Naqu weather station (Fig. 1) between 1/2009 and 12/2019 |
|---|---|---|